# Large but decreasing effect of ozone on the European carbon sink

Rebecca J Oliver[1], Lina M Mercado[1,2], Stephen Sitch[2], David Simpson[3,4], Belinda E Medlyn[5], Yan-Shih Lin[5], Gerd A Folberth[6]

[1] Centre for Ecology and Hydrology, Benson Lane, Wallingford, OX10 8BB, UK

[2] College of Life and Environmental Sciences, University of Exeter, EX4 4RJ, Exeter, UK

[3] EMEP MSC-W Norwegian Meteorological Institute, PB 43, NO-0313, Oslo, Norway

[4] Dept. Space, Earth & Environment, Chalmers University of Technology, Gothenburg, SE-41296 Sweden

[5] Hawkesbury Institute for the Environment, Western Sydney University, Locked Bag 1797, Penrith NSW 2751 Australia

[6] Met Office Hadley Centre, Exeter, UK.

*Correspondence to*: Rebecca Oliver (rfu@ceh.ac.uk)

**Abstract**
The capacity of the terrestrial biosphere to sequester carbon and mitigate climate change is governed by the ability
of vegetation to remove emissions of $CO_2$ through photosynthesis. Tropospheric $O_3$, a globally abundant and
potent greenhouse gas, is, however, known to damage plants, causing reductions in primary productivity. Despite
emission control policies across Europe, background concentrations of tropospheric $O_3$ have risen significantly
over the last decades due to hemispheric-scale increases in $O_3$ and its precursors. Therefore, plants are exposed to
increasing background concentrations, at levels currently causing chronic damage.  Studying the impact of $O_3$ on
European vegetation at the regional scale is important for gaining greater understanding of the impact of $O_3$ on
the land carbon sink at large spatial scales. In this work we take a regional approach and update the JULES land-
surface model using new measurements specifically for European vegetation. Given the importance of stomatal
conductance in determining the flux of $O_3$ into plants, we implement an alternative stomatal closure
parameterization and account for diurnal variations in $O_3$ concentration in our simulations. We conduct our
analysis specifically for the European region to quantify the impact of tropospheric $O_3$, and its interaction with
$CO_2$, on gross primary productivity (GPP) and land carbon storage across Europe. A factorial set of model
experiments showed that tropospheric $O_3$ can suppress terrestrial carbon uptake across Europe over the period
1901 to 2050. By 2050, simulated GPP was reduced by 4 to 9% due to plant $O_3$ damage and land carbon storage
by 3 to 7%. The combined physiological effects of elevated future $CO_2$ (acting to reduce stomatal opening) and
reductions in $O_3$ concentrations resulted in reduced $O_3$ damage in the future. This alleviation of $O_3$ damage by
$CO_2$ induced stomatal closure was around 1 to 2% for low and high sensitivity respectively (on both land carbon
and GPP). Reduced land carbon storage resulted from diminished soil carbon stocks consistent with the reduction
in GPP. Regional variations are identified with larger impacts shown for temperate Europe (GPP reduced by 10
to 20%) compared to boreal regions (GPP reduced by 2 to 8%). These results highlight that $O_3$ damage needs to
be considered when predicting GPP and land carbon, and that the effects of $O_3$ on plant physiology need to be
considered in regional land carbon cycle assessments.








## 1 Introduction

The terrestrial biosphere absorbs around 30% of anthropogenic $CO_2$ emissions and acts to mitigate climate change Le Quéré et al. (2015). Early estimates of the European carbon balance suggest a terrestrial carbon sink of between 135 to 205 TgC yr$^{-1}$ (Janssens et al., 2003). Schulze et al. (2009) determined a larger carbon sink of 274 TgC yr$^{-1}$, and more recent estimates suggest a European terrestrial sink of between 146 to 184 TgC yr$^{-1}$ (Luyssaert et al., 2012). The carbon sink capacity of land ecosystems is dominated by the ability of vegetation to sequester carbon through photosynthesis and release it back to the atmosphere through respiration. Therefore, any change in the balance of these fluxes will alter ecosystem source-sink behaviour.

In recent decades much attention has focussed on the effects of rising atmospheric $CO_2$ on vegetation productivity (Ceulemans and Mousseau, 1994;Norby et al., 2005;Norby et al., 1999;Saxe et al., 1998). The Norby et al. (2005) synthesis of Free Air $CO_2$ Enrichment (FACE) experiments suggests a median stimulation (23 ± 2%) of forest NPP in response to a doubling of $CO_2$. Similar average increases (20%) were observed for $C_3$ crops, although this translated into smaller gains in biomass (17%) and crop yields (13%) (Long et al., 2006). Little attention, however, has been given to tropospheric ozone ($O_3$), a globally abundant air pollutant recognised as one of the most damaging pollutants for forests (Karlsson et al., 2007;Royal-Society, 2008;Simpson et al., 2014b). Tropospheric $O_3$ is a secondary air pollutant formed by photochemical reactions involving carbon monoxide (CO), volatile organic compounds (VOCs), methane ($CH_4$) and nitrogen oxides ($NO_x$) from both man-made and natural sources, as well as downward transport from the stratosphere and lightning which is a source of $NO_x$. The phytotoxic effects of $O_3$ exposure are shown to decrease vegetation productivity and biomass, with consequences for terrestrial carbon sequestration (Felzer et al., 2004;Loya et al., 2003;Mills et al., 2011b;Sitch et al., 2007). Few studies, however, consider the simultaneous effects of exposure to both gases, and few Earth-system models (ESMs) currently explicitly consider the role of tropospheric $O_3$ in terrestrial carbon dynamics (IPCC, 2013), both of which are important to understanding the carbon sequestration potential of the land-surface, and future carbon dynamics regionally and globally (Le Quéré et al., 2016;Sitch et al., 2015).

Due to increased anthropogenic precursor emissions over the industrial period, background concentrations of ground-level $O_3$ have risen (Vingarzan, 2004). $O_3$ levels at the start of the 20[th] century are estimated to be around 10 ppb for the site Montsouris Observatory near Paris, data for Arkona on the Baltic coast increased from ca. 15 ppb in the 1950s to 20-27 ppb by the early 1980s, and the Irish coast site Mace Head shows around 40 ppb by the year 2000 (Logan et al., 2012;Parrish et al., 2012). Present day annual average background $O_3$ concentrations reported in the review of Vingarzan (2004) show $O_3$ concentrations range between approximately 20 and 45 ppb, with the greatest increase occurring since the 1950s. Trends vary from site to site though, even on a decadal basis (Logan et al., 2012;Simpson et al., 2014b), depending, for example, on local/regional trends in precursor (especially NOx) emissions, elevation, and exposure to long-range transport. Nevertheless, there is some indication that background $O_3$ levels over the mid-latitudes of the Northern Hemisphere have continued to rise at a rate of approximately 0.5–2% per year, although not uniform (Vingarzan, 2004). As a result of controls on precursor emissions in Europe and North America, peak $O_3$ concentrations in these regions have decreased or stabilised over recent decades (Cooper et al., 2014;Logan et al., 2012;Parrish et al., 2012;Simpson et al., 2014b).

Nevertheless, climate change may increase the frequency of weather events conducive to peak $O_3$ incidents in the
future (e.g. summer droughts and heat-waves; e.g., (Sicard et al., 2013)), and may increase biogenic emissions of
the $O_3$-precursors isoprene and $NO_x$, although such impacts are subject to great uncertainty (Simpson et al.,
2014b;Young et al., 2013;Young et al., 2009). Intercontinental transport of air pollution from regions such as Asia
that currently have poor emission controls are thought to contribute substantially to rising background $O_3$
concentrations over the last decades (Cooper et al., 2010;Verstraeten et al., 2015). Northern Hemisphere
background concentrations of $O_3$ are now close to established levels for impacts on human health and the terrestrial
environment (Royal-Society, 2008). Therefore, although peak $O_3$ concentrations are in decline across Europe,
plants are exposed to increasing background levels, at levels currently causing chronic damage (Mills et al.,
2011b). Intercontinental transport means future $O_3$ concentrations in Europe will be partly dependent on how $O_3$
precursor emissions evolve globally.
Elevated $O_3$ concentrations impact agricultural yields and nutritional quality of major crops (Ainsworth et al.,
2012;Avnery et al., 2011), with consequences for global food security (Tai et al., 2014). As well as being a
significant air pollutant, $O_3$ is a potent greenhouse gas (Royal-Society, 2008). High levels of $O_3$ are damaging to
ecosystem health and reduce the global land carbon sink (Arneth et al., 2010;Sitch et al., 2007). Reduced uptake
of carbon by plant photosynthesis due to $O_3$ damage allows more $CO_2$ to remain in the atmosphere. This effect of
$O_3$ on plant physiology represents an additional climate warming to the direct radiative forcing of $O_3$ (Collins et
al., 2010;Sitch et al., 2007), the magnitude of which, however, remains highly uncertain (IPCC, 2013).
Dry deposition of $O_3$ to terrestrial surfaces, primarily uptake by stomata on plant foliage and deposition on external
surfaces of vegetation (Fowler et al., 2001;Fowler et al., 2009), is a large sink for ground level $O_3$ (Wild,
2007;Young et al., 2013). On entry to sub-stomatal spaces, $O_3$ reacts with other molecules to form reactive oxygen
species (ROS). Plants can tolerate a certain level of $O_3$ depending on their capacity to scavenge and detoxify the
ROS (Ainsworth et al., 2012). Above this critical level, long-term chronic $O_3$ exposure reduces plant
photosynthesis and biomass accumulation (Ainsworth, 2008;Ainsworth et al., 2012;Matyssek et al., 2010a;Wittig
et al., 2007;Wittig et al., 2009), either directly through effects on photosynthetic machinery such as reduced
Rubisco content (Ainsworth et al., 2012;Wittig et al., 2009) and/or indirectly by reduced stomatal conductance
($g_s$) (Kitao et al., 2009;Wittig et al., 2007), alters carbon allocation to different pools (Grantz et al., 2006;Wittig
et al., 2009), accelerates leaf senescence (Ainsworth, 2008;Nunn et al., 2005;Wittig et al., 2009) and changes plant
susceptibility to biotic stress factors (Karnosky et al., 2002;Percy et al., 2002).
The response of plants to $O_3$ is very wide ranging as reported in the literature from different field studies. The
Wittig et al. (2007) meta-analysis of temperate and boreal tree species showed future concentrations of $O_3$
predicted for 2050 significantly reduced leaf level light saturated net photosynthetic uptake (-19%, range: -3% to
-28% at a mean $O_3$ concentration of 85 ppb) and $g_s$ (-10%, range: +5% to -23% at a mean $O_3$ concentration of 91
ppb) in both broadleaf and needle leaf tree species. In the Feng et al. (2008) meta-analysis of wheat, projected $O_3$
concentrations for the future reduced aboveground biomass (-18% at a mean $O_3$ concentration of 70 ppb)
photosynthetic rate (-20% at a mean $O_3$ concentration of 73 ppb) and $g_s$ (-22% at a mean $O_3$ concentration of 79
ppb). One of few long-term field based $O_3$ exposure studies (AspenFACE) showed that after 11 years of exposing

mature trees to elevated $O_3$ concentrations (mean $O_3$ concentration of 46 ppb), $O_3$ decreased ecosystem carbon content (-9%), and decreased NPP (-10%), although the $O_3$ effect decreased through time (Talhelm et al., 2014). Zak et al. (2011) showed this was partly due to a shift in community structure as $O_3$-tolerant species, competitively inferior in low $O_3$ environments, out competed $O_3$-sensitivie species. GPP was reduced (-12% to -19%) at two Mediterranean ecosystems exposed to high ambient $O_3$ concentrations (ranging between 20 to 72 ppb across sites and through the year) studied by Fares et al. (2013). Biomass of mature beech trees was reduced (-44%) after 8 years of exposure to elevated $O_3$ (~150 ppb) (Matyssek et al., 2010a). After 5 years of $O_3$ exposure (ambient +20 to +40 ppb) in a semi-natural grassland, annual biomass production was reduced (-23%), and in a Mediterranean annual pasture $O_3$ exposure significantly reduced total aboveground biomass (up to -25%) (Calvete-Sogo et al., 2014). However, these were empirical studies at individual sites, and these focus on $O_3$ effects on plant physiology and productivity, but do not quantify the impact on the land carbon sink. Modelling studies are needed to scale site observations to the regional and global scales. Models generally suggest that plant productivity and carbon sequestration will decrease with $O_3$ pollution, though the magnitudes vary. For example, based on a limited dataset to parameterise plant $O_3$ damage for a global set of plant functional types, Sitch et al. (2007) predicted a decline in global GPP of 14 to 23% by 2100. A second study by Lombardozzi et al. (2015) similarly predicted a 10.8% decrease of global GPP. Here we take a regional approach and take advantage of the latest measurements showing changes in plant productivity with accumulated exposure to $O_3$ specifically for a range of European vegetation from different regions (CLRTAP 2017) with which to calibrate the JULES model for plant sensitivity to $O_3$, and conduct our analysis specifically for the European region.

Understanding the response of plants to elevated tropospheric $O_3$ is challenged by the large variation in $O_3$ sensitivity both within and between species (Karnosky et al., 2007;Kubiske et al., 2007;Wittig et al., 2009). Additionally, other environmental stresses that affect stomatal behaviour will affect the rate of $O_3$ uptake and therefore the response to $O_3$ exposure, such as high temperature, drought and changing concentrations of atmospheric $CO_2$ (Mills et al., 2016;Fagnano et al., 2009;Kitao et al., 2009;Löw et al., 2006). Increasing concentrations of atmospheric $CO_2$, for example, are suggested to provide some protection against $O_3$ damage by causing stomata to close (Harmens et al., 2007;Wittig et al., 2007), however the long-term effects of $CO_2$ fertilisation on plant growth and carbon storage remain uncertain (Baig et al., 2015;Ciais et al., 2013). Further, in some studies, stomata have been shown to respond sluggishly, losing their responsiveness to environmental stimuli with exposure to $O_3$ which can lead to higher $O_3$ uptake, increased water-loss and therefore greater vulnerability to environmental stresses such as drought (Mills et al., 2016;Mills et al., 2009;Paoletti and Grulke, 2010;Wilkinson and Davies, 2009).

Given the critical role $g_s$ plays in the uptake of both $CO_2$ and $O_3$, we use an alternative representation and parameterisation of $g_s$ in JULES by implementing the Medlyn *et al.* (2011) $g_s$ formulation. This model is based on the optimal theory of stomatal behaviour and has advantages over the current JULES $g_s$ formulation of Jacobs (1994) including i) a single parameter ($g_1$) compared to two parameters in Jacobs (1994), ii) the $g_1$ parameter is related to the water-use strategy of vegetation and is easier to parameterise with commonly measured leaf or canopy level observations of photosynthesis, $g_s$ and humidity,  and (iii) values of $g_1$ are available for many different plant functional types (PFTs) derived from a global data set of leaf-level measurements (Lin et al., 2015).


The main objective of this work is to assess the impact of historical and projected (1901 to 2050) changes in
tropospheric $O_3$ and atmospheric $CO_2$ concentration on predicted GPP and the land-carbon sink for Europe.
These are the two greenhouse gases that directly affect plant photosynthesis and $g_s$. We use a factorial suite of
model experiments, using the Joint UK land environment simulator (JULES) (Best et al., 2011;Clark et al.,
2011), the land-surface model of the UK Earth System Model (UKESM) (Collins et al., 2011) to simulate plant
$O_3$ uptake and damage, and to investigate the impact of both $O_3$ and $CO_2$ on plant water-use and carbon uptake.
In this work, the JULES model is re-calibrated using the latest observations of vegetation sensitivity to $O_3$, with
the addition of a separate parameterisation for temperate/boreal regions versus the Mediterranean. The $O_3$
sensitivity of each PFT in JULES was re-calibrated for both a high and low sensitivity to account for uncertainty
in the $O_3$ response, in part due to the  observed variation in $O_3$ sensitivity between species. This includes $O_3$
sensitivities for agricultural crops (wheat – high sensitivity) versus natural grassland (low sensitivity), with
separate sensitivities for Mediterranean grasslands. For forests JULES is parameterised with $O_3$ sensitivities for
broadleaf and needle leaf trees (with a high and low $O_3$ sensitivity for both), with separate sensitivities (high and
low) for Mediterranean broadleaf species. We make a separate distinction for the Mediterranean region where
possible because the work of Büker et al. (2015) showed that different $O_3$ dose-response relationships are
needed to describe the $O_3$ sensitivity of dominant Mediterranean trees. In addition, we introduce an alternative $g_s$
scheme into JULES as described above. JULES is forced with spatially varying daily $O_3$ concentrations from a
high resolution atmospheric chemistry model for Europe that are disaggregated to hourly concentrations,
therefore our simulations account for diurnal variations in $O_3$ concentration and $O_3$ responses allowing for
improved estimates of $O_3$ uptake by vegetation. We do not attempt to make a full assessment of the carbon cycle
of Europe, instead we target $O_3$ damage, which is currently a missing component in earlier carbon cycle
assessments (Le Quéré et al., 2017;Sitch et al., 2015). To this end, we prescribe changing $O_3$ and $CO_2$
concentrations from 1901 to 2050, but use a fixed pre-industrial climate. We acknowledge the use of a 'fixed'
pre-industrial climate omits the additional uncertainty of the interaction between climate change and $g_s$ which
will affect the rate of $O_3$ uptake and therefore $O_3$ concentrations. In addition, using uncoupled chemistry and
climate is a further source of uncertainty. To understand the impact of these complex feedback mechanisms is
an important area for future work, but in the current study our aim is to isolate the physiological response of
plants to both $O_3$ and $CO_2$, and determine the sensitivity of predicted GPP and the land carbon sink to this
process, as the impact of $O_3$ on the land carbon sink currently remains largely unknown at large spatial scales
for Europe.



**2 Methods**

**2.1 Representation of $O_3$ effects in JULES**

JULES calculates the land-atmosphere exchanges of heat, energy, mass, momentum and carbon on a sub-daily
time step, and includes a dynamic vegetation model (Best et al., 2011;Clark et al., 2011;Cox, 2001). This work
uses JULES version 3.3 (http://www.jchmr.org) at 0.5° x 0.5° spatial resolution and hourly model time step, the
spatial domain is shown in Fig. S5. JULES has a multi-layer canopy radiation interception and photosynthesis
scheme (10 layers in this instance) that accounts for direct and diffuse radiation, sun fleck penetration through the
canopy, inhibition of leaf respiration in the light and change in photosynthetic capacity with depth into the canopy
(Clark et al., 2011;Mercado et al., 2009). Soil water content also affects the rate of photosynthesis and $g_s$. It is
modelled using a dimensionless soil water stress factor, β, which is related to the mean soil water concentration
in the root zone, and the soil water contents at the critical and wilting point (Best *et al.*, 2011).

To simulate the effects of $O_3$ deposition on vegetation productivity and water use, JULES uses the flux-gradient
approach of Sitch *et al.*, (2007), modified to include non-stomatal deposition following Tuovinen et al. (2009). A
similar approach is taken by Franz et al. (2017) in the OCN model, however plant $O_3$ damage is a function of
accumulated $O_3$ exposure over time. In JULES, plant $O_3$ damage is instantaneous, the degree to which
photosynthesis and $g_s$ are modified at each time step with $O_3$ exposure having already been calibrated against
observations of the change in plant productivity with cumulative $O_3$ exposure for each PFT (i.e. $O_3$ dose-response
functions described later). JULES uses a coupled model of $g_s$ and photosynthesis, the potential net photosynthetic
rate ($A_p$, mol $CO_2$ m$^{-2}$ s$^{-1}$) is modified by an 'O$_3$ uptake' factor (*F,* the fractional reduction in photosynthesis), so
that the actual net photosynthesis ($A_{net}$, mol $CO_2$ m$^{-2}$ s$^{-1}$) is given by equation 1 (Clark *et al.,* 2011, Sitch *et al.,*
2007). Because of the relationship between these two fluxes, the direct effect of $O_3$ damage on photosynthetic rate
also leads to a reduction in $g_s$. An alternative approach was taken by Lombardozzi et al. (2012) in the CLM model
where photosynthesis and $g_s$ are decoupled, so that $O_3$ exposure affects carbon assimilation and transpiration
independently. In JULES, changes in atmospheric $CO_2$ concentration also affect photosynthetic rate and $g_s$,
consequently the interaction between changing concentrations of both $CO_2$ and $O_3$ is allowed for.

$$A_{net} = A_P F \tag{1}$$

The $O_3$ uptake factor (*F*) is defined as:

$$F = 1 - a * max[F_{O3} - F_{O3crit}, 0.0] \tag{2}$$

$F_{O3}$ is the instantaneous leaf uptake of $O_3$ (nmol m$^{-2}$ s$^{-1}$), $F_{O3crit}$ is a PFT-specific threshold for $O_3$ damage (nmol
m$^{-2}$ PLA s$^{-1}$, projected leaf area), and '*a*' is a PFT-specific parameter representing the fractional reduction of
photosynthesis with $O_3$ uptake by leaves. Following Tuovinen et al. (2009), the flux of $O_3$ through stomata, $F_{O3}$,
is represented as follows:

$$F_{O3} = O_3 \left( \frac{g_b \left( \frac{g_L}{K_{O3}} \right)}{g_b + \left( \frac{g_L}{K_{O3}} \right) + g_{ext}} \right) \tag{3a}$$

$O_3$ is the molar concentration of $O_3$ at reference (canopy) level (nmol m$^{-3}$), $g_b$ is the leaf-scale boundary layer
conductance (m s$^{-1}$, eq 3b), $g_l$ is the leaf conductance for water (m s$^{-1}$), $K_{o3}$ accounts for the different diffusivity of
ozone to water vapour and takes a value of 1.51 after Massman (1998), and $g_{ext}$ is the leaf-scale non-stomatal
deposition to external plant surfaces (m s$^{-1}$) which takes a constant value of 0.0004 m s$^{-1}$ after Tuovinen et al.
(2009). The leaf-level boundary layer conductance ($g_b$) is calculated as in Tuovinen *et al.* (2009)

$$g_b = \alpha L d^{-1/2} U^{-1/2} \tag{3b}$$

$\alpha$ is a constant (0.0051 m s$^{-1/2}$), $Ld$ is the cross-wind leaf dimension (m) defined per PFT as 0.05 for trees, 0.02
for grasses (C$_3$ and C$_4$) and 0.04 for shrubs, $U$ is wind speed at canopy height (m s$^{-1}$). The rate of O$_3$ uptake is
dependent on $g_s$, which is dependent on photosynthetic rate. Given $g_s$ is a linear function of photosynthetic rate in
JULES (Clark et al., 2011), from eq 1 it follows that:

$$g_s = g_l F \tag{4}$$

The O$_3$ flux to stomata, $F_{O3}$, is calculated at leaf level and then scaled to each canopy layer differentiating sunlit
and shaded leaf photosynthesis, and finally summed up to the canopy level. Because the photosynthetic capacity,
photosynthesis and therefore $g_s$ decline with depth into the canopy, this in turn affects O$_3$ uptake, with the top leaf
level contributing most to the total O$_3$ flux and the lowest level contributing least.

**2.2 Calibration of O$_3$ uptake model**

Here we use the latest literature on flux based O$_3$ dose-response relationships derived from observed field data
across Europe (CLRTAP, 2017) to determine the key PFT-specific O$_3$ sensitivity parameters in  JULES (*a* and
*Fo$_{3crit}$*). Synthesis of information expressed as O$_3$ flux based dose-response relationships derived from field
experiments is carried out by The United Nations Convention on Long-Range Transboundary Air Pollution
(CLRTAP Convention), this information is then used as a policy tool to inform emission reduction strategies in
Europe to improve air quality (CLRTAP, 2017;Mills et al., 2011a). Derivation of O$_3$ flux based dose-response
relationships for different vegetation types uses the accumulated stomatal O$_3$ flux above a threshold (often referred
to as the phytotoxic O$_3$ dose above a threshold of 'y' i.e. POD$_y$) as the dose metric, and the percentage change in
biomass as the response metric (Emberson et al., 2007;Karlsson et al., 2007). We use these observation based O$_3$
dose-response relationships to calibrate each JULES PFT for sensitivity to O$_3$ using available relationships for the
closest matching vegetation type. For JULES, *Fo$_{3crit}$* is the threshold for O$_3$ damage, and values for this parameter
are taken from the O$_3$ dose-response relationships as the POD$_y$ value. The actual sensitivity to O$_3$ is determined
by the slope of the O$_3$ dose-response relationship, i.e. how much biomass changes with accumulated stomatal
uptake of O$_3$ above the damage threshold, this relates to the parameter *a* in JULES. The parameter '*a*' is a PFT-
specific parameter representing the fractional reduction of photosynthesis with O$_3$ uptake by leaves. Values for
this parameter are found for each PFT by running JULES with different values of '*a*', which alter the instantaneous
photosynthetic rate, but then calculating the accumulated stomatal flux of O$_3$ and the change in productivity, until
the slope of this relationship produced by the JULES simulations matches that of the O$_3$ dose-response
relationships derived from observations. Essentially we calibrate each JULES PFT for sensitivity to O$_3$ by
reproducing the observation-based O$_3$ dose-response relationships.

Each PFT was calibrated for a high and low plant $O_3$ sensitivity to account for uncertainty in the sensitivity of
different plant species to $O_3$, using the approach of Sitch *et al*., (2007). Therefore, when using our results to assess
the impact of $O_3$ at the land surface, we are able to provide a range in our estimates to help address some of the
uncertainty in the $O_3$ response of different vegetation types. In addition, where possible owing to available data,
a distinction was made for Mediterranean regions. This was because the work of Büker et al. (2015) showed that
different $O_3$ dose-response relationships are needed to describe the $O_3$ sensitivity of dominant Mediterranean trees.
For the $C_3$ herbaceous PFT, the dominant land cover type across the European domain in this study (Fig. S1), the
high plant $O_3$ sensitivity was calibrated against observations for wheat to give a representation of agricultural
regions and wheat is one of the most sensitive grasses to $O_3$ (Fig. S2, Table S1). For the low plant $O_3$ sensitivity
JULES was calibrated against the dose-response function for natural grassland to give a representation of natural
grassland and this vegetation has a much lower sensitivity to $O_3$ damage, for the Mediterranean region we used a
function for Mediterranean natural grasslands, all taken from CLRTAP (2017) (Fig. S2, Table S1). Tree/shrub
PFTs were calibrated against observed $O_3$ dose-response functions for the high plant $O_3$ sensitivity: broadleaf
trees (temperate/boreal) = Birch/Beech dose-response relationship, broadleaf trees (Mediterranean) = deciduous
oaks dose-response relationship, needle leaf trees = Norway spruce dose-response relationship, shrubs =
Birch/Beech dose-response relationship, all from CLRTAP (2017) (Fig. S2, Table S1). Data on $O_3$ dose-response
relationships for different vegetation types is very limited, therefore for the low plant $O_3$ sensitivity calibration for
trees/shrubs we assumed a 20% decrease in sensitivity to $O_3$ based on the difference in sensitivity between high
and low sensitive tree species in the Karlsson et al. (2007) study. Due to limitations in data availability, the shrub
parameterisation uses the observed dose-response functions for broadleaf trees. Similarly, the parameterisation
for $C_4$ herbaceous uses the observed dose-responses for $C_3$ herbaceous, however the fractional cover of $C_4$ herbs
across Europe is low (Fig. S1), so this assumption affects a very small percentage of land cover.

To calibrate the JULES $O_3$ uptake model, JULES was run across Europe forced using the WFDEI observational
climate dataset (Weedon, 2013) at 0.5º X 0.5º spatial and three hour temporal resolution. JULES uses interpolation
to disaggregate the forcing data down from 3 hours to an hourly model time step. The model was spun-up over
the period 1979 to 1999 with a fixed atmospheric $CO_2$ concentration of 368.33 ppm (1999 value from Mauna Loa
observations, (Tans and Keeling)). Zero tropospheric ozone concentration was assumed for the control simulation,
for the simulations with $O_3$, spin-up used spatially explicit fields of present day $O_3$ concentration produced using
the UK Chemistry and Aerosol (UKCA) model with standard chemistry from the run evaluated by O'Connor et
al. (2014). A fixed land cover map was used based on IGBP (International Geosphere-Biosphere Programme)
land cover classes (IGBP-DIS), therefore as the vegetation distribution was fixed and the calibration was not
looking at carbon stores, a short spin-up was adequate to equilibrate soil temperature and soil moisture. JULES
was then run for the year 2000 with a corresponding $CO_2$ concentration of 369.52 ppm (from Mauna Loa
observations, (Tans and Keeling)) and monthly fields of spatially explicit tropospheric $O_3$ (O'Connor et al., 2014)
as necessary.

Calibration was performed using four simulations: with i) zero tropospheric $O_3$ concentration, this was the control
simulation (control), ii) tropospheric $O_3$ at current ambient concentration (O3), iii) ambient +20 ppb (O3+20) and
iv) ambient +40 ppb (O3+40). The different $O_3$ simulations (i.e. O3, O3+20 and O3+40) were used to capture the
range of $O_3$ conditions in the data used in the observation-based $O_3$ dose-response relationships used in this study
for calibration, often data were from experiments using artificially manipulated conditions of ambient + 40 ppb
$O_3$ for example. For each JULES $O_3$ simulation, the value of $F_{O3crit}$ was taken from the vegetation specific $O_3$
dose-response relationship as the threshold $O_3$ concentration above which damage to vegetation occurs. An initial
estimate of the parameter '$a$' was used, then for each PFT and each simulation, hourly estimates of NPP (our
proxy for biomass – although not identical they are related) and $O_3$ uptake in excess of $F_{O3crit}$ were accumulated
over a PFT dependent accumulation period. The accumulation periods were ~6 months for broadleaf trees and
shrubs, all year for needle leaf trees, and ~3 months for herbaceous species, through the growing season, following
guidelines in CLRTAP (2017). Additionally, in accordance with the methods used in the CLRTAP (2017) that
describe how the $O_3$ dose-response relationships are derived from observations, we use the stomatal $O_3$ flux per
projected leaf area to top canopy sunlit leaves. The percentage change in total NPP was calculated for each $O_3$
simulation and plotted against the cumulative uptake of $O_3$ over the PFT-specific accumulation period. The linear
regression of this relationship was calculated, and slope and intercept compared against the slope and intercept of
the observed dose-response relationships. Values of the parameter 'a' were adjusted, and the procedure repeated
until the linear regression through the simulation points matched that of the observations (Fig. S2, Table S1).

**2.3 Representation of stomatal conductance and site level evaluation**

In JULES, $g_s$ (m s$^{-1}$) is represented following the closure proposed by (Jacobs, 1994):

$$g_s = 1.6RT_l \frac{A_{net}\beta}{c_a - c_i} \tag{5}$$

In this parameterisation, $c_i$ is unknown and in the default JULES model is calculated as in equation 6, hereafter
called JAC:

$$c_i = (c_a - c_*)f0\left(1 - \frac{dq}{dq_{crit}}\right) + c_* \tag{6}$$

$\beta$ is a soil moisture stress factor, the factor 1.6 accounts for $g_s$ being the conductance for water vapour rather than
$CO_2$, $R$ is the universal gas constant (J K$^{-1}$ mol$^{-1}$), $T_l$ is the leaf surface temperature (K), $c_a$ and $c_i$ (both Pa) are the
leaf surface and internal $CO_2$ partial pressures, respectively, $c_*$ (Pa) is the $CO_2$ photorespiration compensation
point, $dq$ is the humidity deficit at the leaf surface (kg kg$^{-1}$), $dq_{crit}$ (kg kg$^{-1}$) and $f_0$ are PFT specific parameters
representing the critical humidity deficit at the leaf surface, and the leaf internal to atmospheric $CO_2$ ratio ($c_i/c_a$)
at the leaf specific humidity deficit (Best *et al.* 2011), values are shown is Table S1.

In this work, we replace equation 6 with the closure described in Medlyn et al. (2011), using the key PFT specific
model parameter $g_l$ (kPa$^{0.5}$), and $dq$ is expressed in kPa, shown in eq 7, hereafter called MED:

$$c_i = c_a\left(\frac{g_1}{g_1 + \sqrt{dq}}\right) \tag{7}$$

PFT specific values of the $g_l$ parameter were derived for European vegetation from the data base of Lin et al.
(2015) and are shown in Table S1. The $g_l$ parameter represents the sensitivity of $g_s$ to the assimilation rate, i.e.
plant water use efficiency, and was derived as in Lin et al. (2015) by fitting the Medlyn *et al.*, (2011) model to
observations of $g_s$, photosynthesis, and VPD, with no $g_0$ term.

The impact of $g_s$ model formulation (JAC versus MED) on simulated water, $O_3$, carbon and energy fluxes is
compared for two contrasting grid points - wet (low soil moisture stress) and dry (high soil moisture stress) in the
European domain. JULES was spun-up for 20 years (1979-1999) at two grid points in central Europe representing
a wet (low soil moisture stress, lat: 48.25; lon:, 5.25) and a dry site (high soil moisture stress, lat: 38.25; lon:, -
7.75). The modelled soil moisture stress factor (*fsmc*) at the wet site ranged from 0.8 to 1.0 over the year 2000
(1.0 indicates no soil moisture stress), and at the dry site *fsmc* steadily declined from 0.8 at the start of the year to
0.25 by the end of the summer. The WFDEI meteorological forcing dataset was used (Weedon, 2013), along with
atmospheric $CO_2$ concentration for the year 1999 (368.33 ppm), and either no $O_3$ (i.e. the $O_3$ damage model was
switched off) for the control simulations, or spatially explicit fields of present day $O_3$ concentration produced
using the UK Chemistry and Aerosol (UKCA) model from the run evaluated by O'Connor et al. (2014) for the
simulations with $O_3$. Following the spin-up period, JULES was run for one year (2000) with corresponding
atmospheric $CO_2$ concentration, and tropospheric $O_3$ concentrations as described above. The control and $O_3$
simulations were performed for both JAC and MED model formulations. Land cover for the spin-up and main run
was fixed at 20% for each PFT. For the simulations including $O_3$ damage, the high plant $O_3$ sensitivity
parameterisation was used. The difference between these simulations was used to assess the impact of $g_s$ model
formulation on the leaf level fluxes of carbon and water. We calculate and report (results section 3.1) the difference
in mean annual water-use that results from the above simulations using the different $g_s$ models. For each day of
the simulation we calculate the percentage difference in water-use between the two simulations, we then calculate
the mean and standard deviation over the year to give the annual mean leaf-level water-use.

Site level evaluation of the two $g_s$ models compared to FLUXNET observations was carried out to evaluate the
seasonal cycles of latent and sensible heat using the two $g_s$ models JAC and MED compared to observations.
Seven Fluxnet towers were selected to represent a range of land cover types as shown in Table S2. JULES was
setup for each site using observed site-level hourly meteorology, and the vegetation cover was prescribed
according to the fractional covers of the different JULES surface types shown in Table S2. Following a spin-up
period, simulations were run at each site for the years shown in Table S2.

**2.4 Model simulations for Europe**

**2.4.1 Forcing datasets**

We used the WATCH meteorological forcing data set (Weedon et al., 2010;Weedon et al., 2011) at 0.5º x 0.5º
spatial and three hour temporal resolution for our JULES simulations. JULES interpolates this down to an hourly
model time step. For this study, the climate was kept constant by recycling over the period 1901 to 1920, to allow
us to focus on the impact $O_3$, $CO_2$ and their interaction.

JULES was run with prescribed annual mean atmospheric $CO_2$ concentrations. Pre-industrial global $CO_2$
concentrations (1900 to 1960) were taken from Etheridge et al. (1996), 1960 to 2002 were from Mauna Loa
(Keeling and Whorf, 2004), as calculated by the Global Carbon Project (Le Quéré et al., 2016), and 2003-2050
were based on the IPCC SRES A1B scenario and were linearly interpolated to gap fill missing years (Fig. 1).

JULES was run including dynamic vegetation with a land cover mask giving the fraction of agriculture in each
0.5° x 0.5° grid cell based on the Hurtt et al. (2011) land cover database for the year 2000. This means that whilst
the model is allowed to evolve its own vegetation cover, within the agricultural mask only $C_3/C_4$ herbaceous PFTs
are allowed to grow, with no competition from other PFTs. Therefore, through the simulation period, regions of
agriculture are maintained as such and not out-competed by forests for example, allowing for a more accurate
representation of the land cover of Europe in the model. No form of land management is simulated (i.e. no crop
harvesting, ploughing, rotation or grazing), growth and leaf area index (LAI) are determined by resource
availability and phenology. Outside of the agricultural mask, dynamic vegetation means that grid cell PFT
coverage and LAI are the result of resource availability, penology and simulated competition. Across the model
domain, simulated mean annual LAI was dominantly within the range of 2 to 5 $m^2/m^2$ (Fig. S3 and S4). Following
a full spin-up period (to ensure equilibrium vegetation, carbon and water states), there was no significant change
in the fractional cover of each PFT over the simulation period (1901 - 2050). By 2050, increases in boreal forest
cover occurred, but this was less than 2% and limited to very small areas, given this small change we show just
the land cover for 2050 in Fig. S1.

Tropospheric $O_3$ concentration was produced by the EMEP MSC-W model at 0.5° x 0.5° (Simpson et al., 2012),
driven with meteorology from the regional climate model RCA3 (Kjellström et al., 2011;Samuelsson et al., 2011),
which provides a downscaling of the ECHAM A1B-r3 (simulation 11 of Kjellström *et al.*, 2011). This setup
(EMEP+RCA3) is also used by Langner et al. (2012a), Simpson et al. (2014a), Tuovinen et al. (2013), Franz et
al. (2017) and Engardt et al. (2017), where further details and model evaluation can be found. Unfortunately, the
3-dimensional RCA3 data needed by the EMEP model was not available prior to 1960, but as in Engardt et al.
(2017) the meteorology of 1900-1959 had to be approximated by assigning random years from 1960 to 1969. This
procedure introduces some uncertainty of course, although Langner et al. (2012b) show that for the period 1990
to 2100 it is emissions change, rather than meteorological change, that drives modelled $O_3$ concentrations. The
emissions scenarios for 1900-2050 merge data from the International Institute of Applied System Analysis
(IIASA) for 2005-2050 (the so-called ECLIPSE 4a scenario), recently revised EMEP data for 1990, and a scaling
back from 1990 to 1900 using data from Lamarque et al. (2013). The trend in emissions of the major $O_3$ precursors
$NO_x$, NMVOC and Isoprene are shown from 1900 to 2050 over Europe in Fig. S5. Isoprene emissions are not
inputs to the EMEP model, but rather calculated at each time-step using temperature, radiation, and land-cover
specific emission factors (Simpson et al., 2012).  Changes in the assumed background concentration of $CH_4$ (from
RCP6.0) (van Vuuren et al., 2011) are also shown in Fig. S5. Engardt et al. (2017) show the trend in emissions of
SO$_2$ and NH$_3$ from 1900 to 2050 over Europe. The EMEP model accounts for changes in BVOC emissions as a
result of predicted ambient temperature changes.

O$_3$ concentrations from EMEP MSC-W were calculated at canopy height for two land-cover categories: forest
and grassland (Fig. S6 and Fig. S7), which are taken as surrogates for high and low vegetation, respectively. These
canopy-height specific concentrations allow for the large gradients in O$_3$ concentration that can occur in the lowest
10s of metres, giving lower O$_3$ for grasslands than seen at e.g. 20 m in a forest canopy (Gerosa et al., 2017;Simpson
et al., 2012;Tuovinen et al., 2009). These canopy level O$_3$ concentrations are used as input to JULES, using the
EMEP O$_3$ concentrations for forest for the forest JULES PFTs (broadleaf/needle leaf tree and shrub), and the
EMEP O$_3$ concentrations for grassland for the grass/herbaceous JULES PFTs (C$_3$ and C$_4$). This study used daily
mean values of tropospheric O$_3$ concentration from EMEP disaggregated down to the hourly JULES model time-
step. The daily mean O$_3$ forcing was disaggregated to follow a mean diurnal profile of O$_3$, this was generated from
hourly O$_3$ output from EMEP MSC-W for the two land cover categories (forest and grassland as described above)
across the same model domain. O$_3$ concentrations follow a diurnal cycle and peak during the day, therefore
accounting for the diurnal variation in O$_3$ concentrations allows for a more realistic estimation of O$_3$ uptake.

Figure 1 shows large increases in tropospheric O$_3$ from pre-industrial to present day (2001), this is in line with
modelling studies (Young et al., 2013) and site observations (Derwent et al., 2008;Logan et al., 2012;Parrish et
al., 2012), and is predominantly a result of increasing anthropogenic emissions (Young et al., 2013). Figures S6
and S7 show this large increase in ground-level O$_3$ concentrations from 1901 to 2001 occurs in all seasons. Present
day O$_3$ concentration show a strong seasonal cycle, with a spring/summer peak in concentrations in the mid-
latitudes of the Northern Hemisphere (Derwent et al., 2008;Parrish et al., 2012;Vingarzan, 2004). Seasonal cycles
have been changing over the past decades however, attributed to changes in NO$_x$ and other emissions, as well as
changes in transport patterns (Parrish et al., 2013). These changes will likely continue in future as emissions and
meteorological factors impact photo-chemical O$_3$ production and transport patterns. Indeed, the O$_3$ concentrations
used in the simulations in this study show increased O$_3$ levels in winter and in some regions in autumn and spring
in 2050 compared to present day, this may be due to reduced titration of O$_3$ by NO as a result of reduced NO$_X$
emissions in the future (Royal Society, 2008). Summer O$_3$ concentrations are lower in 2050 however, compared
to 2001.

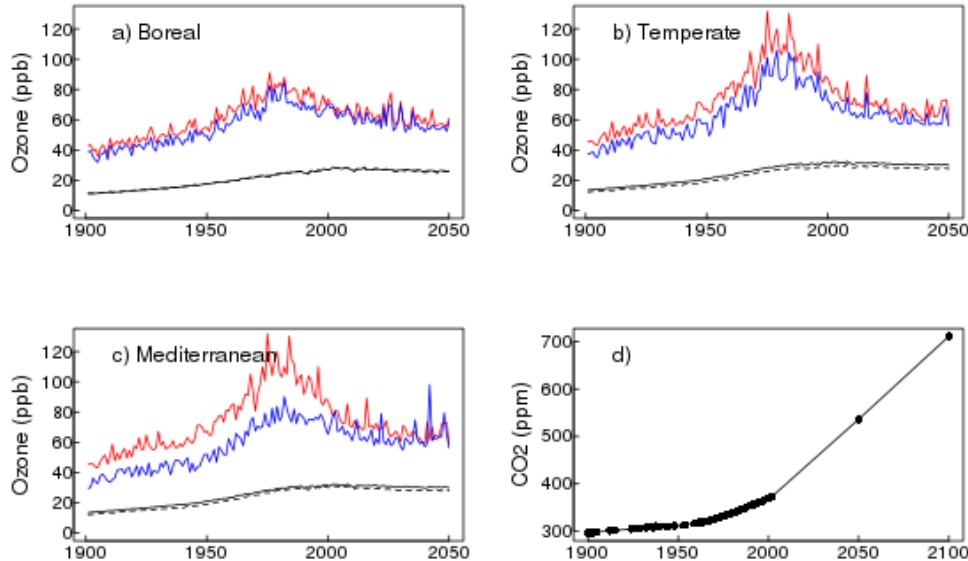


**Figure 1.** Regional time series of canopy height $O_3$ (ppb) forcing from EMEP a) to c), and d) global atmospheric $CO_2$ (ppm) concentration (this does not vary regionally; black dots show data points, the black line shows interpolated points). Each panel for the $O_3$ forcing shows the regional annual average (woody PFTs, black solid line; herbaceous PFTs, black dashed line) and the annual maximum $O_3$ concentration above: woody PFTs (red) and herbaceous PFTs (blue).

### 2.4.2 Spin up and factorial experiments

JULES was spun-up by recycling the climate from the early part of the twentieth century (1901 to 1920) using atmospheric $CO_2$ (296.1 ppm) and $O_3$ concentrations from 1901 (Fig. S3 & Fig. S4). Model spin-up was 2000 years by which point the carbon pools and fluxes were in steady state with zero mean net land – atmosphere $CO_2$ flux. We performed the following transient simulations for the period 1901 to 2050 with continued recycling of the climate as used in the spin-up, for both high and low plant $O_3$ sensitivities:

- **O3**          : Fixed 1901 $CO_2$, Varying $O_3$
- **CO2**          : Varying $CO_2$, Fixed 1901 $O_3$
- **CO2+O3**          : Varying $CO_2$, Varying $O_3$

We use these simulations to investigate the direct effects of changing atmospheric $CO_2$ and $O_3$ concentrations, individually and combined, on plant water-use, GPP and the land C sink through the twentieth century and into the future, specifically over three time periods: historical (1901-2001), future (2001-2050) and over the full time series (1901-2050). For each time period we calculate the difference between the decadal means calculated at the start and end of the analysis period for each variable of interest. Therefore our results report the change in GPP, for example, over the analysis period. For each variable analysed (GPP, NPP, vegetation carbon, soil carbon, total land carbon and *gs*), we use the mean over 10 years to represent each time period, e.g. the mean over 2040 to 2050 is what we call 2050, 1901 to 1910 is what we refer to as 1901. The difference between the simulations gives the effect of $O_3$ and $CO_2$ either separately or in combination over the different time periods. We look at the percentage

change due to either $O_3$ at pre-industrial $CO_2$ concentration (i.e. without the additional effect of atmospheric $CO_2$
on stomatal behaviour - O3 simulation), $CO_2$ (at fixed pre-industrial $O_3$ concentration, CO2 simulation) or the
combined effect of both gases (CO2+O3 simulation), which is calculated as:

$100 * (var[y_1] – var[y_2]) / var[y_2]$                              (8)

Where $var[y_x]$ represents the variable in time period y, e.g. $100 * (varO_3[2050] – varO_3[1901]) / varO_3[1901]$
gives the $O_3$ effect (at fixed $CO_2$) over the full experimental period. The meteorological forcing is prescribed in
these simulations and is therefore the same between the model runs. Other climate factors, such as VPD,
temperature and soil moisture availability are accounted for in our simulations, but our analysis isolates the effects
of $O_3$, $CO_2$ and $O_3 + CO_2$. We also use paired t-test to determine statistically significant differences between the
different (high and low) plant $O_3$ sensitivities.

**2.4.3 Evaluation**
To evaluate our JULES simulations we compare mean GPP from 1991 to 2001 for each of the JULES scenarios
and both high and low plant $O_3$ sensitivities against the observation based globally extrapolated Flux Network
model tree ensemble (MTE) (Jung et al., 2011). We use paired t-test to determine statistically significant
differences in the mean responses.

**3 Results**

**3.1 Impact of $g_s$ model formulation and site level evaluation**

The impact of $g_s$ model on simulated $g_s$ is shown for the site with low soil moisture stress (wet site, Fig. 2). For
the broadleaf tree and $C_3$ herbaceous PFT, the MED model simulates a larger conductance compared to the JAC
model. In other words, with the MED model these two PFTs are parameterised with a less conservative water use
strategy, which, for the grid point shown in Fig. 2, increased the annual mean water use by 35% (±29%) and 45%
(±32%), respectively. In contrast, the needle leaf tree, $C_4$ herbaceous and shrub PFTs are parameterised with a
more conservative water use strategy with the MED model, and the mean annual $g_s$ was decreased by 13% (±12%),
27% (±10%) and 36% (±13%), respectively, compared to the JAC model. This comparison was also done for a
dry site (high soil moisture stress), and similar results were found (Fig. S8). The effect of $g_s$ formulation on
simulated photosynthesis was much smaller because of the lower sensitivity of the limiting rates of photosynthesis
to changes in $c_i$ in the model compared to the effect of the same change in $c_i$ on modelled $g_s$ (Fig. S9 & S10).
Changes in $g_s$ impact the partitioning of simulated energy fluxes. In general, increased $g_s$ results in increased latent
heat and thus decreased sensible heat flux, and vice versa where $g_s$ is decreased (Fig. S9 & S10). Also shown is
the effect of the MED model on $O_3$ flux into the leaf (Fig. S11 and Fig. S8 bottom panel). For the broadleaf tree
and $C_3$ herbaceous PFT, the MED model simulates a larger conductance and therefore a greater flux of $O_3$ through
stomata compared to JAC, and this is indicative of the potential for greater reductions in photosynthesis (Fig. S9
& S10 top row). The reverse is seen for the needle leaf tree, $C_4$ herbaceous and shrub PFTs.

Site level evaluation of the seasonal cycles of latent and sensible heat with both JAC and MED models compared
to FLUXNET observations showed in general, the MED model improved the seasonal cycle of both fluxes (lower
RMSE), but the magnitude of this varied from site to site (Fig. S12). At the deciduous broadleaf site, US-UMB,
MED resulted in improvements of the simulated seasonal cycle particularly in the summer months for both fluxes
(RMSE decreased from 42.7/31.5 to 38.5/28.0 W/m$^2$ for latent/sensible heat respectively). At the second
deciduous broadleaf site IT-CA1 however, there was almost no difference between the two $g_s$ models. Both
evergreen needle leaf forest sites (FI-Hyy and DE-Tha) saw improvements in the simulated seasonal cycles of
latent and sensible heat with the MED model, primarily as a result of lower latent heat flux in the spring and
summer months, and higher sensible heat flux over the same period. At FI-Hyy, RMSE decreased from 10.1/7.4
to 6.7/6.7 W/m$^2$ for latent/sensible heat respectively, and at DE-Tha, RMSE decreased from 16.0/11.9 to 10.5/10.6
W/m$^2$ for latent/sensible heat respectively. With the MED model the monthly mean latent heat flux was improved
at the $C_3$ grass site (CH-Cha) as a result of increased flux in the summer months (RMSE decreased from 15.7 to
13.8 W/m$^2$), however there was no improvement in the sensible heat flux and RMSE with MED was increased
(from 3.9 to 4.9 W/m$^2$). At the $C_4$ grass site (US-SRG), small improvements were made in the seasonal cycle of
both latent and sensible heat with the MED model. At the deciduous savannah site (CG-Tch) which included a
high proportion of shrub PFT in the land cover type used in the site simulation, large improvements in the seasonal
cycle of both fluxes were simulated with the MED model, as a result of a decrease in the latent heat flux and an
increase in the sensible heat flux (RMSE decreased from 39.5/31.6 to 30.4/24.4 W/m$^2$ for latent/sensible heat
respectively).

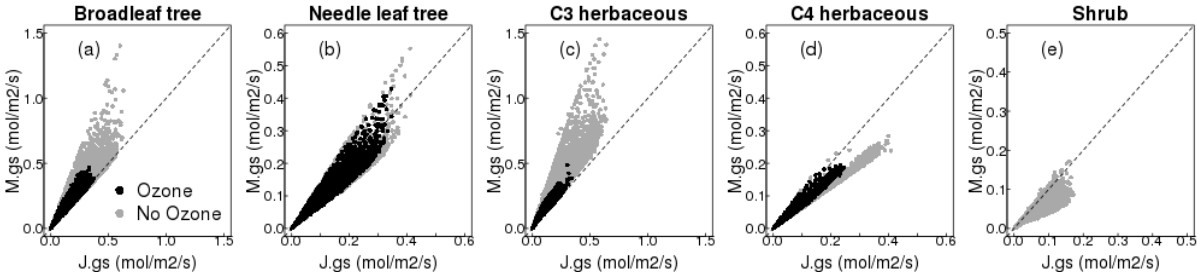


**Figure 2.** Comparison of simulated $g_s$ with MED (y axis) versus JAC (x axis) for all five JULES PFTs at one grid
point (lat: 48.25; lon:, 5.25) shown are hourly values for the year 2000 (see SI section S3 for further details).

**3.2 Evaluation of the JULES $O_3$ model**
For all JULES scenarios similar spatial patterns of GPP are simulated compared to MTE (Fig. 3 and Fig. S13).
MTE estimates a mean GPP for present day in Europe of 938 gC m$^2$ yr$^{-1}$ (Fig. 3). JULES tends to under-predict
GPP relative to the MTE product, estimates of GPP from JULES with both transient $CO_2$ and $O_3$ (CO2+O3
simulation) gives a mean across Europe of 813 gC m$^2$ yr$^{-1}$ (high plant $O_3$ sensitivity) to 881 gC m$^2$ yr$^{-1}$ (low plant
$O_3$ sensitivity), both of which are significantly different to the MTE product ($t=27$, $d.f.=5750$, $p<2.2e^{-16}$ (high);
$t=4.3$, $d.f.=5750$, $p<1.5e^{-05}$ (low); Fig. 3). Forcing with $CO_2$ alone (CO2 simulation) gives a mean GPP across
Europe of 900 to 923 gC m$^2$ yr$^{-1}$ (high and low plant $O_3$ sensitivity respectively), and $O_3$ alone (O3 simulation -
without the protective effect of $CO_2$) reduces estimated GPP to 732 to 799 gC m$^2$ yr$^{-1}$ (Fig. S13). At latitudes
>45°N JULES has a tendency to under-predict MTE-GPP, and at latitudes <45 °N JULES tends to over-predict
MTE-GPP (Fig. S14). These regional differences are highlighted in Fig. S15, where in the Mediterranean region,
JULES tends to over-predict compared to MTE-GPP, so simulations with $O_3$ reduce the simulated GPP bringing
it closer to MTE. In the temperate region however, JULES tends to under-estimate MTE-GPP, so the addition of
$O_3$ reduces simulated GPP further (Fig. S15). In the boreal region, JULES under-predicts GPP, but to a lesser
extent than in the temperate region, and the addition of $O_3$ has less impact on reducing the GPP further (Fig. S15).

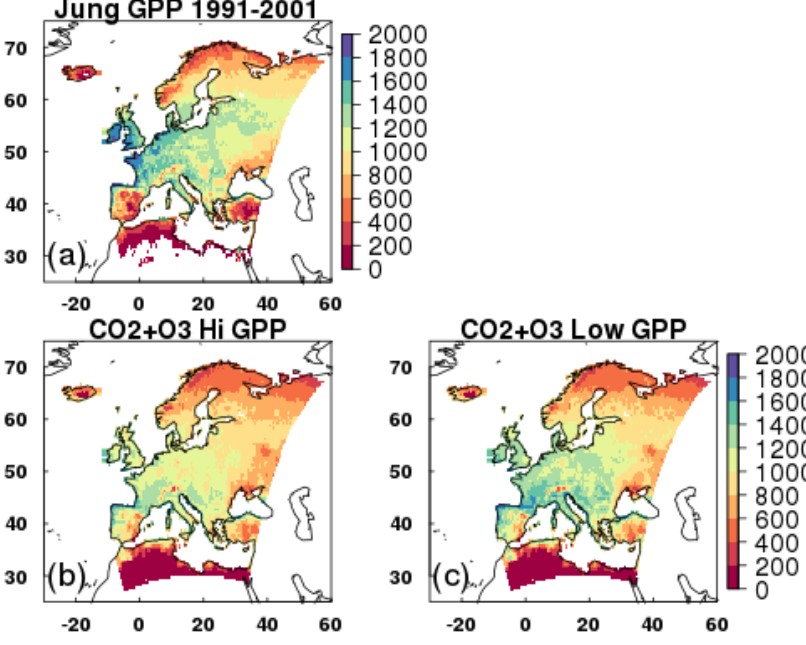

**Figure 3.** Mean GPP (g C m$^2$ yr$^{-1}$) from 1991 to 2001 for a) the observationally based globally extrapolated Flux
Network model tree ensemble (MTE) (Jung *et al.*, 2011); b, c) model simulations with transient $CO_2$ and transient
$O_3$ (CO2+O3), high and low plant $O_3$ sensitivity respectively.


**3.3 European simulations - Historical Period: 1901-2001**

Over the historical period (1901-2001), $O_3$ (O3 simulation) reduced GPP under both the low and high plant $O_3$
sensitivity parameterizations by -3% to -9% respectively (Table 1), and this difference in simulated GPP was
significant ($t=102.2$, $d.f.=6270$, $p<2.2e^{-16}$). Figure 4 highlights regional variations, however, where simulated
reductions in GPP are up to 20% across large areas of Europe, and up to 30% in some Mediterranean regions
under the high plant $O_3$ sensitivity. Some Boreal and Mediterranean regions show small increases in GPP over

this period, associated with $O_3$ induced stomatal closure enhancing water availability in these drier regions (Fig. 5). This allows for greater stomatal conductance later in the year when soil moisture may otherwise have been limiting to growth (up to 10%, Fig. 5), and therefore higher GPP, but these regions comprise only a small area of the entire domain. Indeed, over much of the Europe, $O_3$-induced stomatal closure led to reduced $g_s$ (up to 20%) across large areas of temperate Europe and the Mediterranean, and even greater reductions in some smaller regions of southern Mediterranean (Fig. 6), and these are not associated with notable increases in soil moisture availability (Fig. 5), resulting in depressed GPP over much of Europe as described above. Under the low plant $O_3$ sensitivity, similar spatial patterns occur, but the magnitude of GPP change (up to -10% across much of Europe) and $g_s$ change (-5% to -10%) are lower compared to the high sensitivity. Over the twentieth century the land carbon sink is suppressed (-2% to -6%, Table 1). Large regional variation is shown in Figure 4, with temperate and Mediterranean Europe seeing a large reduction in land carbon storage, particularly under the high plant $O_3$ sensitivity (up to -15%).

Combined, the physiological response to changing $CO_2$ and $O_3$ concentrations (CO2+O3 simulation) results in a net loss of land carbon over the twentieth century under the high plant $O_3$ sensitivity (-2%, Table 1), dominated by loss of soil carbon (Table S3). This reflects the large increases in tropospheric $O_3$ concentration observed over this period (Fig. 1). Under the low plant $O_3$ sensitivity, the land carbon sink has started to recover by 2001 (+1.5%) owing to the recovery of the soil carbon pool beyond 1901 values over this period (Table S3).

To gain perspective on the magnitude of the $O_3$ induced flux of carbon from the land to the atmosphere we relate changes in total land carbon to carbon emissions from fossil fuel combustion and cement production for the EU-28-plus countries from the data of Boden et al. (2013). We recognise that our simulation domain is slightly larger than the EU28-plus as it includes a small area of western Russia so direct comparisons cannot be made, but this still provides a useful measure of the size of the carbon flux. For the period 1970 to 1979 the simulated loss of carbon from the European terrestrial biosphere due to $O_3$ effects on vegetation physiology was on average 1.32 Pg C (high vegetation sensitivity) and 0.71 Pg C (low vegetation sensitivity) (Table 2). This $O_3$ induced reduced C uptake of the land surface is equivalent to around 8% to 16% of the emissions of carbon from fossil fuel combustion and cement production over the same period for the EU28-plus countries (Table 2). Currently the emissions data availability goes up to 2011, over the last observable decade (2002 to 2011) the simulated reduction in land carbon due to $O_3$ has declined, but is still equivalent to 2% to 4% of the emissions of carbon from fossil fuels and cement production for the EU28-plus countries (Table 2). By comparison with one of the largest anthropogenic emissions of carbon for Europe, we show here the potential effect of $O_3$ on reducing the size of the European land carbon sink is notable.

**3.4 European simulations - Future Period: 2001-2050**

Over the 2001 to 2050 period, region-wide GPP with $O_3$ only changing (O3 simulation) increased marginally (+0.1% to +0.2%, high and low plant $O_3$ sensitivity, Table 1, with a significant difference between the two plant $O_3$ sensitivities ($t$=57, $d.f.$=6270 $p$<2.2e$^{-16}$)), although with large spatial variability as discussed below (Fig. 4g & h). Figures S6 and S7 show that despite decreased tropospheric $O_3$ concentrations by 2050 in summer compared

to 2001 levels, all regions are exposed to an increase in $O_3$ over the wintertime, and some regions of Europe,
particularly temperate/Mediterranean experience increases in $O_3$ concentration in spring and autumn. Therefore,
although in the O3 simulation, overall simulated GPP for Europe shows a small increase, large spatial variability
is shown in Fig's 4g &h because of the variability in $O_3$ concentration with region and season. Increased GPP
(dominantly 10%, but up to 20% in some areas) on 2001 levels is simulated across areas of Europe, however,
decreases of up to 21% are simulated in some areas of the Mediterranean, up to 15% in some areas of the boreal
region and up to 27% in the temperate zone (Fig. 4g & h).

When $O_3$ and $CO_2$ effects are combined (CO2+O3 simulation), simulated GPP increases (+15% to +18%,
high/low plant $O_3$ sensitivities respectively, Table 1). This increase is greater than the enhancement simulated
when $CO_2$ affects plant growth independently (CO2 simulation), because additional $O_3$ induced stomatal closure
increases soil water availability in some regions, which enhances growth more in the CO2+O3 simulation,
compared to the CO2 simulation. Nevertheless, although the percentage gain is larger, the absolute value of GPP
by 2050 remains lower in CO2+O3 compared to GPP in the CO2 simulations, highlighting the negative impact of
$O_3$ at the land surface (Table S4).

Despite small increases in GPP in the O3 simulation, the land carbon sink continues to decline from 2001 levels
(-0.7% to -1.6%, low and high plant $O_3$ sensitivity respectively, Table 1). This is because the soil and vegetation
carbon pools continue to lose carbon as they adjust slowly to small changes in input (GPP), i.e. the soil carbon
pool is not in equilibrium in 2001, and is declining in response to reduced litter input as a result of $20^{th}$ C $O_3$
impacts on GPP. Nevertheless, the negative effect of $O_3$ on the future land sink is markedly reduced relative to
the historical period. Figure 4e & f however highlights regional differences. Boreal regions and parts of central
Europe see minimal $O_3$ damage, whereas some areas of southern and northern Europe see further losses of up to
8% on 2001 levels. The CO2+O3 simulation are dominated by the physiological effects of changing $CO_2$, with
land carbon sink increases of up to 7% (Table 1).

**3.5 European simulations – Full experimental period: 1901-2050**

From 1901 to 2050, the O3 simulation reduces GPP (-4% to -9%, with a significant difference between the low
and high plant $O_3$ sensitivity ($t$=95, $d.f.$=6270 $p$<2.2e$^{-16}$)) and land carbon storage (-3% to -7%, Table 1).
Regionally, $O_3$ damage is lowest in the boreal zone, GPP decreases are largely between 5% to 8% / 2% to 4% for
the high/low plant $O_3$ sensitivity respectively, with large areas minimally affected by $O_3$ damage (Figure 7),
consistent with lower $g_s$ of needle leaf trees that dominate this region, and so lower $O_3$ uptake (Fig. S16 & S17).
In the temperate region, $O_3$ damage is extensive with reductions in GPP dominantly from 10% to 15% for the low
and high plant $O_3$ sensitivity respectively. Across significant areas of this region reductions in GPP are up to 20%
under high plant $O_3$ sensitivity (Figure 7). In the Mediterranean region, $O_3$ damage reduces GPP by 5% to 15% /
3% to 6% for the high/low plant $O_3$ sensitivity respectively, with some areas seeing greater losses of up to 20%
under the high plant $O_3$ sensitivity, but this is less extensive than that seen in the temperate zone (Figure 7). In
these drier regions, $O_3$ induced stomatal closure can increase available soil moisture (Fig. S16 & S17).

The CO2+O3 simulation shows that $CO_2$ induced stomatal closure can help alleviate $O_3$ damage by reducing the
uptake of $O_3$ (Table S6). In these simulations, $CO_2$-induced stomatal closure was found to offset $O_3$-suppression
of GPP, such that GPP by 2050 is 3% to 7% lower due to $O_3$ exposure (CO2+O3), rather than 4% to 9% lower in
the absence of increasing $CO_2$ (O3 simulation, Table S6). Figure 6 shows this spatially, $O_3$ damage is reduced
when the effect of atmospheric $CO_2$ on stomatal closure is accounted for, however despite this, the land carbon
sink and GPP remain significantly reduced due to $O_3$ exposure.

From 1901 to 2050, the CO2+O3 simulation results in an increase in European land carbon uptake (+5% to +9%),
and an increase in GPP (+20% to +23%) by 2050 for the high and low plant $O_3$ sensitivity, respectively (Table 1).
Nevertheless, despite this increase there remains a large negative impact of $O_3$ on the European land carbon sink
(Fig. S18). By 2050 the simulated enhancement of land carbon and GPP in response to elevated $CO_2$ alone (CO2
simulation) is reduced by 3% to 6% (land carbon) and 4% to 9% (GPP) for the low and high plant $O_3$ sensitivity
respectively, when $O_3$ is also accounted for (CO2+O3 simulation, Table 1). This is a large reduction in the ability
of the European terrestrial biosphere to sequester carbon.


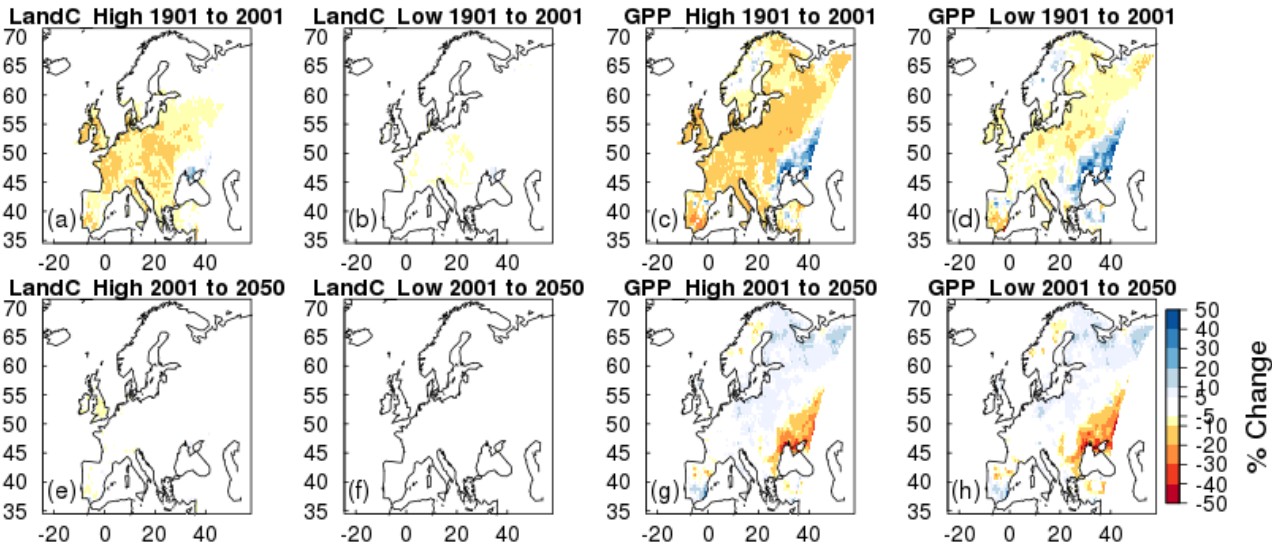

**Figure 4.** Simulated percentage change in total carbon stocks (Land C) and gross primary productivity (GPP) due
to $O_3$ effects at fixed pre-industrial atmospheric $CO_2$ concentration (O3 simulation). Changes are shown for the
periods 1901 to 2001, and 2001 to 2050 for the high and low plant $O_3$ sensitivity.

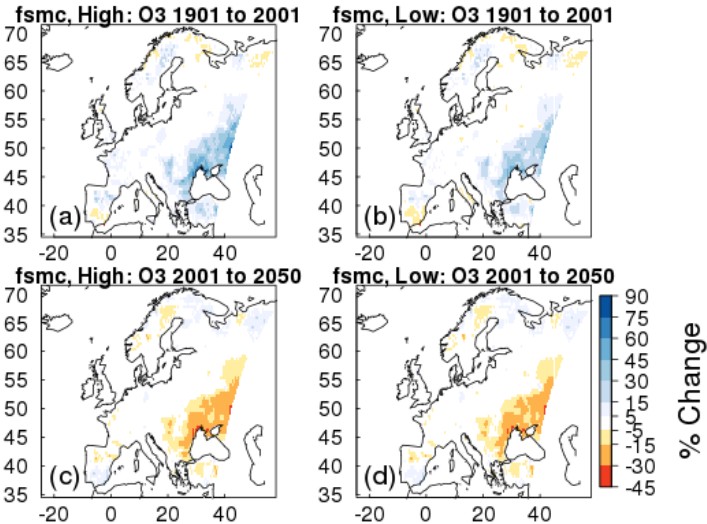


**Figure 5.** Simulated percentage change in plant available soil moisture (*fsmc*) due to $O_3$ effects at fixed pre-industrial atmospheric $CO_2$ concentration (O3 simulation). Changes are shown for the periods 1901 to 2001, and 2001 to 2050 for the high and low plant $O_3$ sensitivity.


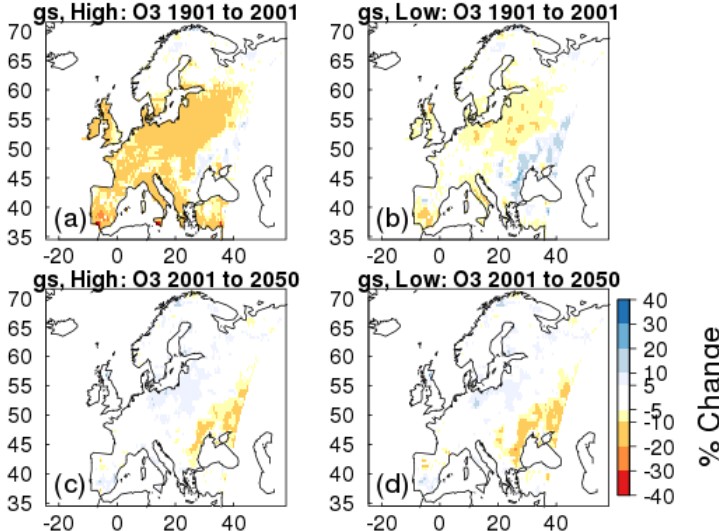


**Figure 6.** Simulated percentage change in stomatal conductance ($g_s$) due to $O_3$ effects at fixed pre-industrial atmospheric $CO_2$ concentration (O3 simulation). Changes are shown for the periods 1901 to 2001, and 2001 to 2050 for the high and low plant $O_3$ sensitivity.


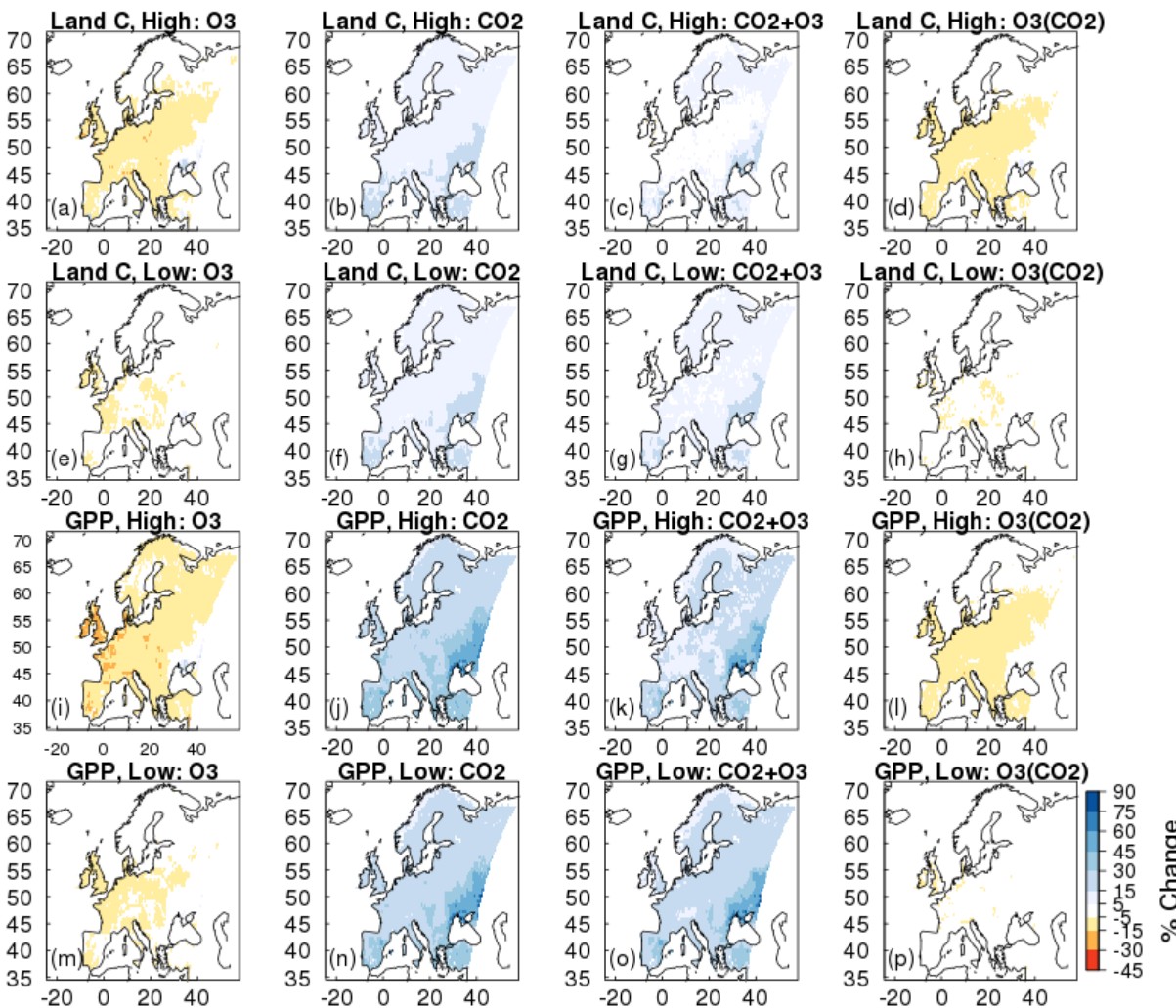


**Figure 7.** Simulated percentage change in total carbon stocks (Land C) and gross primary productivity (GPP) due to i) (a, e, i, m) $O_3$ effects at fixed pre-industrial atmospheric $CO_2$ concentration (O3 simulation), ii) (b, f, j, n) $CO_2$ fertilisation at fixed pre-industrial $O_3$ concentration (CO2 simulation), iii) (c, g, k, o) the interaction between $O_3$ and $CO_2$ effects (CO2+O3 simulation) iv) (d, h, l, p) $O_3$ effects with changing atmospheric $CO_2$ concentration (i.e. $O_3$ damage accounting for the effect of $CO_2$ induced stomatal closure; CO2+O3 – CO2). Changes are depicted for the periods 1901 to 2050 for high and lower ozone plant sensitivity.








| | High Plant $O_3$ Sensitivity | | | | | |
|---|---|---|---|---|---|---|
| | **1901 - 2001** | | **2001 - 2050** | | **1901 - 2050** | |
| | GPP | Land C | GPP | Land C | GPP | Land C |
| | (Pg C yr$^{-1}$) | (Pg C) | (Pg C yr$^{-1}$) | (Pg C) | (Pg C yr$^{-1}$) | (Pg C) |
| Value in 1901: | 9.05 | 167 | - | - | 9.05 | 167 |
| Absolute Change: | | | | | | |
| **O3** | -0.81 | -9.21 | 0.01 | -2.44 | -0.80 | -11.65 |
| **CO2** | 1.16 | 4.24 | 1.42 | 12.98 | 2.58 | 17.22 |
| **CO2 + O3** | 0.13 | -3.28 | 1.66 | 11.11 | 1.79 | 7.83 |
| % Change: | | | | | | |
| **O3** | -8.95 | -5.51 | 0.12 | -1.55 | -8.84 | -6.98 |
| **CO2** | 12.82 | 2.54 | 13.91 | 7.58 | 28.51 | 10.31 |
| **CO2 + O3** | 1.44 | -1.96 | 18.08 | 6.79 | 19.78 | 4.69 |
| | Low Plant $O_3$ Sensitivity | | | | | |
| | **1901 - 2001** | | **2001 - 2050** | | **1901 - 2050** | |
| | GPP | Land C | GPP | Land C | GPP | Land C |
| | (Pg C yr$^{-1}$) | (Pg C) | (Pg C yr$^{-1}$) | (Pg C) | (Pg C yr$^{-1}$) | (Pg C) |
| Value in 1901: | 9.34 | 167.5 | - | - | 9.34 | 167.5 |
| Absolute Change: | | | | | | |
| **O3** | -0.30 | -3.59 | 0.02 | -1.07 | -0.40 | -4.66 |
| **CO2** | 1.15 | 6.43 | 1.35 | 13.14 | 2.50 | 19.57 |
| **CO2 + O3** | 0.65 | 2.50 | 1.50 | 12.35 | 2.15 | 14.85 |
| % Change: | | | | | | |
| **O2** | -3.21 | -2.14 | 0.22 | -0.65 | -4.28 | -2.78 |
| **CO2** | 12.31 | 3.84 | 12.87 | 7.55 | 26.77 | 11.68 |
| **CO2 + O3** | 6.96 | 1.49 | 15.02 | 7.26 | 23.02 | 8.87 |


**Table 1**. Simulated changes in the European land carbon cycle due to changing $O_3$ and $CO_2$ concentrations
(independently and together). Shown are changes in total carbon stocks (Land C) and gross primary productivity
(GPP), over three different periods (historical: 1901 to 2001, future: 2001 to 2050, and full time series: 1901 to
2050). Absolute (top) and relative (bottom) differences are shown. For 2001 to 2050, please refer to Table S4 for
the initial value for each run. See the SI for details of the estimation of the $O_3$ and $CO_2$ effects and their interaction.










| | Mean (Pg C) | | | | |
|---|---|---|---|---|---|
| | 1970-1979 | 1980-1989 | 1990-1999 | 2000-2009 | 2002-2011 |
| **Modelled O₃ effect on land C sink :** | | | | | |
| Higher sensitivity | -1.32 | -1.01 | -0.97 | -0.53 | -0.50 |
| Low sensitivity | -0.71 | -0.58 | -0.50 | -0.29 | -0.26 |
| **Sum of C emissions from fossil fuel combustion and cement production (Pg C)** | 8.39 | 8.63 | 12.26 | 12.83 | 12.75 |
| **C lost from O₃ effect as a % of fossil fuel and cement emissions (%):** | | | | | |
| Higher sensitivity | -15.73 | -11.70 | -7.91 | -4.13 | -3.92 |
| Low sensitivity | -8.46 | -6.72 | -4.08 | -2.26 | -2.04 |


**Table 2.** Simulated change in total land carbon due to $O_3$ damage with changing atmospheric $CO_2$ concentration
for the two vegetation sensitivities. The sum of carbon emissions for each decade from fossil fuel combustion and
cement production for the EU-28 countries plus Albania, Bosnia and Herzegovina, Iceland, Belarus, Serbia,
Moldova, Norway, Turkey, Ukraine, Switzerland and Macedonia (EU28-plus) are shown, the data is from Boden
*et al.*, 2013. The simulated change in land carbon as a result of $O_3$ damage is depicted as a percentage of the EU28-
plus emissions to demonstrate the magnitude of the additional source of carbon to the atmosphere from plant $O_3$
damage.

**4 Discussion**

**4.1 Evaluation of $g_s$ models and JULES $O_3$ model**

Comparison of the new $g_s$ model implemented in this study (MED) with the $g_s$ model currently used as standard
in JULES (JAC) revealed large differences in $g_s$ for each PFT, principally as a result of the data-based
parameterisation of the new model. Water use increased for the broadleaf tree and $C_3$ herbaceous PFTs using the
MED model compared to JAC, but decreased for the needle leaf tree, $C_4$ herbaceous and shrub PFTs which
displayed a more conservative water use strategy compared to JAC. These changes are in line with the work of
De Kauwe et al. (2015) who found a reduction in annual transpiration for evergreen needle leaf, tundra and $C_4$
grass regions when implementing the Medlyn $g_s$ model into the Australian land surface scheme CABLE. Site-
level evaluation of the models against Fluxnet observations showed that in general the MED model improved
simulated seasonal cycles of latent and sensible heat. The magnitude of the improvement varied with site, large
improvements were seen at the deciduous savanna site, and at the NT sites and BT site (US_UMB) in the spring
and summer. However, much smaller improvements were seen at the grass sites. Changes in $g_s$ in this study
resulted in differences in latent and sensible heat fluxes. Changes in the partitioning of energy fluxes at the land
surface could have consequences for the intensity of heatwaves (Cruz et al., 2010;Kala et al., 2016), runoff (Betts
et al., 2007;Gedney et al., 2006) and rainfall patterns (de Arellano et al., 2012), although fully coupled simulations
would be necessary to detect these effects. The differences in simulated $g_s$ led to differences in uptake of $O_3$
between the two models because the rate of $g_s$ is the predominant determinant of the flux of $O_3$ through stomata.
Higher $O_3$ uptake is indicative of greater damage. Therefore, given that $C_3$ herbaceous vegetation is the dominant
land cover class across the European domain used in this study, this suggests a greater $O_3$ impact for Europe would
be simulated with MED model compared to JAC in our simulations where chemistry is uncoupled from the land
surface.

We evaluated the JULES $O_3$ model by comparing modelled GPP against the Jung et al (2011) MTE product.
Similar spatial patterns of GPP were simulated by JULES compared to MTE. Zonal means also showed similar
patterns of GPP, although JULES under predicted GPP compared to MTE at latitudes >45ºN (temperate and boreal
regions; all simulations) and over predicted GPP at latitudes <45ºN (Mediterranean region; all simulations). The
simulations with transient $O_3$ (i.e. O3 and CO2+O3) showed large differences in GPP between the high and low
plant $O_3$ sensitivity simulations, this is to be expected given that the high plant $O_3$ sensitivity simulations were
parameterised to be 'damaged' more by $O_3$, i.e. greater reduction of photosynthesis/$g_s$ with $O_3$ exposure compared
to the low plant $O_3$ sensitivity simulations. This difference was largest in the temperate zone, largely because of
$C_3$ grass cover being the dominant land cover here and the difference in the sensitivity to $O_3$ between the high and
low calibrations is significantly larger for $C_3$ grasses compared to the needle leaf trees that dominate in the boreal
region. Additionally, a longer growing season in the temperate region may allow for greater uptake of $O_3$ into
vegetation. $C_3$ grass is also the dominant land cover in the Mediterranean region with a different calibration used
for Mediterranean grasses for the low plant $O_3$ sensitivity which is less sensitive to $O_3$ than the temperate $C_3$
grasses, but high soil moisture stress is common throughout the growing season in the Mediterranean limiting the
uptake of $O_3$ through stomata, which likely diminishes the difference between the high and low calibrations.

**4.2 Lower than expected $O_3$ damage?**

Our estimates suggest present day $O_3$ reduced GPP by 3% to 9% on average across Europe and NPP by 5% to
11% (Table S3). Anav et al. (2011) simulated a 22% reduction of GPP across Europe for 2002 using the
ORCHIDEE model. Present day $O_3$ exposure reduced GPP by 10% to 25% in Europe, and 10.8% globally in the
study by Lombardozzi et al. (2015) using the Community land model (CLM). $O_3$ reduced NPP by 11.2% in Europe
from 1989 to 1995 using the Terrestrial Ecosystem Model (TEM) (Felzer et al., 2005). Globally, concentrations
of $O_3$ predicted for 2100 reduced GPP by 14% to 23% using a former parameterisation of $O_3$ sensitivity in JULES
(Sitch et al., 2007). The recent study by Franz et al. (2017) showed mean GPP declined by 4.7% over the period
2001 to 2010 using the OCN model over the same European domain and using the same $O_3$ forcing produced by
EMEP MSC-W as used in this study. Our estimates of changes in current day GPP and NPP are at the lower end
of previously modelled estimates. Simulated $O_3$ impacts will depend in a large part on the scenario of $O_3$
concentrations used as forcing, meteorological forcing and how sensitive vegetation is parameterised to be to $O_3$
damage, in addition to the different process representation of $O_3$ damage in each model. It is therefore difficult to
hypothesise as to exactly why modelled estimates differ, but suggests that an ensemble approach to modelling $O_3$
impacts on the terrestrial biosphere would be beneficial to understand some of these differences and provide
estimates of $O_3$ damage with uncertainties.

**4.3 Impacts of O₃ at the land surface**

In this study, $O_3$ has a detrimental effect on the size of the land carbon sink for Europe. This is primarily through a decrease in the size of the soil carbon pool as a result of reduced litter input to the soil, consistent with reduced GPP/NPP. Field studies show that in some regions of Europe, soil carbon stocks are decreasing (Bellamy et al., 2005;Capriel, 2013;Heikkinen et al., 2013;Sleutel et al., 2003). The study of Bellamy et al. (2005), for example, showed that carbon was lost from soils across England and Wales between 1978 to 2003 at a mean rate of 0.6% per year with little effect of land use on the rate of carbon loss, suggesting a possible link to climate change. It is understood that climate change is likely to affect soil carbon turnover. Increased temperatures increase microbial decomposition activity in the soil, and therefore increase carbon losses through higher rates of respiration (Cox et al., 2000;Friedlingstein et al., 2006;Jones et al., 2003). However, some studies have found that $O_3$ can decrease soil carbon content. Talhelm et al. (2014), for example, found $O_3$ reduced carbon content in near surface mineral soil of forest soils exposed to 11 years of $O_3$ fumigation. Hofmockel et al. (2011) found elevated $O_3$ reduced the carbon content in more stable soil organic matter pools, and Loya et al. (2003) showed that the fraction of soil carbon formed in forest soils over a 4 year experimental period when fumigated with both $CO_2$ and $O_3$ was reduced by 51% compared to the soil fumigated with $CO_2$ alone. It is agreed that amongst other factors that change with $O_3$ exposure such as litter quality and composition, reduced litter quantity also has significant detrimental consequences for soil carbon stocks (Andersen, 2003;Lindroth, 2010;Loya et al., 2003). Results from this study therefore suggest that increasing tropospheric $O_3$ may be a contributing factor to the declining soil carbon stocks observed across Europe as a result of reduced litter input to the soil carbon pool consistent with reduced NPP.

We acknowledge, however, that our model simulations do not include coupling of Nitrogen and Carbon cycles, or land management practices. We include a representation of agricultural regions through the model calibration against the wheat $O_3$ sensitivity function (CLRTAP, 2017), and in our simulations the high plant $O_3$ sensitivity scenario uses this calibration against wheat for all $C_3$/$C_4$ land cover which dominates our model domain. Wheat is known to be one of the most $O_3$ sensitive crop species however, so it is possible that our simulations over-estimate the $O_3$ impact at the land surface. However, the low plant $O_3$ sensitivity calibration against natural grasslands provides a counter estimate of the impact of $O_3$ at the land surface, therefore it is important to consider the range our results provide (i.e. both the high and low plant $O_3$ sensitivity) as an indicator of the impact of $O_3$ on the land surface. As with all uncoupled modelling studies, a change in $g_s$ and flux will impact the $O_3$ concentration itself. Therefore adopting the Medlyn formulation with a higher $g_s$ and subsequently higher $O_3$ flux for broadleaf and $C_3$ PFTs (Fig 2) would lead to reduced $O_3$ concentration, which in turn would act to dampen the effect of higher $g_s$ on $O_3$ flux, although the higher uptake of $O_3$ by vegetation may lead to more damage and increase $O_3$ concentrations, in an uncoupled chemistry-land modelling system such as this it is not possible to predict which process would dominate. Additionally, this version of JULES does not have a crop module; it has no land management practices such as harvesting, ploughing or crop rotation – processes which may have counteracting effects on the land carbon sink. Further, without a coupled Carbon and Nitrogen cycle, it is likely that the $CO_2$ fertilisation response of GPP and the land carbon sink is over estimated in some regions of our simulations since nitrogen availability limits terrestrial carbon sequestration of natural ecosystems in the temperate and boreal zone

(Zaehle, 2013). This would have consequences for our modelled $O_3$ impact, particularly into the future where the
large $CO_2$ fertilisation effect was responsible for partly offsetting the negative impact of $O_3$. Although in our
simulations a high fraction of land cover is agricultural which we assume would be optimally fertilised. Our
simulations also use a fixed climate, so we do not include the effect of climate change on shifting plant phenology.
Therefore, our results may underestimate plant $O_3$ damage, since if the growing season started earlier or finished
later, plants in some regions would be exposed to higher $O_3$ concentrations. Nevertheless, we emphasise that this
study provides a sensitivity assessment of the impact of plant $O_3$ damage on GPP and the land carbon sink.

Another caveat we fully acknowledge is that at the leaf-level JULES is parameterised to reduce $g_s$ with $O_3$
exposure. Whilst this response is commonly observed (Wittig et al., 2007;Ainsworth et al., 2012), there is evidence
to suggest that $O_3$ impairs stomata in some species, making them non-responsive to environmental stimuli (Hayes
et al., 2012;Hoshika et al., 2012a;Mills et al., 2009;Paoletti and Grulke, 2010). In drought conditions the
mechanism is thought to involve $O_3$ stimulated ethylene production which interferes with the stomatal response
to ABA signalling (Wilkinson and Davies, 2009;Wilkinson and Davies, 2010). Such stomatal sluggishness can
result in higher $O_3$ uptake and injury, increased water-loss, and therefore greater vulnerability to environmental
stresses (Mills et al., 2016). McLaughlin (2007a;2007b) and Sun et al. (2012) provide evidence of increased
transpiration and reduced streamflow in forests at the regional scale in response to ambient levels of $O_3$, and
suggest this could increase the frequency and severity of droughts. Hoshika et al. (2012b) however found that
despite sluggish stomatal control in $O_3$ exposed trees, whole tree water use was lower in these trees because of
lower gas exchange and premature leaf shedding of injured leaves. To our knowledge, the study of Hoshika et al.
(2015) is the first to include an explicit representation of sluggish stomatal control in a land-atmosphere model,
they show that sluggish stomatal behaviour has implications for carbon and water cycling in ecosystems.
However, it is by no means a ubiquitous response, and it is not fully understood which species respond this way
and under what conditions (Mills et al., 2016;Wittig et al., 2007). Nevertheless, this remains an important area of
future work.

In this work we implement the stomatal closure proposed in Medlyn et al., (2011), this uses the parameter $g_1$.
Hoshika et al. (2013) show a significant difference in the $g_1$ parameter (higher in elevated $O_3$ compared to ambient)
in Siebold's beech in June of their experiment. However, this is only at the start of the growing season, further
measurements show no difference in this parameter between $O_3$ treatments. Quantifying an $O_3$ effect directly on
$g_1$ would require a detailed meta-analysis of empirical data on photosynthesis and $g_s$ for different PFTs, which is
currently lacking in the literature. With such information lacking, here we take an empirical approach to modelling
plant $O_3$ damage, essentially by applying a reduction factor to the simulated plant photosynthesis based on
observations of whole plant losses of biomass with accumulated $O_3$ exposure, for which there is a lot more
available data (e.g. CLRTAP, 2017).

The calculation of $O_3$ deposition in the EMEP model uses the stomatal conductance formulation presented in
Emberson et al. (2000;2001), which depends on temperature, light, humidity and soil moisture (commonly
referred to as DO$_3$SE). Because we link two different model systems, the $g_s$ values in the EMEP model differ from
those obtained using the Medlyn formulation. We acknowledge this inconsistency as a caveat of our study,
however comparison of *gmax* (maximum $g_s$) values from both models (EMEP (*gmax* is an input parameter
determining the maximum $g_s$) and JULES (*gmax* is not used as an input parameter in JULES, instead we calculated
*gmax* for each PFT taking the mean across the model domain for the year 2001) suggests the differences are small
for deciduous forest (EMEP 150-200, JULES ~180, all units in mmol $O_3$/$m^2$ (PLA)/s), and $C_3$/$C_4$ crops (EMEP
270-300, JULES ~260-390), but are larger for coniferous forest (EMEP 140-200, JULES ~60-70) and shrubs
(EMEP 60-200, JULES 360-390). It should be noted that the role of EMEP in this study is not to provide $g_s$, but
to provide $O_3$ at the top of the vegetation canopy. This firstly entails a calculation of the large-scale ozone
concentrations for Europe, which are represented by the gridded values of grid-cell average concentration, and
secondly to calculate the vertical gradients between these grid-cell centres (at ca. 45m) and the top of the
vegetation canopy. $O_3$ deposition is important for both steps; it is known to have a substantial impact on the
lifetime and concentrations of $O_3$ in the planetary boundary layer (Garland and Derwent, 1979;Val Martin et al.,
2014), and also in determining the local vertical gradients above different land-covers (CLRTAP, 2017;Gerosa et
al., 2017;Tuovinen et al., 2009). Vertical gradients between the 45m level and the top of forest canopies tend to
be limited (Fuentes et al., 2007;Karlsson et al., 2006) due to the good mixing normally induced by forest
roughness. Vertical gradients between 45m and the top of shorter vegetation such as grasslands or crops can be
larger however (CLRTAP, 2017;Gerosa et al., 2017). Accounting for such land-cover specific gradient effects has
been shown to have large impacts on estimates of $O_3$ metrics (Simpson et al., 2007).

These offline simulations show the sensitivity of GPP and the land carbon sink to tropospheric $O_3$, suggesting that
$O_3$ is an important predictor of future GPP and the land carbon store across Europe. There are uncertainties in our
estimates however from the use of uncoupled tropospheric chemistry, meteorology and stomatal function. For
example, increased frequency of drought in the future would reduce stomatal conductance (assuming no sluggish
stomatal response) and thus $O_3$ uptake. Since our offline simulations do not include this feedback it is possible the
$O_3$ effect is over estimated here. Given the complexity of potential interactions and feedbacks it remains difficult
to diagnose the importance of individual factors (e.g. the direct physiological response) in a fully coupled
simulation. Once the importance of a process is demonstrated offline, it provides evidence of the need to
incorporate such process in coupled regional and global simulations.
**4.4 $O_3$ as a missing component of carbon cycle assessments?**
Comprehensive analyses of the European carbon balance suggest a large biogenic carbon sink (Janssens et al.,
2003;Luyssaert et al., 2012;Schulze et al., 2009). However, estimates are hampered by large uncertainties in key
components of the land carbon balance, such as estimates of soil carbon gains and losses (Ciais et al.,
2010;Janssens et al., 2003;Schulze et al., 2009;Schulze et al., 2010). We suggest that the effect of $O_3$ on plant
physiology is a contributing factor to the decline in soil carbon stores observed across Europe, and as such this $O_3$
effect is a missing component of European carbon cycle assessments. Over the full experimental period (1901 to
2050), our results show elevated $O_3$ concentrations reduce the amount of carbon that can be stored in the soil by
3% to 9% (low and high plant $O_3$ sensitivity, respectively), which almost completely offsets the beneficial effects
of $CO_2$ fertilisation on soil carbon storage under the high plant $O_3$ sensitivity . This would contribute to a change

in the size of a key carbon sink for Europe, and is particularly important when we consider the evolution of the land carbon sink into the future given the impact of $O_3$ on soil carbon sequestration and the high uncertainty of future tropospheric $O_3$ concentrations. Schulze et al. (2009) and Luyssaert et al. (2012) extended their analysis of the European carbon balance to include additional non-$CO_2$ greenhouse gases ($CH_4$ and $N_2O$). Both studies found that emissions of these offset the biogenic carbon sink, reducing the climate mitigation potential of European ecosystems. This highlights the importance of accounting for all fluxes and stores in carbon/greenhouse gas balance assessments, of which $O_3$ and its indirect effect on the $CO_2$ flux via direct effects on plant physiology is currently missing.

**4.5 The interaction between $O_3$ and $CO_2$**

We looked at the interaction between $CO_2$ and $O_3$ effects. Our results support the hypothesis that elevated atmospheric $CO_2$ provides some protection against $O_3$ damage because of lower $g_s$ that reduces uptake of $O_3$ through stomata (Harmens et al., 2007;Wittig et al., 2007). In the present study, reductions in GPP and the land carbon store due to $O_3$ exposure were lower when simulated with concurrent changes in atmospheric $CO_2$. Despite acclimation of photosynthesis after long-term exposure to elevated atmospheric $CO_2$ of field grown plants (Ainsworth and Long, 2005;Medlyn et al., 1999), there is no evidence to suggest that $g_s$ acclimates (Ainsworth et al., 2003;Medlyn et al., 2001). This suggests the protective effect of elevated atmospheric $CO_2$ against $O_3$ damage will be sustained in the long term. However, although meta-analysis suggest a general trend of reduced $g_s$ with elevated $CO_2$ (Ainsworth and Long, 2005;Medlyn et al., 1999), this is not a universal response. Stomatal responses on exposure to elevated $CO_2$ with FACE treatment varied with genotype and growth stage in a fast-growing poplar community (Bernacchi et al., 2003;Tricker et al., 2009). In other mature forest stands, limited stomatal response to elevated $CO_2$ was observed after canopy closure (Ellsworth, 1999;Uddling et al., 2009). Also, some studies found that stomatal responses to $CO_2$ were significant only under high atmospheric humidity (Cech et al., 2003;Leuzinger and Körner, 2007;Wullschleger et al., 2002). These examples illustrate that stomatal responses to elevated atmospheric $CO_2$ are not universal, and as such the protective effect of $CO_2$ against $O_3$ injury cannot be assumed for all species, at all growth stages under wide ranging environmental conditions.

**5 Conclusion**

What is abundantly clear is that plant responses to both $CO_2$ and $O_3$ are complicated by a host of factors that are only partly understood, and it remains difficult to identify general, global patterns given that effects of both gases on plant communities and ecological interactions are highly context and species specific (Ainsworth and Long, 2005;Fuhrer et al., 2016;Matyssek et al., 2010b). This study quantifies the sensitivity of the land carbon sink for Europe and GPP to changing concentrations of atmospheric $CO_2$ and $O_3$ from 1901 to 2050. We have used a state of the art land surface model calibrated for European vegetation to give our best estimates of this sensitivity within the limits of data availability to calibrate the model for $O_3$ sensitivity, current knowledge and model structure. In summary, this study has shown that potential gains in terrestrial carbon sequestration over Europe resulting from elevated $CO_2$ can be partially offset by concurrent rises in tropospheric $O_3$ over 1901-2050. Specifically, we have shown that the negative effect of $O_3$ on the land carbon sink was greatest over the twentieth century, when $O_3$

concentrations increased rapidly from pre-industrial levels. Over this period soil carbon stocks were diminished
over agricultural areas, consistent with reduced NPP and litter input. This loss of soil carbon was largely
responsible for the decrease in the size of the land carbon sink over Europe. The $O_3$ effect on the land carbon store
and flux was reduced into the future as $CO_2$ concentration rose considerably and changes in $O_3$ concentration were
less pronounced. However, there remained a large cumulative negative impact on the land carbon sink for Europe
by 2050. The interaction between the two gases was found to reduce $O_3$ injury owing to reduced stomatal opening
in elevated atmospheric $CO_2$. However, primary productivity and land carbon storage remained suppressed by
2050 due to plant $O_3$ damage. Expressed as a percentage of the emissions from fossil fuel and cement production
for the EU28-plus countries, the carbon emissions from $O_3$-induced plant injury are a source of anthropogenic
carbon previously not accounted for in carbon cycle assessments. Our results demonstrate the sensitivity of
modelled terrestrial carbon dynamics to the direct effect of tropospheric $O_3$ and its interaction with atmospheric
$CO_2$ on plant physiology, demonstrating this process is an important predictor of future GPP and trends in the
land-carbon sink. Nevertheless, this process remains largely unconsidered in regional and global climate model
simulations that are used to model carbon sources and sinks and carbon-climate feedbacks.
**Data availability**
The JULES model can be downloaded from the Met Office Science Repository Service
(https://code.metoffice.gov.uk/trac/jules - see here for a helpful how to http://jules.jchmr.org/content/getting-
started). Model output data presented in this paper and the exact version of JULES with namelists are available
upon request from the corresponding author.
**Supplementary Information**
Supplementary_Information_Oliver_et_al.docx
**Competing Interests**
The authors declare that they have no conflict of interest
**Acknowledgements**

RJO and LMM were supported by the EU FP7 (ECLAIRE, 282910) and JWCRP (UKESM, NEC05816). This
work was also supported by EMEP under UNECE. SS and LMM acknowledge the support of the NERC
SAMBBA project (NE/J010057/1). The UK Met Office contribution was funded by BEIS under the Hadley Centre
Climate Programme (GA01101). GAF also acknowledges funding from the EU's Horizon 2020 research and
innovation programme (CRESCENDO, 641816). We also thank Magnuz Engardt of SMHI for providing the
RCA3 climate dataset. This work used eddy covariance data acquired and shared by the FLUXNET community,
including these networks: AmeriFlux, AfriFlux, AsiaFlux, CarboAfrica, CarboEuropeIP, CarboItaly, CarboMont,

ChinaFlux, Fluxnet-Canada, GreenGrass, ICOS, KoFlux, LBA, NECC, OzFlux-TERN, TCOS-Siberia, and USCCC. The ERA-Interim reanalysis data are provided by ECMWF and processed by LSCE. The FLUXNET eddy covariance data processing and harmonization was carried out by the European Fluxes Database Cluster, AmeriFlux Management Project, and Fluxdata project of FLUXNET, with the support of CDIAC and ICOS Ecosystem Thematic Center, and the OzFlux, ChinaFlux and AsiaFlux offices.

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
