# Peer review of "Large but decreasing effect of ozone on the European carbon"

_Biogeosciences, 2017_

## Referee Comment (RC1) · Anonymous Referee #1 · 20 Oct 2017

This paper investigated the interaction between CO2 and O3, the two greenhouse gases that directly affect plant photosynthesis, and indirectly gs. The goal of the paper is to quantify the impact of tropospheric O3, and its interaction with CO2, on gross primary productivity and land carbon storage across Europe from 1901 to 2050 using the JULES land-surface model. In principle, the analysis is highly topical and needed.

Throughout the abstract, it should be more quantitative in nature. For example, line 37-38, by how much does the tropospheric O3 suppress terrestrial carbon uptake?

Line 40-41, How much of the combined effects of elevated future CO2 (acting to reduce stomatal opening) and reductions in O3 concentrations resulted in reduced O3 damage? Moreover, elevated future CO2 will lead to climate warming simultaneously, so how do the authors remove the response of GPP and land carbon uptake to climate

warming due to the increased CO2 concentration? Warming will also increase evaporation (evapotranspiration) and reduce soil water availability, is this also considered?

Line 43, how large are the regional variations in temperate boreal regions?

Overall, some specific problems should be described in Introduction. For the O3 effect on the land C sink, what have we learned from the previous studies? What bioregions, and with what methods, have been studied?

Line 81-83, The authors mentioned few studies have considered the simultaneous effects of exposure to both O3 and CO2, so what have learned from these previous studies? Please specify previous findings.

Line 86-99, Please describe the O3 concentration for historical and current level in quantity. How does the O3 change over the last decades?

Line 103-104, High levels of O3 are reducing the land carbon sink. How many carbon loss was led to by O3 at regional and global scale based on previous studies?

Line 121-122, are you also going to study the effects of high temperature and drought?
 Please explain the CUO1 in Figure S2 caption. As shown in Table S1, the g1 parameter in NT (Needle leaf tree) is similar to that of shrub. Does it mean plant water use efficiency in NT and SH are same?

Figure1, could you provide some O3 concentration data from observations?

Line 342-345, what is the uncertainty (or SD) for these percentage number? It may be better if the authors mentioned how these number are calculated in methods. Line 352-353, For the broadleaf tree and C3 herbaceous PFT, the Medlyn model simulates a larger conductance and therefore a greater flux of O3 through stomata compared to Jacobs, but it also led to a greater flux of CO2 through stomata simultaneously, which may be helpful for increasing photosynthesis.

Line 366-368, Some Boreal and Mediterranean regions show increased GPP over this

period, associated with O3 induced stomatal closure enhancing water availability. But O3 induced stomatal closure also reduce the flux of CO2 through stomata simultaneously, which have a negative impact on GPP.

Line 373-375, is the different response of GPP to low and high plant O3 sensitivity are significant?

Line 437-440, CO2 induced stomatal closure can help alleviate O3 damage by reducing the uptake of O3, but it will also increase available soil moisture simultaneously.

Contradictions are reported in Figure 4 and 5. In Figure 4a, the areas with great increasing in plant available soil moisture have less change in gs in Figure 5a. Why? In figure 4c, the areas with decreasing in plant available soil moisture have large reduction in gs.

In table 1, O3 increased GPP but decreased land carbon over the period 2001-2050. Why does land C decrease when GPP is increasing?

The discussion could be improved by using subtitles more clearly.

Line 525-541, the authors listed a lot of results from the literatures, but the reader is left to decide what and why is the difference between this study and previous studies? More discussion on comparing this study with previous studies in detail would be helpful.

---

## Referee Comment (RC2) · Anonymous Referee #2 · 16 Nov 2017

Oliver et al. quantify the impact of ozone damage to European GPP and total land carbon stock on an annual basis. The authors apply a new stomatal conductance parameterization to their model, and force the model with surface ozone concentrations, meteorology and global $CO_2$ concentration to investigate the roles of $CO_2$ fertilization vs. O3 damage on GPP and total land carbon stock from 1901 to 2050. This new stomatal conductance parameterization simulates higher stomatal conductance than the previous, causing higher uptake of ozone through plant stomata. They find that there are spatial variations in the response of GPP and land carbon stock to $CO_2$ fertilization vs. O3 damage. On a regional basis, $CO_2$ fertilization dominates the response (vs ozone damage) when $CO_2$ is allowed to evolve, but ozone does limit the land carbon sink. The impact of ozone damage from 1901 to 2050 is dominated by 1901-2001

due to increasing surface ozone concentrations during that time. Overall, it seems like there is a lot more to discuss in regards to the previous work that has been done on the leaf level to global scale on this topic (e.g., Karnosky et al., 2003) and how the Oliver et al. findings contribute substantially to knowledge. It is not really clear how these results advance Sitch et al. 2007 except examining the region-scale over Europe. A huge limitation to this study is that $CO_2$ and meteorology are uncoupled, as well as meteorology, ozone, and stomatal conductance.

These are the major issues that I see with regards to Oliver et al. 1. Using recycled early 20th C climate is problematic. I understand that the authors want to isolate the physiological response of plants of $CO_2$ vs. $O_3$ here, but ozone is high during drought and heat waves, and stomata close at that time. So if there is increasing aridity and hydrological and temperature extremes into the 21st C, then the ozone response should be much lower than the authors suggest. The authors do at some point say that their work here is an upper bound, but upper bounds have already been published. 2. Further, Langner et al. 2012 is not the appropriate work here to justify the authors' approach. Langner et al. examine the impact of climate change following the A1B scenario on surface ozone (they do not consider changes in anthropogenic precursor emissions from present to future under A1B). Langner et al. use biogenic emissions to explain some of the cross-model differences in changes from present to future in ozone due to climate. This is quite different from using the full A1B scenario which considers changes in climate & anthropogenic precursor emissions, which is what Oliver et al. do. 3. The authors change their stomatal conductance parameterization but do not explain why. Their phrasing implies that the new gs model is truth, whereas the Jacobs 1994 model is not (e.g., "studies using the Jacobs [1994] formulation may underestimate" on line 523). I understand that parameterizations of stomatal conductance are uncertain in general, and hard to evaluate, but it seems like there should be some reasoning and evaluation here. Further, please clarify the re-calibration (lines 130-133). This seems like a major part of your analysis and I think evaluation & inclusion of this evaluation in the main part of the paper is warranted 4. Is Jacobs gs used in the ozone dry deposition

parameterization that is used in the EMEP model used to project the ozone concentrations? Typically stomatal conductance in the dry deposition parameterizations is some form of Wesely (1989). If Wesely is used, how does the magnitude of Medlyn differs from the magnitude of stomatal conductance from Wesely? If Wesely is used, then $CO_2$ fertilization is not in there, nonetheless ozone damage. Another caveat is that ozone damage can feedback onto ozone concentrations as demonstrated by Sadiq et al. ACP 2017. 5. A large part of the results hinge on the seasonality of surface ozone concentrations, and how they change from PI to present. There is some discussion of this on pages 401-405 as authors examine the change in seasonality from 2001 to 2050, but there is no citation of previous work examining changes in ozone seasonality, or the implications of this for their conclusions. Also, the authors say that tropospheric ozone is increasing (e.g., line 74), but I think this is a bit misleading - due to strong changes in seasonality that are observed - please revise. 6. In general, the paper is a bit poorly organized. Many times the authors say "see details in SI" when it's not clear what information is in there, and why it is relevant. Further it seems like some info in the SI should really be in the actual paper. In addition, the authors neglect to mention many substantial caveats (such as the uncertainties around $CO_2$ fertilization w.r.t. nutrient cycling, using uncoupled tropospheric chemistry & stomatal dry deposition, and stomatal sluggishness) until the very end. I think the paper would be much better if much of the discussion was moved to the introduction and used to frame the work, and motivate the authors' objectives.

Minor comments 1. The authors use the term "significant" a lot - but don't do any sort of statistical testing. Please only use the word significant when describing results that are statistically significant. 2. Line 78: Lightning is a source of NOx, not O3 - please revise 3. Lines 86-87: Parrish et al. 2012 is not really the appropriate citation here 4. Lines 93-94: "Intercontinental transport" doesn't mean that background ozone has increased, there has always been intercontinental transport 5. Line 101: Citations for ozone impacts on crop yields and nutritional quality are needed 6. Line 106: Do the authors mean indirect here? 7. Line 110: Fowler et al. 2009 isn't really the appropriate

citation here - i.e., for saying that dry deposition is a substantial sink of tropospheric ozone 8. Line 152 - as the authors mention in the discussion, ozone can directly impact gs - please revise accordingly 9. Line 160-168: please clarify the spatial domain and the resolution of this model; also, is the resolution the same as the meteorological and ozone forcing files? 10. Lines 193-194: kappa_O3 is not exactly the ratio of the resistances; it's the ratio of the diffusivities 11. Lines 220-222: What is CLRTAP (2017)? It is not in the references. Why is it being treated as the "truth"? 12. Section 2.3 - please clarify that Lin et al. 2015 fit g1 parameters based on the Medlyn et al. 2011 equation for stomatal conductance (except no g0 term), which is not exactly the same as putting equation 7 into equation 5; it's confusing to refer to this equation as Medlyn et al. (2011); also, I do not think that multiplying the Anet/(Ca-Ci) by R*T is the right way to convert from mol s-1 m-2 to m/s. Lastly, this combing equation 13. Lines 252-254: Please clarify "the effect" that Hoshika et al. 2013 find; does O3 increase or decrease WUE? This seems relevant to your discussion/conclusions 14. Lines 300-301: Clarify the "disaggregation" of ozone from the daily mean to the hourly time step. As ozone has a diurnal cycle, and stomatal conductance does as well, this could have a substantial impact on your work, and should be discussed. 15. Lines 305-306: Clarify the calculation of the ozone gradient from the lowest atmosphere grid box to canopy height 16. Further details on crops in JULES should be included in Section 2.4.1 in addition to the discussion. 17. Lines 282-283: Please specify the ozone sensitivity used for forests 18. Line 882: I don't think "in prep" studies can be cited 19. Lines 258-259:What are the two model grid points? What does wet vs. dry refer to? This info is used later on in the paper (Figure 2), so it would be helpful for more information on this. 20. Please clarify in the Figure 2 caption what exactly the readers are looking at (this is just one grid cell, with each sub-tile PFT gs shown?). Why just one grid-cell? Is the data shown hourly? What is the time period? 21. Lines 384-396: It's not clear why the authors are examining different decades for their analysis here. Second, it seems like the authors could pretty easily sample their model for an apples-to-apples comparison with Boden et al. 2013. Third, suggesting that the O3 impact on the land carbon sink

is a source of carbon is not really appropriate (lines 395-396); re-phrasing would allow for the same take-away 22. Lines 401-402: Ozone precursor emission controls do not always lead to ozone reductions because formation chemistry is nonlinear; please revise 23. Line 401: Large spatial variability is not apparent to me - it would be helpful if the authors were more specific 24. Lines 405-408: it's not clear what figure the authors are talking about here. 25. Figure 6 - specify whether your numbers correspond to rows or columns

---

## Author Comment (AC1) · 30 Jan 2018

Response to RC1:

We would like to thank the reviewer for their time taken to read and comment on this manuscript. The comments have been very helpful to improve the manuscript. We hope we have addressed the comments to the full satisfaction of the reviewer. We attach the revised manuscript with track changes so it can be seen what has been changed and where.

This paper investigated the interaction between CO2 and O3, the two greenhouse gases that directly affect plant photosynthesis, and indirectly gs. The goal of the paper is to quantify the impact of tropospheric O3, and its interaction with CO2, on gross

primary productivity and land carbon storage across Europe from 1901 to 2050 using the JULES land-surface model. In principle, the analysis is highly topical and needed.

RC1) Throughout the abstract, it should be more quantitative in nature. For example, line 37-38, by how much does the tropospheric O3 suppress terrestrial carbon uptake?

AC: We have modified the abstract to make it more quantitative (lines 37 to 47).

RC2)Line 40-41, How much of the combined effects of elevated future CO2 (acting to reduce stomatal opening) and reductions in O3 concentrations resulted in reduced O3 damage? Moreover, elevated future CO2 will lead to climate warming simultaneously, so how do the authors remove the response of GPP and land carbon uptake to climate warming due to the increased CO2 concentration? Warming will also increase evaporation (evapotranspiration) and reduce soil water availability, is this also considered?

AC: We have added a sentence to the abstract to show that the alleviation of O3 damage by CO2 induced stomatal closure was around 1 to 2% for the low and high plant O3 sensitivities respectively (for both GPP and land C, line 42). This is discussed in more detail in the original manuscript in the Results section 3.4 pg. 18 on lines 609 to 614 and in Table S6.

This study uses a fixed climate. We cycle over the climate from 1901 to 1920 so we maintain natural climate variability, but we do not have climate change. This allows us to focus on the direct effects of changing atmospheric CO2 and O3 concentrations, and their complex interaction, on plant physiology through the twentieth century and into the future. We acknowledge the use of a 'fixed' pre-industrial climate omits the additional factor of the interaction between climate change and gs which will affect the rate of O3 uptake and therefore O3 concentrations. Nevertheless, these simulations are an important tool to understand the direct impacts of O3 at the land surface. This work demonstrates the sensitivity of GPP and the land carbon sink to tropospheric O3, highlighting that it is an important predictor of future GPP. We do state in the original manuscript that we use a fixed climate (methods section 2.4.1 line 373), but we realise that we do not make it clear enough early on in the manuscript that we use a fixed climate, so we have amended this in the introduction (pg.6 , lines 214 to 223). We also add a paragraph to our discussion about potential impacts on our results (section 4.3, pg. 26, lines 808 to 816).

RC3)Line 43, how large are the regional variations in temperate boreal regions? Overall, some specific problems should be described in Introduction. For the O3 effect on the land C sink, what have we learned from the previous studies? What bioregions, and with what methods, have been studied?

AC: We have added a new paragraph to the introduction to discuss the findings from previous studies from different regions (pg. 4, lines 134 to 157). Also, we have quantified the regional variations in the abstract (line 44), and in the original manuscript we discuss these spatial variations in greater detail in the results section.

RC4)Line 81-83, The authors mentioned few studies have considered the simultaneous effects of exposure to both O3 and CO2, so what have learned from these previous studies? Please specify previous findings.

AC: See response to comment above. We have added a new paragraph to the introduction to discuss previous studies, both field and model-based, and what has been shown from these studies (pg. 4, lines 134 to 157).

RC5)Line 86-99, Please describe the O3 concentration for historical and current level in quantity. How does the O3 change over the last decades?

AC: We have added more information about the change in O3 concentrations from historical to present day (pg.3, lines 88 to 98).

RC6)Line 103-104, High levels of O3 are reducing the land carbon sink. How many carbon loss was led to by O3 at regional and global scale based on previous studies?

AC: We have added a paragraph to the introduction to discuss the findings of previous studies (pg. 4, lines 134 to 157).

RC7)Line 121-122, are you also going to study the effects of high temperature and drought?

AC: This links to an earlier comment – in this study we have used a fixed pre-industrial climate. We cycle over the climate from 1901 to 1920 so we maintain natural climate variability, but we do not have climate change. This allows us to focus on the direct effects of changing atmospheric $CO_2$ and $O_3$ concentrations, and their complex interaction, on plant physiology through the twentieth century and into the future. We realise that we do not make it clear enough early on in the manuscript that we use a fixed climate, so we have amended this in the introduction (pg.6 , lines 214 to 223). We also add a paragraph to our discussion about potential impacts on our results (section 4.3, pg. 26, lines 808 to 816).

RC8)Please explain the CUO1 in Figure S2 caption. As shown in Table S1, the g1 parameter in NT (Needle leaf tree) is similar to that of shrub. Does it mean plant water use efficiency in NT and SH are same?

AC: We have clarified this by adding the following to the Figure S2 caption: "The x axis is cumulative uptake of $O_3$ (CUO) above the critical $O_3$ threshold (FO3crit)." The parameter values for g1 were derived from the extensive database of Lin et al., (2015). The parameter g1 is a measure of water-use efficiency. In the model, the plant water-use efficiencies of NT and SH would be similar but not identical since WUE is the ratio of carbon gain to water loss and the two PFTs have different photosynthetic rates owing to different parameter values.

RC9)Figure1, could you provide some $O_3$ concentration data from observations?

AC: This links to an earlier comment #5. Comparison of these long-term $O_3$ trends with observations is difficult for many reasons, not least lack of reliable data before recent decades, and limited representativity and inconsistencies in data from recent years (Logan et al., 2012;Parrish et al., 2012). For example, ozone levels at the start of the 20th century are estimated to be around 10 ppb for the site Montsouris Observatory

near Paris, data for Arkona on the Baltic coast increased from ca. 15 ppb in the 1950s to 20-27 ppb by the early 1980s, and the Irish coast site Mace Head shows around 40 ppb by the year 2000 (Logan et al., 2012, Parrish et al., 2012, and refs within). Trends vary from site to site though, even on a decadal basis (Logan et al., 2012;Simpson et al., 2014), depending, for example, on local/regional trends in precursor (especially NOx) emissions, elevation, and exposure to long-range transport. As a result of this spatial variation in O3 concentrations across Europe, comparison of the EMEP O3 forcing in Fig. 1 (plotted as a mean across regions) with individual sites would potentially be misleading.

RC10)Line 342-345, what is the uncertainty (or SD) for these percentage number? It may be better if the authors mentioned how these number are calculated in methods.

AC: These numbers were re-calculated to get the standard deviation. Previously the annual mean for each simulation was calculated, and this used to calculate the percentage difference. To get the standard deviation the daily means were calculated and the percentage difference was calculated for each day, then the mean and standard deviation were calculated, these values are now reported in the manuscript (section 3.1, pg. 14, lines 482 to 487). We explain how these numbers are calculated in the SI section S3 (lines 111 to 114). This re-calculation slightly changes the percentage differences in annual mean leaf-level stomatal conductance, but the direction of change remains the same, i.e. MED increases water-use for BT and C3, and reduced water-use for NT, C4 and SH. The standard deviations are quite large reflecting the large spread in the data, partly due to the seasonal cycle.

RC11)Line 352-353, For the broadleaf tree and C3 herbaceous PFT, the Medlyn model simulates a larger conductance and therefore a greater flux of O3 through stomata compared to Jacobs, but it also led to a greater flux of CO2 through stomata simultaneously, which may be helpful for increasing photosynthesis.

AC: Figures S7 and S8 (top rows) show 1:1 plots of Anet, plotting MED (y axis) against

JAC (x axis). These plots show that in the model, Anet is not as sensitive to the change in gs scheme as gs itself (Fig. 2 and Fig. S6). Although the greater conductance for BT and C3 with MED will result in high internal $CO_2$ concentrations, this doesn't result in a large change in modelled photosynthetic rates because in the model, the sensitivity of the limiting rates of photosynthesis to changes in ci is much lower that the sensitivity of gs to the same change (see section 3.1, pg. 14, lines 488 to 490 where this is mentioned). Therefore, the WUE for BT and C3 will change, they are less WUE with MED.

RC12)Line 366-368, Some Boreal and Mediterranean regions show increased GPP over this period, associated with O3 induced stomatal closure enhancing water availability. But O3 induced stomatal closure also reduce the flux of CO2 through stomata simultaneously, which have a negative impact on GPP.

AC: This is a trade-off between the opposing effects of O3 induced stomatal closure enhancing soil water availability and also reducing GPP. The overall effect occurs seasonally, which is not shown in Figs. 4, 5, & 6. O3 induced stomatal closure occurs during spring/early summer when O3 concentrations are highest, at this point GPP is reduced, but in these dry regions this leads to increased soil moisture that, in the model, allows growth later in the year when conditions are still favourable but soil moisture may otherwise have been limiting. We have clarified this point in the text (section 3.3, pg. 16, lines 534-538): "Some Boreal and Mediterranean regions show small increases in GPP over this period, associated with O3 induced stomatal closure enhancing water availability in these drier regions (Fig. 5). This allows for greater stomatal conductance later in the year when soil moisture may otherwise have been limiting to growth (up to 10%, Fig. 6), and therefore higher GPP, but these regions comprise only a small area of the entire domain."

RC13)Line 373-375, is the different response of GPP to low and high plant O3 sensitivity are significant?

AC: On the advice of the reviewer, we carry out statistical testing of the different responses of GPP to the low and high plant O3 sensitivities. We use the software R, and use paired t-tests to determine whether the O3 effect on GPP is significantly different between the two different plant sensitivity parameterisations (section 3.3 line 531; section 3.4 line 570; section 3.5 line 597).

RC14)Line 437-440, CO2 induced stomatal closure can help alleviate O3 damage by reducing the uptake of O3, but it will also increase available soil moisture simultaneously.

AC: We agree. The CO2 induced stomatal closure is the dominant effect that helps alleviate O3 damage. Figures S14 and S15 show that in the model the effect of CO2 on gs is large, whereas the effect of CO2 on soil moisture availability (fsmc in these plots) is small in comparison. Simulated gs declines with increasing CO2 which may increase available soil moisture, however CO2 enhances GPP and growth of the vegetation which can increase LAI leading to higher water-use on a leaf area basis. These responses are all captured in our simulation however with both CO2 and O3 changing, and in our calculation of the O3 effect with CO2 rising. We look at the difference in this simulation to the simulation with O3 changing but CO2 concentration fixed at pre-industrial concentrations, this gives us the alleviation of O3 damage by increasing CO2 and all associated effects, such as changes in soil moisture, but it is the effect on stomata that dominates. We have clarified the calculation of the alleviation of O3 damage by increasing CO2 in the legend to Table S6.

AC15)Contradictions are reported in Figure 4 and 5. In Figure 4a, the areas with great increasing in plant available soil moisture have less change in gs in Figure 5a. Why? In figure 4c, the areas with decreasing in plant available soil moisture have large reduction in gs.

AC: The fsmc formulation (factor determining plant available soil moisture) varies with soil moisture content and is non-linear. It has a value of zero below wilting point then linearly increases to a value of 1 at field capacity, and remains at this value beyond. The wilting point and field capacity depend on soil texture. Therefore a small percentage change in soil water content in dry regions (Mediterranean) can result in a large percentage increase in fsmc. Likewise an increase in soil moisture in mesic areas (e.g. northern Europe) may translate into relatively small percentage changes in fsmc.

In Figs 5a and 6a, the areas that see a large increase in plant available soil moisture see a small increase in gs (up to 10%). This is looking at the change over the period 1901 to 2001 when only O3 is changing. Over this period there is a large increase in O3, so the O3 induced stomatal closure is large, causing the increase in fsmc in this region. The changes are seasonal, these plots show the annual mean which will average out some of the change. In this region, for example, gs increases a lot in JJA and DJF, but there is minimal change in SON/MAM. Figures 5c and 6c are for a different time period, 2001 to 2050. Over this period the O3 effect is reduced considerably, so by 2050 plant available water is reduced on 2001 levels because the O3 induced stomatal closure is less. Stomatal conductance decreases in this region during this period.

RC16)In table 1, O3 increased GPP but decreased land carbon over the period 2001-2050. Why does land C decrease when GPP is increasing?

AC: We refer to this in the results section (section 3.4, pg. 17, lines 585 to 589). GPP is a fast flux, whereas the land carbon store is a slower pool of carbon, it takes longer for this carbon store to adjust to changes in the flux, especially when those changes are fairly small as is the case here. This highlights the importance of using a carbon cycle model to look at the impacts of O3 on the terrestrial biosphere.

RC) The discussion could be improved by using subtitles more clearly.

AC: We have amended this and added subtitles.

RC17)Line 525-541, the authors listed a lot of results from the literatures, but the reader is left to decide what and why is the difference between this study and previous stud-
ies? More discussion on comparing this study with previous studies in detail would be helpful.

AC: This paragraph discusses findings from field-based studies looking at plant O3 impacts. We have removed this paragraph and put it in the introduction as it seemed more appropriate here to place our study in context (pg. 4, lines 134 to 157).

Refs: Logan, J. A., Staehelin, J., Megretskaia, I. A., Cammas, J. P., Thouret, V., Claude, H., De Backer, H., Steinbacher, M., Scheel, H. E., Stübi, R., Fröhlich, M., and Derwent, R.: Changes in ozone over Europe: Analysis of ozone measurements from sondes, regular aircraft (MOZAIC) and alpine surface sites, Journal of Geophysical Research, 117, 1-23, 2012. Parrish, D. D., Law, K. S., Staehelin, J., Derwent, R., Cooper, O. R., Tanimoto, H., Volz-Thomas, A., Gilge, S., Scheel, H. E., Steinbacher, M., and Chan, E.: Long-term changes in lower tropospheric baseline ozone concentrations at northern mid-latitudes, Atmos. Chem. Phys., 12, 11485-11504, 10.5194/acp-12-11485-2012, 2012. Simpson, D., Arneth, A., Mills, G., Solberg, S., and Uddling, J.: Ozone—the persistent menace: interactions with the N cycle and climate change, Current Opinion in Environmental Sustainability, 9, 9-19, 2014.

Please also note the supplement to this comment:
https://www.biogeosciences-discuss.net/bg-2017-409/bg-2017-409-AC1-supplement.pdf

**Supplement:**

[revised manuscript text omitted]

FLUXNET2015 sites are shown as black line with grey vertical bars, JULES with the JAC $g_s$ model is shown in red and JULES with the MED $g_s$ model are shown in purple. Also shown are the root mean squared error (rmse)

for each simulation.

**S5 Evaluation of JULES O₃ model**

**Commented [ORJ45]:** RC2 3)
Site level evaluation of the gs formulations.

[Figure]

**Figure S10**. Mean GPP (g C m$^2$ yr$^{-1}$) from 1991 to 2001 for a) the observations based globally extrapolated Flux

Network model tree ensemble (MTE) (Jung et al., 2011); b, c) model simulations with transient $CO_2$ and fixed

$O_3$; d, e)model simulations with fixed $CO_2$ and transient $O_3$, and f, g) our model simulations with transient $CO_2$

and transient $O_3$. All model simulations show GPP for high and low plant $O_3$ sensitivity respectively.

[Figure]

**Figure S11**. Zonal mean GPP from 1991 to 2001 for FLUXNET-MTE (Jung) and all JULES scenario simulations with both high (solid lines) and low (dashed lines) plant $O_3$ sensitivity.

[Figure]

Figure S12. Mean GPP from 1991 to 2001 shown by region, comparing MTE (Jung) and all JULES scenario
simulations with both high and low plant $O_3$ sensitivity.

Commented [ORJ46]: RC2 3)

**S6 Estimation of effects due to $O_3$, $CO_2$ and $O_3$ with $CO_2$**

[revised manuscript text omitted]

**Table S6.** Percentage reduction in simulated GPP and Land C by 2050 due to future $O_3$ effects at pre-industrial (PI) $CO_2$ concentration, and under increasing future $CO_2$ concentration. The difference between these defines the alleviation of the $O_3$ effect by $CO_2$. **$O_3$** = Fixed 1901 $CO_2$, Varying $O_3$ ; **$CO_2$** = Varying $CO_2$, Fixed 1901 $O_3$ ; **$CO_2$ + $O_3$** = Varying $CO_2$, Varying $O_3$. Calculated as: †) $O_3$ effect with fixed pre-industrial $CO_2$: 100*(fixCO$_2$_varO$_3$[2050] − value[1901])/value[1901], where value[1901] represents the hypothetical value at 2050 from a run with fixCO$_2$_fixO$_3$ which is equivalent to the initial state, i.e. the value in 1901 ; ‡) $O_3$ effect with increasing $CO_2$: 100*(varCO$_2$_varO$_3$[2050] - varCO$_2$_fixO$_3$[2050])/varCO$_2$_fixO$_3$[2050] ; ††) the alleviation of $O_3$ damage by $CO_2$ is the difference between the two runs: ‡ - †.

> **Commented [ORJ47]:** RC1 14)

**Acknowledgments**

This work used eddy covariance data acquired and shared by the FLUXNET community, including these networks: AmeriFlux, AfriFlux, AsiaFlux, CarboAfrica, CarboEuropeIP, CarboItaly, CarboMont, ChinaFlux, Fluxnet-Canada, GreenGrass, ICOS, KoFlux, LBA, NECC, OzFlux-TERN, TCOS-Siberia, and USCCC. The ERA-Interim reanalysis data are provided by ECMWF and processed by LSCE. The FLUXNET eddy covariance data processing and harmonization was carried out by the European Fluxes Database Cluster, AmeriFlux Management Project, and Fluxdata project of FLUXNET, with the support of CDIAC and ICOS Ecosystem Thematic Center, and the OzFlux, ChinaFlux and AsiaFlux offices.

---

## Author Response (AR1)

[revised manuscript text omitted]

Commented [ORJ5]: RC2 minor comment 4.

Field Code Changed

Commented [ORJ6]: RC2 minor comment 5.

Commented [ORJ7]: RC2 minor comment 6.

Commented [ORJ8]: RC2 minor comment 7.

concentrations for the future reduced aboveground biomass (-18%) , CI -13% to -24%), photosynthetic rate (-20%) and $g_s$ (-22%). One of few long-term field based $O_3$ exposure studies (AspenFACE) showed that after 11 years of exposing mature trees to elevated $O_3$ concentrations, $O_3$ decreased ecosystem carbon content (-9%), and decreased NPP (-10%), although the $O_3$ effect decreased through time (Talhelm et al., 2014). Zak et al. (2011) showed this was partly due to a shift in community structure as $O_3$-tolerant species, competitively inferior in low $O_3$ environments, out competed $O_3$-sensitivie species. Zak et al. (2011) GPP was reduced (-12% to -19%) at two Mediterranean ecosystems exposed to elevated $O_3$ (dominated by either *Pinus* species or *Citrus* species) studied by Fares et al. (2013). Biomass of mature beech trees was reduced (-44%) after 8 years of exposure to elevated $O_3$ (Matyssek et al., 2010a). After 5 years of $O_3$ exposure in a semi-natural grassland, annual biomass production was reduced (-23%), and in a Mediterranean annual pasture $O_3$ exposure significantly reduced total aboveground biomass (up to -25%) (Calvete-Sogo et al., 2014). However, these were empirical studies at individual sites, and these focus on $O_3$ effects on plant physiology and productivity, but do not quantify the impact on the land carbon sink. Modelling studies are needed to scale site observations to the regional and global scales. Models generally suggest that plant productivity and carbon sequestration will decrease with $O_3$ pollution, though the magnitudes vary. For example, based on a limited dataset to parameterise plant $O_3$ damage for a global set of plant functional types, Sitch et al. (2007) predicted a decline in global GPP of 14 to 23% by 2100. A second study by Lombardozzi et al. (2015) similarly predicted a 10.8% decrease of global GPP. Here we take a regional approach and take advantage of new measurements specifically for European vegetation and conduct a dedicated analysis for the European region. Results from the present study suggest projected $O_3$ concentrations for 2050 will reduce mean GPP for Europe (-4% to -9%), NPP (-6% to -11%), total carbon content (-3% to -7%) and $g_s$ (-4% to -9%). Using GPP as a proxy for $A_{sat}$ (these variables are not identical but they are related), our mean GPP and $g_s$ estimates fall within the range given by the meta-analysis of Wittig et al. (2007). The remaining studies are not meta-analyses, so are site- and species- specific, our estimates appear to compare more conservatively with these, however these are a mean value for Europe and spatially our estimates show greater variability.

Commented [ORJ9]: RC1 3/4/6

Understanding the response of plants to elevated tropospheric $O_3$ is challenged by the large variation in $O_3$ sensitivity both within and between species (Karnosky et al., 2007;Kubiske et al., 2007;Wittig et al., 2009). Additionally, other environmental stresses that affect stomatal behaviour will affect the rate of $O_3$ uptake and therefore the response to $O_3$ exposure, such as high temperature, drought and changing concentrations of atmospheric $CO_2$ (Mills et al., 2016;Fagnano et al., 2009;Kitao et al., 2009;Löw et al., 2006), such that the response of vegetation to $O_3$ is a balance between opposing drivers of stomatal behaviour. Increasing concentrations of atmospheric $CO_2$, for example, are suggested to provide some protection against $O_3$ damage by causing stomata to close (Harmens et al., 2007;Wittig et al., 2007), however the long-term effects of $CO_2$ fertilisation The long-term effects of $CO_2$ fertilization on plant growth and carbon storage remain are nevertheless uncertain (Baig et al., 2015;Ciais et al., 2013). Further, in some studies, stomata have been shown to respond sluggishly, losing their responsiveness to environmental stimuli with exposure to $O_3$ which can lead to higher $O_3$ uptake, increased water-loss and therefore greater vulnerability to environmental stresses such as drought (Mills et al., 2016;Mills et al., 2009;Paoletti and Grulke, 2010;Wilkinson and Davies, 2009). ….Mention uncertainties

Field Code Changed around CO2 fertilisation here, nutrient cycling and stomatal sluggishness here. Maybe introduce Medlyn model
here

[revised manuscript text omitted]

Commented [ORJ45]: RC2 3)
Site level evaluation of the gs formulations.

**S5 Evaluation of JULES O$_3$ model**

[Figure]

Figure S10. Mean GPP (g C m$^2$ yr$^{-1}$) from 1991 to 2001 for a) the observations based globally extrapolated Flux Network model tree ensemble (MTE) (Jung et al., 2011); b, c) model simulations with transient $CO_2$ and fixed $O_3$; d, e)model simulations with fixed $CO_2$ and transient $O_3$, and f, g) our model simulations with transient $CO_2$ and transient $O_3$. All model simulations show GPP for high and low plant $O_3$ sensitivity respectively.

[Figure]

**Figure S11**. Zonal mean GPP from 1991 to 2001 for FLUXNET-MTE (Jung) and all JULES scenario
simulations with both high (solid lines) and low (dashed lines) plant $O_3$ sensitivity.

[Figure]

**Figure S12.** Mean GPP from 1991 to 2001 shown by region, comparing MTE (Jung) and all JULES scenario simulations with both high and low plant $O_3$ sensitivity.

**Commented [ORJ46]:** RC2 3)

**S6 Estimation of effects due to $O_3$, $CO_2$ and $O_3$ with $CO_2$**

[revised manuscript text omitted]

**Table S6.** Percentage reduction in simulated GPP and Land C by 2050 due to future $O_3$ effects at pre-industrial (PI) $CO_2$ concentration, and under increasing future $CO_2$ concentration. The difference between these defines the alleviation of the $O_3$ effect by $CO_2$. **$O_3$** = Fixed 1901 $CO_2$, Varying $O_3$ ; **$CO_2$** = Varying $CO_2$, Fixed 1901 $O_3$ ; **$CO_2 + O_3$** = Varying $CO_2$, Varying $O_3$. Calculated as: †) $O_3$ effect with fixed pre-industrial $CO_2$: 100*(fixCO2_varO3[2050] − value[1901])/value[1901], where value[1901] represents the hypothetical value at 2050 from a run with fixCO2_fixO3 which is equivalent to the initial state, i.e. the value in 1901 ; ‡) $O_3$ effect with increasing $CO_2$: 100*(varCO2_varO3[2050] - varCO2_fixO3[2050])/varCO2_fixO3[2050] ; ††) the alleviation of $O_3$ damage by $CO_2$ is the difference between the two runs: ‡ - †.

**Commented [ORJ47]:** RC1 14)

**Acknowledgments**

This work used eddy covariance data acquired and shared by the FLUXNET community, including these networks: AmeriFlux, AfriFlux, AsiaFlux, CarboAfrica, CarboEuropeIP, CarboItaly, CarboMont, ChinaFlux, Fluxnet-Canada, GreenGrass, ICOS, KoFlux, LBA, NECC, OzFlux-TERN, TCOS-Siberia, and USCCC. The ERA-Interim reanalysis data are provided by ECMWF and processed by LSCE. The FLUXNET eddy covariance data processing and harmonization was carried out by the European Fluxes Database Cluster, AmeriFlux Management Project, and Fluxdata project of FLUXNET, with the support of CDIAC and ICOS Ecosystem Thematic Center, and the OzFlux, ChinaFlux and AsiaFlux offices.

**References**

Büker, P., Feng, Z., Uddling, J., Briolat, A., Alonso, R., Braun, S., Elvira, S., Gerosa, G., Karlsson, P. E., Le Thiec, D., Marzuoli, R., Mills, G., Oksanen, E., Wieser, G., Wilkinson, M., and Emberson, L. D.: New flux based dose-response relationships for ozone for European forest tree species, Environmental Pollution, 163-174, 2015.

CLRTAP: The UNECE Convention on Long-range Transboundary Air Pollution. Manual on Methodologies and Criteria for Modelling and Mapping Critical Loads and Levels and Air Pollution Effects, Risks and Trends: Chapter III Mapping Critical Levels for Vegetation, accessed via, http://icpvegetation.ceh.ac.uk/publications/documents/Chapter3-Mappingcriticallevelsforvegetation_000.pdf, 2017.

IGBP-DIS: International Geosphere-Biosphere Programme, Data and Information System, Potsdam, Germany. Available from Oak Ridge National Laboratory Distributed Active Archive Center, Oak Ridge, TN, anailable at: http://www.daac.ornl.gov,

Jung, M., Reichstein, M., Margolis, H. A., Cescatti, A., Richardson, A. D., Arain, M. A., Arneth, A., Bernhofer, C., Bonal, D., Chen, J., Gianelle, D., Gobron, N., Kiely, G., Kutsch, W., Lasslop, G., Law, B. E., Lindroth, A., Merbold, L., Montagnani, L., Moors, E. J., Papale, D., Sottocornola, M., Vaccari, F., and Williams, C.: Global patterns of land-atmosphere fluxes of carbon dioxide, latent heat, and sensible heat derived from eddy covariance, satellite, and meteorological observations, Journal of Geophysical Research: Biogeosciences, 116, n/a-n/a, 10.1029/2010JG001566, 2011.

Karlsson, P. E., Braun, S., Broadmeadow, M., Elvira, S., Emberson, L., Gimeno, B. S., Le Thiec, D., Novak, K., Oksanen, E., Schaub, M., Uddling, J., and Wilkinson, M.: Risk assessments for forest trees: The performance of the ozone flux versus the AOT concepts, Environmental Pollution, 146, 608-616, http://dx.doi.org/10.1016/j.envpol.2006.06.012, 2007.

O'Connor, F. M., Johnson, C. E., Morgenstern, O., Abraham, N. L., Braesicke, P., Dalvi, M., Folberth, G. A., Sanderson, M. G., Telford, P. J., Voulgarakis, A., Young, P. J., Zeng, G., Collins, W. J., and Pyle, J. A.: Evaluation of the new UKCA climate-composition model – Part 2: The Troposphere, Geosci. Model Dev., 7, 41-91, 10.5194/gmd-7-41-2014, 2014.

Tans, P., and Keeling, R.: Dr. Pieter Tans, NOAA/ESRL (www.esrl.noaa.gov/gmd/ccgg/trends/) and Dr. Ralph Keeling, Scripps Institution of Oceanography (scrippsco2.ucsd.edu/).

Weedon, G. P.: Readme file for the "WFDEI" dataset.available at: http://www.eu-watch.org/gfx_content/documents/README-WFDEI.pdf, 2013.

**Response to RC1:**

We would like to thank the reviewer for their time taken to read and comment on this manuscript. The comments have been very helpful to improve the manuscript. We hope we have addressed the comments to the full satisfaction of the reviewer. We attach the revised manuscript with track changes so it can be seen what has been changed and where. In our response below, the reviewer comments are in bold to distinguish from our responses.

**This paper investigated the interaction between CO2 and O3, the two greenhouse gases that directly affect plant photosynthesis, and indirectly gs. The goal of the paper is to quantify the impact of tropospheric O3, and its interaction with CO2, on gross primary productivity and land carbon storage across Europe from 1901 to 2050 using the JULES land-surface model. In principle, the analysis is highly topical and needed.**

**1) Throughout the abstract, it should be more quantitative in nature. For example, line 37-38, by how much does the tropospheric O3 suppress terrestrial carbon uptake?**

We have modified the abstract to make it more quantitative (lines 37 to 47).

**2) Line 40-41, How much of the combined effects of elevated future CO2 (acting to reduce stomatal opening) and reductions in O3 concentrations resulted in reduced O3 damage? Moreover, elevated future CO2 will lead to climate warming simultaneously, so how do the authors remove the response of GPP and land carbon uptake to climate warming due to the increased CO2 concentration? Warming will also increase evaporation (evapotranspiration) and reduce soil water availability, is this also considered?**

We have added a sentence to the abstract to show that the alleviation of $O_3$ damage by $CO_2$ induced stomatal closure was around 1 to 2% for the low and high plant $O_3$ sensitivities respectively (for both GPP and land C, line 42). This is discussed in more detail in the original manuscript in the Results section 3.4 pg. 18 on lines 609 to 614 and in Table S6.

This study uses a fixed climate. We cycle over the climate from 1901 to 1920 so we maintain natural climate variability, but we do not have climate change. This allows us to focus on the direct effects of changing atmospheric $CO_2$ and $O_3$ concentrations, and their complex interaction, on plant physiology through the twentieth century and into the future. We acknowledge the use of a 'fixed' pre-industrial climate omits the additional factor of the interaction between climate change and $g_s$ which will affect the rate of $O_3$ uptake and therefore $O_3$ concentrations. Nevertheless, these simulations are an important tool to understand the direct impacts of $O_3$ at the land surface. This work demonstrates the sensitivity of GPP and the land carbon sink to tropospheric $O_3$, highlighting that it is an important predictor of future GPP. We do state in the original manuscript that we use a fixed climate (methods section 2.4.1 line 373), but we realise that we do not make it clear enough early on in the manuscript that we use a fixed climate, so we have amended this in the introduction (pg.6 , lines 214 to 223). We also add a paragraph to our discussion about potential impacts on our results (section 4.3, pg. 26, lines 808 to 816).

**3) Line 43, how large are the regional variations in temperate boreal regions? Overall, some specific problems should be described in Introduction. For the O3 effect on the land C sink, what have we learned from the previous studies? What bioregions, and with what methods, have been studied?**

We have added a new paragraph to the introduction to discuss the findings from previous studies from different regions (pg. 4, lines 134 to 157). Also, we have quantified the regional variations in the abstract (line 44), and in the original manuscript we discuss these spatial variations in greater detail in the results section.

**4) Line 81-83, The authors mentioned few studies have considered the simultaneous effects of exposure to both O3 and CO2, so what have learned from these previous studies? Please specify previous findings.**

See response to comment above. We have added a new paragraph to the introduction to discuss previous studies, both field and model-based, and what has been shown from these studies (pg. 4, lines 134 to 157).

**5) Line 86-99, Please describe the O3 concentration for historical and current level in quantity. How does the O3 change over the last decades?**

We have added more information about the change in O3 concentrations from historical to present day (pg.3, lines 88 to 98).

**6) Line 103-104, High levels of O3 are reducing the land carbon sink. How many carbon loss was led to by O3 at regional and global scale based on previous studies?**

We have added a paragraph to the introduction to discuss the findings of previous studies (pg. 4, lines 134 to 157).

**7) Line 121-122, are you also going to study the effects of high temperature and drought?**

This links to an earlier comment – in this study we have used a fixed pre-industrial climate. We cycle over the climate from 1901 to 1920 so we maintain natural climate variability, but we do not have climate change. This allows us to focus on the direct effects of changing atmospheric $CO_2$ and $O_3$ concentrations, and their complex interaction, on plant physiology through the twentieth century and into the future. We realise that we do not make it clear enough early on in the manuscript that we use a fixed climate, so we have amended this in the introduction (pg.6 , lines 214 to 223). We also add a paragraph to our discussion about potential impacts on our results (section 4.3, pg. 26, lines 808 to 816).

**8) Please explain the CUO1 in Figure S2 caption. As shown in Table S1, the g1 parameter in NT (Needle leaf tree) is similar to that of shrub. Does it mean plant water use efficiency in NT and SH are same?**

We have clarified this by adding the following to the Figure S2 caption: "The x axis is cumulative uptake of $O_3$ (CUO) above the critical $O_3$ threshold ($F_{O3crit}$)."

The parameter values for g1 were derived from the extensive database of Lin et al., (2015). The parameter g1 is a measure of water-use efficiency. In the model, the plant water-use efficiencies of NT and SH would be similar but not identical since WUE is the ratio of carbon gain to water loss and the two PFTs have different photosynthetic rates owing to different parameter values.

**9) Figure1, could you provide some O3 concentration data from observations?**

This links to an earlier comment #5. Comparison of these long-term $O_3$ trends with observations is difficult for many reasons, not least lack of reliable data before recent decades, and limited representativity and inconsistencies in data from recent years (Logan et al., 2012;Parrish et al., 2012). For example, ozone levels at the start of the 20th century are estimated to be around 10 ppb for the site

Montsouris Observatory near Paris, data for Arkona on the Baltic coast increased from ca. 15 ppb in
the 1950s to 20-27 ppb by the early 1980s, and the Irish coast site Mace Head shows around 40 ppb
by the year 2000 (Logan et al., 2012, Parrish et al., 2012, and refs within). Trends vary from site to
site though, even on a decadal basis (Logan et al., 2012;Simpson et al., 2014), depending, for
example, on local/regional trends in precursor (especially NOx) emissions, elevation, and exposure to
long-range transport. As a result of this spatial variation in $O_3$ concentrations across Europe,
comparison of the EMEP $O_3$ forcing in Fig. 1 (plotted as a mean across regions) with individual sites
would potentially be misleading.

**10) Line 342-345, what is the uncertainty (or SD) for these percentage number? It may be better
if the authors mentioned how these number are calculated in methods.**

These numbers were re-calculated to get the standard deviation. Previously the annual mean for each
simulation was calculated, and this used to calculate the percentage difference. To get the standard
deviation the daily means were calculated and the percentage difference was calculated for each day,
then the mean and standard deviation were calculated, these values are now reported in the manuscript
(section 3.1, pg. 14, lines 482 to 487). We explain how these numbers are calculated in the SI section
S3 (lines 111 to 114). This re-calculation slightly changes the percentage differences in annual mean
leaf-level stomatal conductance, but the direction of change remains the same, i.e. MED increases
water-use for BT and C3, and reduced water-use for NT, C4 and SH. The standard deviations are
quite large reflecting the large spread in the data, partly due to the seasonal cycle.

**11) Line 352-353, For the broadleaf tree and C3 herbaceous PFT, the Medlyn model simulates a
larger conductance and therefore a greater flux of O3 through stomata compared to Jacobs, but
it also led to a greater flux of CO2 through stomata simultaneously, which may be helpful for
increasing photosynthesis.**

Figures S7 and S8 (top rows) show 1:1 plots of *Anet,* plotting MED (y axis) against JAC (x axis).
These plots show that in the model, *Anet* is not as sensitive to the change in $gs$ scheme as $gs$ itself
(Fig. 2 and Fig. S6). Although the greater conductance for BT and C3 with MED will result in high
internal $CO_2$ concentrations, this doesn't result in a large change in modelled photosynthetic rates
because in the model, the sensitivity of the limiting rates of photosynthesis to changes in $ci$ is much
lower that the sensitivity of $g_s$ to the same change (see section 3.1, pg. 14, lines 488 to 490 where this
is mentioned). Therefore, the WUE for BT and C3 will change, they are less WUE with MED.

**12) Line 366-368, Some Boreal and Mediterranean regions show increased GPP over this
period, associated with O3 induced stomatal closure enhancing water availability. But O3
induced stomatal closure also reduce the flux of CO2 through stomata simultaneously, which
have a negative impact on GPP.**

This is a trade-off between the opposing effects of $O_3$ induced stomatal closure enhancing soil water
availability and also reducing GPP. The overall effect occurs seasonally, which is not shown in Figs.
4, 5, & 6. $O_3$ induced stomatal closure occurs during spring/early summer when $O_3$ concentrations are
highest, at this point GPP is reduced, but in these dry regions this leads to increased soil moisture that,
in the model, allows growth later in the year when conditions are still favourable but soil moisture
may otherwise have been limiting. We have clarified this point in the text (section 3.3, pg. 16, lines
534-538): "Some Boreal and Mediterranean regions show small increases in GPP over this period,
associated with $O_3$ induced stomatal closure enhancing water availability in these drier regions (Fig.
5). This allows for greater stomatal conductance later in the year when soil moisture may otherwise have been limiting to growth (up to 10%, Fig. 6), and therefore higher GPP, but these regions comprise only a small area of the entire domain."

**13) Line 373-375, is the different response of GPP to low and high plant O3 sensitivity are significant?**

On the advice of the reviewer, we carry out statistical testing of the different responses of GPP to the low and high plant $O_3$ sensitivities. We use the software R, and use paired t-tests to determine whether the $O_3$ effect on GPP is significantly different between the two different plant sensitivity parameterisations (section 3.3 line 531; section 3.4 line 570; section 3.5 line 597).

**14) Line 437-440, CO2 induced stomatal closure can help alleviate O3 damage by reducing the uptake of O3, but it will also increase available soil moisture simultaneously.**

We agree. The $CO_2$ induced stomatal closure is the dominant effect that helps alleviate $O_3$ damage. Figures S14 and S15 show that in the model the effect of $CO_2$ on $g_s$ is large, whereas the effect of $CO_2$ on soil moisture availability (fsmc in these plots) is small in comparison. Simulated $g_s$ declines with increasing $CO_2$ which may increase available soil moisture, however $CO_2$ enhances GPP and growth of the vegetation which can increase LAI leading to higher water-use on a leaf area basis. These responses are all captured in our simulation however with both $CO_2$ and $O_3$ changing, and in our calculation of the $O_3$ effect with $CO_2$ rising. We look at the difference in this simulation to the simulation with $O_3$ changing but $CO_2$ concentration fixed at pre-industrial concentrations, this gives us the alleviation of $O_3$ damage by increasing $CO_2$ and all associated effects, such as changes in soil moisture, but it is the effect on stomata that dominates. We have clarified the calculation of the alleviation of $O_3$ damage by increasing $CO_2$ in the legend to Table S6.

**15) Contradictions are reported in Figure 4 and 5. In Figure 4a, the areas with great increasing in plant available soil moisture have less change in gs in Figure 5a. Why? In figure 4c, the areas with decreasing in plant available soil moisture have large reduction in gs.**

The *fsmc* formulation (factor determining plant available soil moisture) varies with soil moisture content and is non-linear. It has a value of zero below wilting point then linearly increases to a value of 1 at field capacity, and remains at this value beyond. The wilting point and field capacity depend on soil texture. Therefore a small percentage change in soil water content in dry regions (Mediterranean) can result in a large percentage increase in *fsmc*. Likewise an increase in soil moisture in mesic areas (e.g. northern Europe) may translate into relatively small percentage changes in *fsmc*.

In Figs 5a and 6a, the areas that see a large increase in plant available soil moisture see a small increase in *gs* (up to 10%). This is looking at the change over the period 1901 to 2001 when only $O_3$ is changing. Over this period there is a large increase in $O_3$, so the $O_3$ induced stomatal closure is large, causing the increase in *fsmc* in this region. The changes are seasonal, these plots show the annual mean which will average out some of the change. In this region, for example, *gs* increases a lot in JJA and DJF, but there is minimal change in SON/MAM. Figures 5c and 6c are for a different time period, 2001 to 2050. Over this period the $O_3$ effect is reduced considerably, so by 2050 plant available water is reduced on 2001 levels because the $O_3$ induced stomatal closure is less. Stomatal conductance decreases in this region during this period.

**16) In table 1, O3 increased GPP but decreased land carbon over the period 2001-2050. Why does land C decrease when GPP is increasing?**

We refer to this in the results section (section 3.4, pg. 17, lines 585 to 589). GPP is a fast flux, whereas the land carbon store is a slower pool of carbon, it takes longer for this carbon store to adjust to changes in the flux, especially when those changes are fairly small as is the case here. This highlights the importance of using a carbon cycle model to look at the impacts of $O_3$ on the terrestrial biosphere.

**The discussion could be improved by using subtitles more clearly.**

We have amended this and added subtitles.

**17) Line 525-541, the authors listed a lot of results from the literatures, but the reader is left to decide what and why is the difference between this study and previous studies? More discussion on comparing this study with previous studies in detail would be helpful.**

This paragraph discusses findings from field-based studies looking at plant $O_3$ impacts. We have removed this paragraph and put it in the introduction as it seemed more appropriate here to place our study in context (pg. 4, lines 134 to 157).

Logan, J. A., Staehelin, J., Megretskaia, I. A., Cammas, J. P., Thouret, V., Claude, H., De Backer, H., Steinbacher, M., Scheel, H. E., Stübi, R., Fröhlich, M., and Derwent, R.: Changes in ozone over Europe: Analysis of ozone measurements from sondes, regular aircraft (MOZAIC) and alpine surface sites, Journal of Geophysical Research, 117, 1-23, 2012.
Parrish, D. D., Law, K. S., Staehelin, J., Derwent, R., Cooper, O. R., Tanimoto, H., Volz-Thomas, A., Gilge, S., Scheel, H. E., Steinbacher, M., and Chan, E.: Long-term changes in lower tropospheric baseline ozone concentrations at northern mid-latitudes, Atmos. Chem. Phys., 12, 11485-11504, 10.5194/acp-12-11485-2012, 2012.
Simpson, D., Arneth, A., Mills, G., Solberg, S., and Uddling, J.: Ozone—the persistent menace: interactions with the N cycle and climate change, Current Opinion in Environmental Sustainability, 9, 9-19, 2014.

**Response to RC2:**

We thank the reviewer for the time taken to read the manuscript and comment on it. The comments
are very helpful and improve the manuscript. We hope we have addressed all the comments to the full
satisfaction of the reviewer. We attach the revised manuscript with track changes so it can be seen
what has been changed and where.
**RC) Oliver et al. quantify the impact of ozone damage to European GPP and total land carbon**
**stock on an annual basis. The authors apply a new stomatal conductance parameterization to**
**their model, and force the model with surface ozone concentrations, meteorology and global**
**CO2 concentration to investigate the roles of CO2 fertilization vs. O3 damage on GPP and total**
**land carbon stock from 1901 to 2050. This new stomatal conductance parameterization**
**simulates higher stomatal conductance than the previous, causing higher uptake of ozone**
**through plant stomata. They find that there are spatial variations in the response of GPP and**
**land carbon stock to CO2 fertilization vs. O3 damage. On a regional basis, CO2 fertilization**
**dominates the response (vs ozone damage) when CO2 is allowed to evolve, but ozone does limit**
**the land carbon sink. The impact of ozone damage from 1901 to 2050 is dominated by 1901-**
**2001 due to increasing surface ozone concentrations during that time. Overall, it seems like**
**there is a lot more to discuss in regards to the previous work that has been done on the leaf level**
**to global scale on this topic (e.g., Karnosky et al., 2003) and how the Oliver et al. findings**
**contribute substantially to knowledge. It is not really clear how these results advance Sitch et al.**
**2007 except examining the region-scale over Europe. A huge limitation to this study is that CO2**
**and meteorology are uncoupled, as well as meteorology, ozone, and stomatal conductance.**
AC: This study makes significant developments to the model from that used in Sitch et al., 2007. In
short these developments include:
- Re-calibration of the model for ozone impacts on vegetation using up-to-date functions published in
2017.
- A representation of ozone damage on crops and accounting for regional differences where possible
(i.e. Mediterranean regions).
- A new $g_s$ model including parameters derived from field observations which have physical meaning
(i.e. measureable quantities).
- A term for non-stomatal deposition of ozone.
- A diurnal cycle of ozone forcing at a much higher spatial resolution than in Sitch *et al*., global
simulations (i.e. 0.5 x0.5 vs 3.75x 2.5) from a high resolution atmospheric chemistry model for
Europe.
The final paragraph of the introduction was re-arranged to highlight these advances (pg. 6, lines 193
to 214).
We also include greater discussion on previous studies in this area and move a paragraph from the
discussion to the introduction (pg. 4, lines 134 to 157).
**RC1) Using recycled early 20th C climate is problematic. I understand that the authors want to**
**isolate the physiological response of plants of CO2 vs. O3 here, but ozone is high during drought**
**and heat waves, and stomata close at that time. So if there is increasing aridity and hydrological**
**and temperature extremes into the 21st C, then the ozone response should be much lower than**
**the authors suggest. The authors do at some point say that their work here is an upper bound,**
**but upper bounds have already been published.**
AC: The aim of these simulations was to investigate the direct effects of changing atmospheric $CO_2$
and $O_3$ concentrations, and their complex interaction, on plant physiology through the twentieth century and into the future. These offline simulations are not coupled and therefore do not have
feedbacks between climate, $O_3$ formation and stomatal behaviour, but nonetheless they are an
important tool to understand the direct impacts of $O_3$ at the land surface. This work demonstrates the
sensitivity of GPP and the land carbon sink to tropospheric $O_3$, highlighting that it is an important
predictor of future GPP and the land carbon sink. We do state in the original manuscript that we use a
fixed climate (methods section 2.4.1 line 373), however, we realise we do not make it clear from the
beginning that we are running offline simulations, therefore we have modified the manuscript to make
this point clear in the introduction (pg.6 , lines 214 to 223).
An important point that we make in the original manuscript is: "our results demonstrate the
sensitivity of modelled terrestrial carbon dynamics to tropospheric $O_3$ and its interaction with
atmospheric $CO_2$, highlighting that such effects of $O_3$ on plant physiology significantly add to the
uncertainty of future trends in the land carbon sink and climate-carbon feedbacks. Given the potential
to limit the climate mitigation effect of European terrestrial ecosystems, we suggest plant $O_3$ damage
should be incorporated into carbon cycle assessments". Here the point we mean to make is that our
work shows the sensitivity of modelled GPP and land carbon to the direct effect of $O_3$ on plant
physiology, however, this process remains largely unconsidered in regional and global climate model
simulations that do account for climate-carbon feedbacks and are used to model carbon sources and
sinks even though it is likely contribute to the large uncertainty in future modelled carbon-climate
feedbacks. We modify the text to make this point more clearly at the end of the conclusions (section
5, pg. 28, lines 879 to 883).
We add to the discussion a paragraph outlining the potential implications for our results of using
uncoupled simulations (section 4.3, pg. 26, lines 808 to 816).
It is computationally expensive to run coupled simulations. Offline studies are valuable in
determining the relevance of individual responses and are relatively cheap computationally. Once the
importance of a process is demonstrated off line, it provides evidence of the need to incorporate such
processes in coupled simulations.
**RC2) Further, Langner et al. 2012 is not the appropriate work here to justify the authors'**
**approach. Langner et al. examine the impact of climate change following the A1B scenario on**
**surface ozone (they do not consider changes in anthropogenic precursor emissions from present**
**to future under A1B). Langner et al. use biogenic emissions to explain some of the cross-model**
**differences in changes from present to future in ozone due to climate. This is quite different**
**from using the full A1B scenario which considers changes in climate & anthropogenic precursor**
**emissions, which is what Oliver et al. do.**
AC: This should be a different Langer et al., 2012 reference here, this has now been corrected:
Langner, J., Engardt, M. & Andersson, C. European summer surface ozone 1990–2100, *Atmos. Chem.*
*Physics,* **2012b***, 12*, 10097-10105
Section 2.4.1, pg. 11, line 399.
**RC3) The authors change their stomatal conductance parameterization but do not explain why.**
**Their phrasing implies that the new gs model is truth, whereas the Jacobs 1994 model is not**
**(e.g., "studies using the Jacobs [1994] formulation may underestimate" on line 523). I**
**understand that parameterizations of stomatal conductance are uncertain in general, and hard**

**to evaluate, but it seems like there should be some reasoning and evaluation here. Further, please clarify the re-calibration (lines 130-133). This seems like a major part of your analysis and I think evaluation & inclusion of this evaluation in the main part of the paper is warranted.**

AC: The main advance of the Medlyn model over Jacobs, and other empirical $gs$ formulations, is the availability of observational-derived parameters for European vegetation. We discuss the advantages of the Medlyn model over the Jacobs formulation in the original text and that is our reasoning for using it in these simulations. We apologise if this is not clear, and have moved this to a separate paragraph in the introduction and expanded our reasoning (pg. 6, lines 181 to 191). We do not mean to imply that the Medlyn model is truth compared to Jacobs, and have changed the wording on line 697 (section 4.1, pg. 26) accordingly to read "studies using the Jacobs $g_s$ formulation would simulate a lower $O_3$ impact for Europe".

We have included site level evaluation of the seasonal cycle of latent and sensible heat at some FLUXNET sites comparing the two $g_s$ models against observations. This is in the supplementary information, section S4 (Fig. S9 and Table S2). We refer to this evaluation in the main text (section 2.3, pg. 10, line 365 and section 3.1, pg. 14, line 497).

We mention the calibration in the introduction, but we do not feel here is the place to expand or clarify further. We expand upon the re-calibration in the Methods (section 2.2), and have updated this section in the manuscript to clarify it further. We put additional details in the supplementary information because these are quite technical details so we feel they are not necessary in the main text.

Validation of land-surface models such as JULES for $O_3$ impacts is not straightforward because of small scale, site specific biotic and abiotic factors that affect the growth response of vegetation to $O_3$. These include competition within and between species leading to differential $O_3$ responses as was seen at the Aspen FACE experiment (King et al., 2005;Karnosky et al., 2007;Kubiske et al., 2007), attack by pests and diseases, nutrient limitation, drought stress. Nevertheless, we now include an evaluation of the $O_3$ model against the flux network model tree ensemble (MTE) product of (Jung et al., 2011). We compare mean GPP from 1991 to 2001 for each of the JULES scenarios and both high and low plant $O_3$ sensitivities against Jung et al., (2011). See methods section 2.4.3, results section 3.2 with new Figure 3, and section S5 in the supplementary information with new figures S10, S11 and S12.

**RC4) Is Jacobs gs used in the ozone dry deposition parameterization that is used in the EMEP model used to project the ozone concentrations? Typically stomatal conductance in the dry deposition parameterizations is some form of Wesely (1989). If Wesely is used, how does the magnitude of Medlyn differs from the magnitude of stomatal conductance from Wesely? If Wesely is used, then CO2 fertilization is not in there, nonetheless ozone damage. Another caveat is that ozone damage can feedback onto ozone concentrations as demonstrated by Sadiq et al. ACP 2017.**

AC: Calculations of $O_3$ deposition in the EMEP model are rather detailed compared to most chemical transport models. We make use of the stomatal conductance algorithm (now commonly referred to as $DO_3SE$) originally presented in Emberson et al. (2000;2001), which depends on temperature, light, humidity and soil moisture. Calculation of non-stomatal sinks, in conjunction with an ecosystem specific calculation of vertical $O_3$ profiles, is an important part of this calculation as discussed in

Tuovinen et al. (2004;2009) or Simpson et al. (2003). The methodology and robustness of the
calculations of $O_3$ deposition and stomatal conductance have been explored in a number of
publications (Emberson et al., 2007;Tuovinen et al., 2004;Tuovinen et al., 2009;Tuovinen et al.,
2007).

Of course, the $gs$ values used in the EMEP model differ from those obtained using a Medlyn
formulation. Comparing EMEP's maximum $gs$ values ($gmax$) with the 95th-100th percentiles of $gs$
found in JULES simulations, we find very similar values for deciduous forest (EMEP 150-200,
JULES ~180, all units in mmole $O_3/m^2$ (PLA)/s), and C3/C4 crops (EMEP 270-300, JULES ~260-
390), but large differences for coniferous forest (EMEP 140-200, JULES ~60-70) and shrubs (EMEP
60-200, JULES 360-390). The role of EMEP in this study is not to provide $gs$, however, but to
provide $O_3$ at the top of the vegetation canopy. The main driver of such $O_3$ levels is the regional-scale
production and transport of ozone, and the main impact of $gs$ is just in affecting the vertical $O_3$
gradients just above the plant canopy. Differences in $gs$ are known to have minimal impact on
canopy-top $O_3$ for trees, mainly due to the efficient turbulent mixing above tall canopies, but also due
to non-stomatal sink processes. For shorter vegetation, substantial $O_3$ gradients, driven by deposition,
occur in the lowest 10s of metres of the atmosphere, and stomatal sinks (as given by $gs$) can have a
significant role. However, calculations of such gradients made with the EMEP model for CLRTAP
(2017) showed that such differences amounted to ca. 10% when comparing $O_3$ concentrations at 1m
height above high-gs crops ($gmax$=450 mmole $O_3/m^2$ (PLA)/s) species compared to moderate-$gs$
($gmax$ 270 mmole $O_3/m^2$ (PLA)/s).

These inconsistencies are not ideal, but inevitable given that we link two different model systems.
There are of course many uncertainties in all estimates of deposition and stomatal ozone flux (e.g.
Tuovinen et al., 2009), and we believe that this particular uncertainty is an acceptable part of our
procedure.

The referee's comments about $CO_2$ and the impacts mentioned by Sadiq are also relevant, but again
there are many uncertainties associated with such effects and assessments too.

In order to keep a concise text, but mention the above points, we have added a summary of the above
points to the manuscript in the discussion section 4.3, pg. 25, lines 790 to 806.

**RC5) A large part of the results hinge on the seasonality of surface ozone concentrations, and**
**how they change from PI to present. There is some discussion of this on pages 401-405 as**
**authors examine the change in seasonality from 2001 to 2050, but there is no citation of previous**
**work examining changes in ozone seasonality, or the implications of this for their conclusions.**
**Also, the authors say that tropospheric ozone is increasing (e.g., line 74), but I think this is a bit**
**misleading - due to strong changes in seasonality that are observed - please revise.**

AC: We have added a paragraph to the manuscript to acknowledge and discuss the importance of the
seasonality of surface ozone concentrations, citing previous work examining these changes, and the
implications of this for our results (section 2.1.4, pg. 12, lines 420 to 442). Line 74 has been revised
(now line 75).

**RC6) In general, the paper is a bit poorly organized. Many times the authors say "see details in**
**SI" when it's not clear what information is in there, and why it is relevant. Further it seems like**
**some info in the SI should really be in the actual paper. In addition, the authors neglect to**

**mention many substantial caveats (such as the uncertainties around CO2 fertilization w.r.t. nutrient cycling, using uncoupled tropospheric chemistry & stomatal dry deposition, and stomatal sluggishness) until the very end. I think the paper would be much better if much of the discussion was moved to the introduction and used to frame the work, and motivate the authors' objectives.**

AC: We apologise for the lack of clarity when referring to the supplementary information, we have amended this to make clear what section in the SI we refer to and why. In response to referee requests we have revamped the introduction to clarify the specific focus of the manuscript (i.e. carbon cycle impact of the plant physiological response to $O_3$ and $CO_2$), and therefore make it easier to understand what is and what is not included.

We discuss the caveats of the study at length in the original manuscript. These are very important, so we are sure to make clear that we are fully aware of the caveats. We also now include an additional paragraph in the discussion section 4.3 on the potential implications of uncoupled tropospheric chemistry and stomatal dry deposition for our results which was previously missing. We also introduce the issue of sluggish stomata and $CO_2$ fertilization in the introduction to help frame the study. However, on the whole we think that discussion of the caveats is more appropriate in the discussion.

**Minor comments**
**RC1. The authors use the term "significant" a lot - but don't do any sort of statistical testing. Please only use the word significant when describing results that are statistically significant.**

AC: We have revised our use of significant where appropriate.

**RC2. Line 78: Lightning is a source of NOx, not O3 – please revise**

AC: This has been amended to read "…. and lightning which is a source of $NO_x$". (line 79)

**RC3. Lines 86-87: Parrish et al. 2012 is not really the appropriate citation here**

AC: We have changed this reference for Vingarzan (2004). (line 88)

**RC4. Lines 93-94: "Intercontinental transport" doesn't mean that background ozone has increased, there has always been intercontinental transport.**

 AC: This sentence has been changed to: "Intercontinental transport of air pollution from regions such as Asia that currently have poor emission controls are thought to contribute largely to rising background $O_3$ concentrations in Europe over the last decades (Cooper et al., 2010;Verstraeten et al., 2015)." (line 103)

**RC5. Line 101: Citations for ozone impacts on crop yields and nutritional quality are needed**

AC: We have added the following references: Ainsworth et al., (2010) and Avnery et al., (2011). (line 114)

**RC6. Line 106: Do the authors mean indirect here?**

AC: We mean direct – ozone has a direct effect on radiative forcing of the climate. The indirect effect is ozone damage of vegetation which reduces uptake of carbon by plant photosynthesis, allowing more $CO_2$ to remain in the atmosphere. (line 118)

**RC7. Line 110: Fowler et al. 2009 isn't really the appropriate citation here - i.e., for saying that dry deposition is a substantial sink of tropospheric ozone**

AC: We have added an additional reference: Fowler et al., (2001). (line 123)

**RC8. Line 152 - as the authors mention in the discussion, ozone can directly impact gs - please revise accordingly**

AC: This has been amended at line 195.

**RC9. Line 160-168: please clarify the spatial domain and the resolution of this model; also, is the resolution the same as the meteorological and ozone forcing files?**

AC: We added the following sentence to clarify the resolution of the model (line 247): "This work uses JULES version 3.3 (http://www.jchmr.org) at $0.5°$ x $0.5°$ spatial resolution and hourly model time step, the spatial domain is shown in Fig. S5." We also explicitly state the resolution of the all the forcing data (meteorology, $CO_2$, ozone and land cover) to show that they are all the same $0.5°$ x $0.5°$ resolution.

**RC10. Lines 193-194: kappa_O3 is not exactly the ratio of the resistances; it's the ratio of the diffusivities**

AC: This has been changed to : "$K_{o3}$ accounts for the different diffusivity of ozone to water vapour and takes a value of 1.51 after Massman (1998)" (line 280).

**RC11. Lines 220-222: What is CLRTAP (2017)? It is not in the references. Why is it being treated as the "truth"?**

AC: The reference for CLRTAP (2017) is now in the reference list. It is a report on ozone impacts on vegetation, providing a synthesis of the latest peer reviewed literature, collated by a panel of experts and so is considered the state-of the art knowledge. It provides the $O_3$ dose response functions compiled from numerous field studies that we use to calibrate our model PFTs for sensitivity to $O_3$. We have expanded section 2.2 which explains this more clearly.

**RC12. Section 2.3 - please clarify that Lin et al. 2015 fit g1 parameters based on the Medlyn et al. 2011 equation for stomatal conductance (except no g0 term), which is not exactly the same as putting equation 7 into equation 5; it's confusing to refer to this equation as Medlyn et al. (2011); also, I do not think that multiplying the Anet/(Ca-Ci) by R*T is the right way to convert from mol s-1 m-2 to m/s.**

AC: We clarify this in the following sentence (line 352): "The $g_1$ parameter represents the sensitivity of $g_s$ to the assimilation rate, i.e. plant water use efficiency, and was derived as in Lin et al. (2015) by fitting the Medlyn *et al*., (2011) model to observations of $g_s$, photosynthesis, and VPD, with no $g_0$ term." At line 346 we also say "In this work, we replace equation 6 with the closure described in Medlyn et al. (2011), ….". and then refer to it from then on as the MED model instead of the Medlyn et al (2011) model.

**RC13. Lines 252-254: Please clarify "the effect" that Hoshika et al. 2013 find; does O3 increase or decrease WUE? This seems relevant to your discussion/conclusions.**

AC: We clarify by adding the following (line 355): "Hoshika et al., (2013) show a significant difference in the g1 parameter (higher in elevated $O_3$ compared to ambient) in Siebold's beech in June of their experiment. However, this is only at the start of the growing season, further measurements show no difference in this parameter between $O_3$ treatments."

**RC14. Lines 300-301: Clarify the "disaggregation" of ozone from the daily mean to the hourly time step. As ozone has a diurnal cycle, and stomatal conductance does as well, this could have a substantial impact on your work, and should be discussed.**

AC: We have added the following sentence to clarify the disaggregation (line 408): "The daily mean $O_3$ forcing was disaggregated to follow a mean diurnal profile of $O_3$, this was generated from hourly $O_3$ output from EMEP MSC-W for the two land cover categories across the same domain as in this study."

**RC15. Lines 305-306: Clarify the calculation of the ozone gradient from the lowest atmosphere grid box to canopy height**

AC: The ozone forcing used in this study was produced by the EMEP MSC-W model, here we provide a reference to the model documentation (Simpson et al., 2012) so readers can follow up further details. It is beyond the scope of this study to document how EMEP MSC-W works.

**RC16. Further details on crops in JULES should be included in Section 2.4.1 in addition to the discussion.**

AC: We have amended this to the following: "The agricultural mask means that only $C_3/C_4$ herbaceous PFTs are allowed to grow, with no competition from other PFTs, no form of land management is simulated." We discuss the limitations of this in the discussion (lines 761).

**RC17. Lines 282-283: Please specify the ozone sensitivity used for forests**

AC: This has been removed as it is now explained in more detail in section 2.2.

**RC18. Line 882: I don't think "in prep" studies can be cited.**

AC: This has been removed.

**RC19. Lines 258-259:What are the two model grid points? What does wet vs. dry refer to? This info is used later on in the paper (Figure 2), so it would be helpful for more information on this.**

AC: More information to clarify this is provided in the SI section S3, but this was probably not clear because we did not make it clear which section in the SI to refer to. We have rectified this, and now state "see SI section S3 for further details" (line 364).

**RC20. Please clarify in the Figure 2 caption what exactly the readers are looking at (this is just one grid cell, with each sub-tile PFT gs shown?). Why just one grid-cell? Is the data shown hourly? What is the time period?**

AC: Shown are hourly values for the year 2000, from a single grid cell fixed to have 20% land cover of each PFT – therefore we are comparing $g_s$ for each PFT under the same conditions. This information is in the SI section S3 which we hope will be clearer now as we refer to it appropriately earlier on in the manuscript.

The figure caption has been amended to: "**Figure 2.** Comparison of simulated $g_s$ with MED (y axis) versus JAC (x axis) for all five JULES PFTs at one grid point (lat: 48.25; lon:, 5.25) shown are hourly values for the year 2000 (see SI section S3 for further details). Shown are stomatal conductance ($g_s$, top row), and the flux of $O_3$ through the stomata (flux_o3, bottom row)."

**RC21. Lines 384-396: It's not clear why the authors are examining different decades for their analysis here. Second, it seems like the authors could pretty easily sample their model for an apples-to-apples comparison with Boden et al. 2013. Third, suggesting that the O3 impact on the land carbon sink is a source of carbon is not really appropriate (lines 395-396); re-phrasing would allow**
**for the same take-away**

AC: We analyse different decades because it shows how the $O_3$ effect has changed through time. The Boden et al data is available on a country by country basis without lat/lon information for the spatial extent of coverage. Therefore it is best to stick to our domain for comparison, but clearly acknowledge that our domain is slightly larger in extent.

**RC22. Lines 401-402: Ozone precursor emission controls do not always lead to ozone reductions because formation chemistry is nonlinear; please revise.**

AC: We have removed this sentence.

**RC23. Line 401: Large spatial variability is not apparent to me - it would be helpful if the authors were more specific.**

AC: To my eye the spatial variation is apparent in Fig. 4g & h. Nevertheless, we do describe this variation in more detail in the results section.

**RC24. Lines 405-408: it's not clear what figure the authors are talking about here.**

AC: This is Fig. 4g & h. This has been updated in the text.

**RC25. Figure 6 - specify whether your numbers correspond to rows or columns.**

AC: They refer to columns. We have amended the legend for figure 7 to make this clearer.

Refs:

CLRTAP: The UNECE Convention on Long-range Transboundary Air Pollution. Manual on Methodologies and Criteria for Modelling and Mapping Critical Loads and Levels and Air Pollution Effects, Risks and Trends: Chapter III Mapping Critical Levels for Vegetation, accessed via, http://icpvegetation.ceh.ac.uk/publications/documents/Chapter3-Mappingcriticallevelsforvegetation_000.pdf, 2017.
Cooper, O. R., Parrish, D. D., Stohl, A., Trainer, M., Nedelec, P., Thouret, V., Cammas, J. P., Oltmans, S. J., Johnson, B. J., Tarasick, D., Leblanc, T., McDermid, I. S., Jaffe, D., Gao, R., Stith, J., Ryerson, T., Aikin, K., Campos, T., Weinheimer, A., and Avery, M. A.: Increasing springtime ozone mixing ratios in the free troposphere over western North America, Nature, 463, 344-348, http://www.nature.com/nature/journal/v463/n7279/suppinfo/nature08708_S1.html, 2010.

Emberson, L. D., Ashmore, M. R., Cambridge, H. M., Simpson, D., and Tuovinen, J.-P.: Modelling stomatal ozone flux across Europe, Environmental Pollution, 109, 403–413, 2000.

Emberson, L. D., Simpson, D., Tuovinen, J.-P., Ashmore, M. R., and Cambridge, H. M.: Modelling and mapping ozone deposition in Europe, Water Air Soil Pollution, 130, 577–582, 2001.

Emberson, L. D., Büker, P., and Ashmore, M. R.: Assessing the risk caused by ground level ozone to European forest trees: A case study in pine, beech and oak across different climate regions, Environmental Pollution, 147, 454–466, 2007.

Jung, M., Reichstein, M., Margolis, H. A., Cescatti, A., Richardson, A. D., Arain, M. A., Arneth, A., Bernhofer, C., Bonal, D., Chen, J., Gianelle, D., Gobron, N., Kiely, G., Kutsch, W., Lasslop, G., Law, B. E., Lindroth, A., Merbold, L., Montagnani, L., Moors, E. J., Papale, D., Sottocornola, M., Vaccari, F., and Williams, C.: Global patterns of land-atmosphere fluxes of carbon dioxide, latent heat, and sensible heat derived from eddy covariance, satellite, and meteorological observations, Journal of Geophysical Research: Biogeosciences, 116, n/a-n/a, 10.1029/2010JG001566, 2011.

Karnosky, D. F., Skelly, J. M., Percy, K. E., and Chappelka, A. H.: Perspectives regarding 50years of research on effects of tropospheric ozone air pollution on US forests, Environmental Pollution, 147, 489-506, 2007.

King, J. S., Kubiske, M. E., Pregitzer, K. S., Hendrey, G. R., McDonald, E. P., Giardina, C. P., Quinn, V. S., and Karnosky, D. F.: Tropospheric $O_3$ compromises net primary production in young stands of trembling aspen, paper birch and sugar maple in response to elevated atmospheric $CO_2$., New Phytologist, 168, 623-635, 2005.

Kubiske, M., Quinn, V., Marquardt, P., and Karnosky, D.: Effects of Elevated Atmospheric CO2 and/or O3 on Intra-and Interspecific Competitive Ability of Aspen, Plant biology, 9, 342-355, 2007.

Lin, Y.-S., Medlyn, B. E., Duursma, R. A., Prentice, I. C., Wang, H., Baig, S., Eamus, D., de Dios, V. R., Mitchell, P., and Ellsworth, D. S.: Optimal stomatal behaviour around the world, Nature Climate Change, 5, 459-464, 2015.

Massman, W. J.: A review of the molecular diffusivities of H2O, CO2, CH4, CO, O3, SO2, NH3, N2O, NO, and NO2 in air, O2 and N2 near STP, Atmospheric Environment, 32, 1111-1127, http://dx.doi.org/10.1016/S1352-2310(97)00391-9, 1998.

Medlyn, B. E., Duursma, R. A., Eamus, D., Ellsworth, D. S., Prentice, I. C., Barton, C. V., Crous, K. Y., de Angelis, P., Freeman, M., and Wingate, L.: Reconciling the optimal and empirical approaches to modelling stomatal conductance, Global Change Biology, 17, 2134-2144, 2011.

Simpson, D., Tuovinen, J.-P., Emberson, L., and Ashmore, M.: Characteristics of an ozone deposition module II: Sensitivity analysis, Water Air Soil Pollution, 143, 123–137, 2003.

Tuovinen, J.-P., Ashmore, M., Emberson, L., and Simpson, D.: Testing and improving the EMEP ozone deposition module, Atmospheric Environment, 38, 2373–2385, 2004.

Tuovinen, J.-P., Simpson, D., Emberson, L., Ashmore, M., and Gerosa, G.: Robustness of modelled ozone exposures and doses, Environmental Pollution, 146, 578–586, 2007.

Tuovinen, J.-P., Emberson, L., and Simpson, D.: Modelling ozone fluxes to forests for risk assessment: status and prospects, Annals of Forest Science, 66, 1-14, 2009.

Verstraeten, W. W., Neu, J. L., Williams, J. E., Bowman, K. W., Worden, J. R., and Boersma, K. F.: Rapid increases in tropospheric ozone production and export from China, Nature Geoscience 8, 690-695, 2015.

---

## Referee Report (RR1)

Thanks to the authors for moving the discussion of the calibration to the main text. I think this strengthens the paper, and clarifies the novelty of the work. I think the work should be published after the authors address minor revisions.

General comments.
I don't find the naming of the different simulations to O3, CO2, and CO2+O3 very helpful especially when the authors are referring to the gases as O3 and CO2 at the same time. Something like run_O3, or run_CO2, run_both_O3_CO2?

I'm a bit confused with respect to the title of the paper, especially because the authors have a section dedicated to their findings that the impact isn't as large as expected.

Line-by-line comments.
Lines 39: "its interaction with CO2" does not sit well with me because CO2 and O3 are not directly interacting — rather there are interactive effects of CO2 and O3. Please revise.
Lines 44-46: At this point in the abstract the readers don't know what high and low sensitivity are
Line 62: citation should be parenthetical
Line 86: authors should define what they mean by background ozone
Line 94: "long-range transport of ozone"
Line 100: why are there two "e.g."s? What is Sicard et al. (2013) a reference here for?
Line 103: I would suggest cutting "currently have poor emission controls"
Line 108-109: Is there a lot of transport of ozone into Europe? I think references are needed here, or this line should be cut.
Line 112-113: Having this statement here could be misleading — for example, if a reader thought the impact of ozone on vegetation was through it's ability to trap heat. It would be more clear if after "direct radiative forcing of ozone", the authors added ", a potent greenhouse gas, "
Lines 112-114: I find this a bit confusing - the authors discuss "high levels" and "elevated" ozone, but in the paragraph prior discuss mostly background concentrations. Can the authors make the transition a little smoother?
Lines 113: Please cut "future concentrations of ozone predicted for 2050"
Line 139: Here the authors refer to 46 ppb as "elevated" - in the previous sentences much higher concentrations are used. I would recommend just giving the concentrations, not qualifying them here and elsewhere
Line 153: What's the time frame for Lombardozzi?
Line 194: please define ozone dose-response relationship
Line 220: I thought the authors re-arranged the supplementary figures to reflect the order they are mentioned in the text? This is the first occurrence of a supplementary figure and it's figure S5.
Line 227: of "stomatal ozone deposition"
Line 231-233: "because the impact of cumulative ozone exposure on plant productivity has already been calibrated with observations (described below)"
Line 240: Can this be changed to interactive effects of CO2 and O3?
Line 225: Reference is missing the year - here and in line 332 - also I'm not sure why there is a comma before the parenthetical citation
Line 288: But how is the actually PODy/ FO3crit determined? Please specify here
Line 298: Is "a" the ozone plant sensitivity? Please specify here
Line 302: It's unclear to me how the authors incorporate the work of Büker et al. from this paragraph.
Line 380: The g0 term has not previously been defined. I would just say a version of the Medlyn (2011) model that does not have an intercept
Line 416: "of" is missing. Can interaction be changed to interactive effects?
Line 431: phenology is misspelled

Lines 460-474: The point I wanted clarified here is that the agricultural mask does not change from 1900 to 2100

Lines 517 & 519: I don't think an equation for the percentage change is necessary.

Figure S5 - Tg of N in b)? Tg Carbon in c)?, Gg of carbon in d)? Please specify. Are the NOx numbers for anthropogenic sources or anthropogenic and natural? What about NMVOCs?

Table 1 - One of the "O3"s actually reads "O2"

Line 829: I would not say there are large improvements

Lines 841-855: A discussion of how incorporating ozone damage into JULES leads to a worse agreement with the MTE product & relevance for the authors' work is needed here

Line 860-1: Can the authors give the time frame here? Does the range represent high vs. low sensitivity? Is this for O3 +CO2?

Line 870: "Simulated ozone impacts will dependent on model ozone concentrations, meteorology, plant sensitivity to ozone, and process representation of ozone damage"

Lines 857-876: This section title is a bit of an oversell if the authors can't explain the differences. How can the authors "expect" results if they end of concluding the studies are so different anyway?

Line 907: It seems like here the authors need to discuss the caveat that both high and low sensitivity simulations underpredict GPP

Lines 909-910: "may dampen"

Line 941: Can this be changed to interactive effects of O3 and CO2?

Line 884-886: please cut this discussion - the authors' point is made - the information is lacking.

Line 888-908: The authors need to spell out the transition at the beginning of this paragraph — i.e., that another caveat of their study is that ozone is offline and the depositional sink is different here and in the model that was used to create forcing dataset. The comparison of the two gmax values is an apples-to-oranges comparison (one is model input, one is model output!) and I think it should be cut. I find the rest of the discussion not appropriate here - it's is already discussed in the methods. Please clearly state the caveat and cut most of this discussion (i.e., after Lines 891-908).

Line 933-934: Why are the future tropospheric ozone concentrations highly uncertain?

---

## Author Response (AR2)

**Suggestions for revision or reasons for rejection (will be published if the paper is accepted for final publication)**

I find the manuscript improved in the second round of submission. However, there are still major revisions necessary. But the only new analysis that I suggest is a supplementary table or figure showing changes in major ozone precursor emissions throughout the time period analyzed. The other major revisions I suggest involve word-choice, organization, and clarity of the manuscript, which I think need substantial improvement before publication. I would like to see more discussion of the calibration of the ozone damage parameterization discussed and model evaluation of the high vs. low ozone sensitivity simulations in the main text (especially in regards to the implications of the model evaluation for the investigation here), as they are central to the novelty of the study and thus its findings. I also think the authors need to describe supplemental material and findings in the main text; not only refer the readers to them without context.

We would like to thank the reviewer for their detailed reading of the manuscript and suggestions for edits to improve the manuscript. We hope that the reviewer will find the manuscript much improved again. We have provided the additional figure in the SI (figure S5) showing the trend in precursor emissions of $CH_4$, $NO_X$, NMVOC and Isoprene from 1900 to 2050 over Europe. The remaining revisions were largely to do with word-choice, organisation and clarity, and we hope we have sufficiently improved this. We have moved a lot of information from the SI to the main text to help with clarity, and to ensure some of the key points, such as the model evaluation, are discussed in the main text instead of the SI. We address the comments line by line below.

To help readers, I encourage the authors to name their sensitivity simulations and refer to their sensitivity simulations by these names, as opposed to saying things like "the CO2 only run", or "O3 and CO2 simulations", "varying CO2 and O3 together", as these phrases are rather ambiguous.

We have hopefully clarified this. We call our simulations O3, CO2 and CO2+O3 and we now refer to our simulations using these names.

Line by line comments:

Line 31: I disagree that the "impact of the gas on European vegetation and the land carbon sink is largely unknown" - the authors show in their discussion of the literature that there is a substantial amount of work on this. I urge the authors to motivate their work in a way that complements the previous work.

We have removed this sentence and replaced it with the following to clarify the motivation behind our work and its novelty (Lines 33 to 40):

"Studying the impact of $O_3$ on European vegetation at the regional scale is important for gaining greater understanding of the impact of $O_3$ on the land carbon sink at large spatial scales. In this work we take a regional approach and update the JULES land-surface model using new measurements specifically for European vegetation. Given the importance of stomatal conductance in determining the flux of $O_3$ into plants, we implement an alternative stomatal closure parameterization and account for diurnal variations in $O_3$ concentration in our simulations. We conduct our analysis specifically for the European region to quantify the impact of tropospheric $O_3$, and its interaction with $CO_2$, on gross primary productivity (GPP) and land carbon storage across Europe."

Line 35: I don't think the authors can call their new stomatal conductance parameterization "an improved" one. I would suggest finding another way to describe it.

We have changed this to describe it as the following (Line 37):

"an alternative stomatal closure parameterization".

Line 41: Where is this discussed in the paper?

We have removed this sentence.

Line 82: Please cite papers showing that ozone damage is "key" take into account

We have changed key to important, and have added two references here (Line 83/84):

(Le Quéré et al., 2016;Sitch et al., 2015).

Line 98-101: Please clarify the relevance of this text to the analysis

The text in this paragraph is discussing the observed changes in $O_3$ concentration through the 20th century. The text in these two lines is discussing future $O_3$ concentrations - they will depend on emissions of $O_3$ precursors, of which intercontinental transport is an important factor, and climate change which may increase the occurrence of peak $O_3$ episodes, and the emission of $O_3$ precursors isoprene and $NO_X$.

Lines 107-108: The authors should revise their phrasing here.

This has been amended to read (Line 108):

" Intercontinental transport means future $O_3$ concentrations in Europe will be partly dependent on how $O_3$ precursor emissions evolve globally."

Line 118: A paper that examines the ozone budget and source/sink terms would be more appropriate to reference here. Fowler et al. (2009) can be the citation for "primarily uptake by … vegetation", but these papers don't show that it's an important sink. Admittedly it's hard to find papers with the surface ozone budget, but there are many papers that examine the parts of the tropospheric ozone budget.

We moved the Fowler et al., references and added the Young et al., 2013 and Wild 2007 reference (Line 120 - 121):

"Dry deposition of $O_3$ to terrestrial surfaces, primarily uptake by stomata on plant foliage and deposition on external surfaces of vegetation (Fowler et al., 2001;Fowler et al., 2009), is a large sink for ground level $O_3$ (Wild, 2007; Young et al., 2013)."

Line 131-132: Instead of saying "future concentrations of ozone predicted for 2050", the authors should give the concentration, as these numbers are highly dependent on emission scenario and not necessarily generalizable to a given year.

This paragraph has been updated to add specific $O_3$ concentrations (Line 131 - 148):

"The response of plants to $O_3$ is very wide ranging as reported in the literature from different field studies. The Wittig et al. (2007) meta-analysis of temperate and boreal tree species showed future concentrations of $O_3$ predicted for 2050 significantly reduced leaf level light saturated net photosynthetic uptake (-19%, range: -3% to -28% at a mean $O_3$ concentration of 85 ppb) and $g_s$ (-10%, range: +5% to -23% at a mean $O_3$ concentration of 91 ppb) in both broadleaf and needle leaf tree species. In the Feng et al. (2008) meta-analysis of wheat, projected $O_3$ concentrations for the future reduced aboveground biomass (-18% at a mean $O_3$ concentration of 70 ppb) photosynthetic rate (-20% at a mean $O_3$ concentration of 73 ppb) and $g_s$ (-22% at a mean $O_3$ concentration of 79 ppb). One of few long-term field based $O_3$ exposure studies (AspenFACE) showed that after 11 years of exposing mature trees to elevated $O_3$ concentrations (mean $O_3$ concentration of 46 ppb), $O_3$ decreased ecosystem carbon content (-9%), and decreased NPP (-10%), although the $O_3$ effect decreased through time (Talhelm et al., 2014). Zak et al. (2011) showed this was partly due to a shift in community structure as $O_3$-tolerant species, competitively inferior in low $O_3$ environments, out competed $O_3$-sensitivie species. GPP was reduced (-12% to -19%) at two Mediterranean ecosystems exposed to high ambient $O_3$ concentrations (ranging between 20 to 72 ppb across sites and through the year) studied by Fares et al. (2013). Biomass of mature beech trees was reduced (-44%) after 8 years of exposure to elevated $O_3$ (~150 ppb) (Matyssek et al., 2010a). After 5 years of $O_3$ exposure (ambient +20 to +40 ppb) in a semi-natural grassland, annual biomass production was reduced (-23%), and in a Mediterranean annual pasture $O_3$ exposure significantly reduced total aboveground biomass (up to -25%) (Calvete-Sogo et al., 2014)."

Line 150: What are the "new measurements"? The g1 parameter? g1 can't be measured, only inferred. Please revise.

Here we are referring to the data we use to calibrate JULES for plant sensitivity to $O_3$ uptake. This wasn't clear in the text so we have revised accordingly (Line 154 to 157):

"Here we take a regional approach and take advantage of the latest measurements showing changes in plant productivity with exposure to $O_3$ specifically for a range of European vegetation from different regions (CLRTAP 2017) with which to calibrate the JULES model for plant sensitivity to $O_3$, and conduct a dedicated analysis for the European region."

Line 151: "conduct a dedicated analysis" has little meaning. Please revise.

This has been changed to (Line 157):

"and conduct our analysis specifically for the European region."

Line 157-158: I would cut everything in this sentence starting with "such that" because I think that it implies independent responses.

This has been removed.

Line 169-170: Here the phrase about not including stomatal sluggishness is a bit awkward.

This phrase has been removed (Line 173):

"This model is based on the optimal theory of stomatal behaviour and has the following advantages over the current JULES $g_s$ formulation of Jacobs (1994):....."

Line 167-176: This is a rather technical paragraph for the introduction. I wonder if the authors could illustrate the novelty of the study without as much jargon.

We have revised this paragraph to remove the jargon (Line 172 - 178):

"Given the critical role $g_s$ plays in the uptake of both $CO_2$ and $O_3$, we use an alternative representation and parameterisation of $g_s$ in JULES by implementing the Medlyn *et al*. (2011) $g_s$ formulation. This model is based on the optimal theory of stomatal behaviour and has advantages over the current JULES $g_s$ formulation of Jacobs (1994) including i) a single parameter ($g_1$) compared to two parameters in Jacobs (1994), ii) the $g_1$ parameter is related to the water-use strategy of vegetation and is easier to parameterise with commonly measured leaf or canopy level observations of photosynthesis, $g_s$ and humidity, and (iii) values of $g_1$ are available for many different plant functional types (PFTs) derived from a global data set of leaf-level measurements (Lin et al., 2015)."

Line 183: "look at the interaction between O3 and CO2" is ambiguous. Same thing for Lines 196-197, and other points in the text. Please revise.

This has been revised to the following (Line 185):

"......to investigate the impact of both $O_3$ and $CO_2$ on plant water-use and carbon uptake."

Line 183-185: As discussed below, I don't think this is a reason for why this study is novel, and urge the authors to cut this from the introduction.

This has been removed.

Line 189: I'm not certain how the high and low ozone sensitivity simulations represent the large variation within and between species specifically, rather than just the large uncertainty in the ozone response generally. Please clarify that for both the high and low sensitivity simulations, there is a distinction between the sensitivities of crops vs. grasslands. What about forests?

This paragraph has been clarified (Line 186 - 193):

"In this work, the JULES model is re-calibrated using the latest observations of vegetation sensitivity to $O_3$, with the addition of a separate parameterisation for temperate/boreal regions versus the Mediterranean. The $O_3$ sensitivity of each PFT in JULES was re-calibrated for both a high and low sensitivity to account for uncertainty in the $O_3$ response, in part due to the observed variation in $O_3$ sensitivity between species. This includes $O_3$ sensitivities for agricultural crops (wheat – high sensitivity) versus natural grassland (low sensitivity), with separate sensitivities for Mediterranean grasslands. For forests JULES is parameterised with $O_3$ sensitivities for broadleaf and needle leaf trees (both high and low $O_3$ sensitivity), with separate sensitivities (high and low) for Mediterranean broadleaf species."

Line 193-6: Please clarify here that the authors are forcing with daily ozone concentrations that are scaled to a diurnal cycle. The authors' phrasing implies that hourly concentrations are archived and used to force the model.

The phrasing has been amended here (Line 196 - 197):

"JULES is forced with spatially varying daily $O_3$ concentrations from a high resolution atmospheric chemistry model for Europe that are disaggregated to hourly concentrations,……"

Line 199-201: The authors should also note here that not using coupled chemistry and climate also creates additional uncertainty.

A sentence has been added (Line 204 -205):

"In addition, using uncoupled chemistry and climate is a further source of uncertainty."

Line 216: The order of the supplemental figures should reflect the order that they are mentioned in the main text

This has been changed.

Line 226: Lombardozzi and colleagues's work shows that there are separate impacts of ozone on stomatal vs. photosynthesis. This merits mention, as this is how ozone damage is configured in one of the only other land surface models with ozone damage. Ozone damage is also a function of cumulative ozone exposure, rather than an instantaneous effect. Please comment on this.

We have amended this paragraph as follows to mention other models that include $O_3$ damage and discuss $O_3$ damage as a function of cumulative $O_3$ exposure (Line 227 - 240):

"To simulate the effects of $O_3$ deposition on vegetation productivity and water use, JULES uses the flux-gradient approach of Sitch *et al.,* (2007), modified to include non-stomatal deposition following Tuovinen et al. (2009). A similar approach is taken by Franz et al. (2017) in the OCN model, however plant $O_3$ damage is a function of accumulated $O_3$ exposure over time. In JULES, plant $O_3$ damage is instantaneous, the degree to which photosynthesis and $g_s$ are modified at each time step with $O_3$ exposure having already been calibrated against observations of the change in plant productivity with cumulative $O_3$ exposure for each PFT (i.e. $O_3$ dose-response functions described later). JULES uses a coupled model of $g_s$ and photosynthesis, the potential net photosynthetic rate ($A_p$, mol $CO_2$ $m^{-2}$ $s^{-1}$) is modified by an 'O3 uptake' factor (*F,* the fractional reduction in photosynthesis), so that the actual net photosynthesis ($A_{net}$, mol $CO_2$ $m^{-2}$ $s^{-1}$) is given by equation 1 (Clark *et al.,* 2011, Sitch *et al.,* 2007). Because of the relationship between these two fluxes, the direct effect of $O_3$ damage on photosynthetic rate also leads to a reduction in $g_s$. An alternative approach was taken by Lombardozzi et al. (2012) in the CLM model where photosynthesis and $g_s$ are decoupled, so that $O_3$ exposure affects carbon assimilation and transpiration independently. In JULES, changes in atmospheric $CO_2$ concentration also affect photosynthetic rate and $g_s$, consequently the interaction between changing concentrations of both $CO_2$ and $O_3$ is allowed for."

Line 247-249: Nonstomatal conductances are highly variable and substantially larger than this single prescribed value on average across sites. Including this term as 0.04 cm/s should only decrease stomatal uptake by a very small amount. I disagree that this adds any value to the authors' study.

Non-stomatal resistances are highly variable and uncertain. This was discussed extensively for the original EMEP formulation in Emberson et al. (2000), where the choice of $g_{ext}$ = 0.04 cm/s was explained, and EMEP

has so far retained the same choice due to the uncertainties of alternative formulations (e.g. Tuovinen et al., 2009). As a first approach in JULES, for this work we followed the same approach to add a term for non-stomatal deposition of $O_3$. We appreciate there are more complex processes involved, and this is an area for development within JULES. However, for this work it represents a significant step on from previous studies with JULES where non-stomatal conductance wasn't considered, and as such it requires mentioning here. The $O_3$ forcing we use is at canopy height from the EMEP model which includes many of the complex processes in the resistance network already.

Emberson, L., Simpson, D., Tuovinen, J.-P., Ashmore, M. & Cambridge, H. Towards a model of ozone deposition and stomatal uptake over Europe The Norwegian Meteorological Institute, Oslo, Norway, The Norwegian Meteorological Institute, Oslo, Norway, 2000

Tuovinen, J.-P., Emberson, L., and Simpson, D.: Modelling ozone fluxes to forests for risk assessment: status and prospects, Annals of Forest Science, 66, 1-14, 2009.

Line 253: How are leaf dimensions defined?

Leaf dimensions are defined per PFT, this has been added to the text as follows (Line 263 - 264):

"$Ld$ is the cross-wind leaf dimension (m) defined per PFT as 0.05 for trees, 0.02 for grasses ($C_3$ and $C_4$) and 0.04 for shrubs,"

Line 262: What about the resistance to turbulence in the canopy? This is highly uncertain, but can be substantial. Could it be that too much ozone is getting deep into the canopy?

JULES uses an $O_3$ concentration at reference level, this does not change with depth into the canopy due to changes in turbulence and mixing. This is something that needs consideration for development within JULES as it is possible that leaves at the bottom of the canopy will see too much $O_3$. But currently, because this process is highly uncertain, in JULES the simplest approach is taken whereby each layer of the canopy has the same $O_3$ concentration and the uptake of $O_3$ is dependent on the rate of stomatal conductance at that canopy layer, which does vary with depth into the canopy depending on light and Nitrogen availability. This would not be an issue for short vegetation such as grasslands and crops which are the dominant land cover in our study, but may be more significant for forests. However, some studies show minimal vertical $O_3$ concentration gradients within forest canopies. For example, Karlsson et al (2006) found daytime mean $O_3$ of 34.5 ppb at 13m, and 33.1 ppb at 3m, in an 18-20m tall Norway Spruce forest. The 13m values were just 4% lower than measurements made at 13m in clearing, suggesting rather uniform conditions. The same study reported a 6% difference in $O_3$ between 10m and 20m observations in a separate 20-25m Norway spruce site. Reactions with biogenic VOC emissions can also reduce $O_3$ in the canopy, but even at a chemically very reactive oak forest in the USA, $O_3$ concentrations gradients within the upper canopy were small (Fuentes et al, 2007).

Karlsson, P., Hansson, M., Hoglund, H.-O. & Pleijel, H. Ozone concentration gradients and wind conditions in Norway spruce (Picea abies) forests in Sweden Atmos. Environ., 2006, 40, 1610-1618.

Fuentes, J. D., Wang, D., Bowling, D. R., Potosnak, M., Monson, R. K., Goliff, W. S. & Stockwell, W. R. Biogenic hydrocarbon chemistry within and above a mixed deciduous forest J. Atmos. Chem., 2007, 56, 165-185.

Line 270-273: I'm finding this hard to follow, especially because how the authors refer to the sensitivity simulations is "high/low plant ozone sensitivity". I understand that within each sensitivity simulation there are variations among land cover types in terms of the degree of the sensitivity to ozone applied. Further, ozone "dose-response functions" is never defined. Again, it seems like this calibration is a fundamental part of the authors' analysis. I would suggest that some supplemental material is moved to the main text and cleaned up so the methods are very clear.
Line 286-9: What observations? I am certain FO3crit cannot be measured, only inferred. Cumulative ozone uptake over what time period?

To address both points above (Line 270 to 289), we have amended section 2.2 (Calibration of $O_3$ uptake model) to improve clarity and have moved information for the SI into the main text (Line 277 - 352):

"Here we use the latest literature on flux based $O_3$ dose-response relationships derived from observed field data across Europe (CLRTAP, 2017) to determine the key PFT-specific $O_3$ sensitivity parameters in JULES (*a* and *$Fo_{3crit}$*). Synthesis of information expressed as $O_3$ flux based dose-response relationships derived from field experiments is carried out by The United Nations Convention on Long-Range Transboundary Air Pollution (CLRTAP Convention), this information is then used as a policy tool to inform emission reduction strategies in Europe to improve air quality (CLRTAP, 2017;Mills et al., 2011a). Derivation of $O_3$ flux based dose-response relationships for different vegetation types uses the accumulated stomatal $O_3$ flux above a threshold (often referred to as the phytotoxic $O_3$ dose above a threshold of 'y' i.e. $POD_y$) as the dose metric, and the percentage change in biomass as the response metric (Emberson et al., 2007;Karlsson et al., 2007). We use these observation based $O_3$ dose-response relationships to calibrate each JULES PFT for sensitivity to $O_3$ using available relationships for the closest matching vegetation type. For JULES, *$Fo_{3crit}$* is the threshold for $O_3$ damage, and values for this parameter are taken from the $O_3$ dose-response relationships as the $POD_y$ value. The actual sensitivity to $O_3$ is determined by the slope of the $O_3$ dose-response relationship, i.e. how much biomass changes with accumulated stomatal uptake of $O_3$ above the damage threshold, this relates to the parameter *a* in JULES. The parameter '*a*' is a PFT-specific parameter representing the fractional reduction of photosynthesis with O3 uptake by leaves. Values for this parameter are found for each PFT by running JULES with different values of '*a*', which alter the instantaneous photosynthetic rate, but then calculating the accumulated stomatal flux of $O_3$ and resulting change in productivity over the same period, until the slope of this relationship produced by the JULES simulations matches that of the $O_3$ dose-response relationships derived from observations. Essentially we calibrate each JULES PFT for sensitivity to $O_3$ by reproducing the observed $O_3$ dose-response relationships.

Each PFT was calibrated for a high and low plant $O_3$ sensitivity to account for uncertainty in the sensitivity of different plant species to $O_3$, using the approach of Sitch *et al.*, (2007). Therefore, when using our results to assess the impact of $O_3$ at the land surface, we are able to provide a range in our estimates to help address some of the uncertainty in the $O_3$ response of different vegetation types. In addition, where possible owing to available data, a distinction was made for Mediterranean regions. This was because the work of Büker et al. (2015) showed that different $O_3$ dose-response relationships are needed to describe the $O_3$ sensitivity of dominant Mediterranean trees. For the $C_3$ herbaceous PFT, the dominant land cover type across the European domain in this study (Fig. S1), the high plant $O_3$ sensitivity was calibrated against observations for wheat to give a representation of agricultural regions and wheat is one of the most sensitive grasses to $O_3$ (Fig. S2, Table S1). For the low plant $O_3$ sensitivity JULES was calibrated against the dose-response function for natural grassland to give a representation of natural grassland and this vegetation has a much lower sensitivity to $O_3$ damage, for the Mediterranean region we used a function for Mediterranean natural grasslands, all taken from CLRTAP (2017) (Fig. S2, Table S1). Tree/shrub PFTs were calibrated against observed $O_3$ dose-response functions for the high plant $O_3$ sensitivity: broadleaf trees (temperate/boreal) = Birch/Beech dose-response relationship, broadleaf trees (Mediterranean) = deciduous oaks dose-response relationship, needle leaf trees = Norway spruce dose-response relationship, shrubs = Birch/Beech dose-response relationship, all from CLRTAP (2017) (Fig. S2, Table S1). Data on $O_3$ dose-response relationships for different vegetation types is very limited, therefore for the low plant $O_3$ sensitivity calibration for trees/shrubs we assumed a 20% decrease in sensitivity to $O_3$ based on the difference in sensitivity between high and low sensitive tree species in the Karlsson et al. (2007) study. Due to limitations in data availability, the shrub parameterisation uses the observed dose-response functions for broadleaf trees. Similarly, the parameterisation for $C_4$ herbaceous uses the observed dose-responses for $C_3$ herbaceous, however the fractional cover of $C_4$ herbs across Europe is low (Fig. S1), so this assumption affects a very small percentage of land cover.

To calibrate the JULES $O_3$ uptake model, JULES was run across Europe forced using the WFDEI observational climate dataset (Weedon, 2013) at $0.5^o$ X $0.5^o$ spatial and three hour temporal resolution. JULES uses interpolation to disaggregate the forcing data down from 3 hours to an hourly model time step. The model was spun-up over the period 1979 to 1999 with a fixed atmospheric $CO_2$ concentration of 368.33 ppm (1999 value from Mauna Loa observations, (Tans and Keeling)). Zero tropospheric ozone concentration was assumed for the control simulation, for the simulations with $O_3$, spin-up used spatially explicit fields of present day $O_3$

concentration produced using the UK Chemistry and Aerosol (UKCA) model with standard chemistry from the run evaluated by O'Connor et al. (2014). A fixed land cover map was used based on IGBP (International Geosphere-Biosphere Programme) land cover classes (IGBP-DIS), therefore as the vegetation distribution was fixed and the calibration was not looking at carbon stores, a short spin-up was adequate to equilibrate soil temperature and soil moisture. JULES was then run for the year 2000 with a corresponding $CO_2$ concentration of 369.52 ppm (from Mauna Loa observations, (Tans and Keeling)) and monthly fields of spatially explicit tropospheric $O_3$ (O'Connor et al., 2014) as necessary.

Calibration was performed using four simulations:  with i) zero tropospheric $O_3$ concentration, this was the control simulation (control), ii) tropospheric $O_3$ at current ambient concentration (O3), iii) ambient +20 ppb (O3+20) and iv) ambient +40 ppb (O3+40). The different $O_3$ simulations (i.e. O3, O3+20 and O3+40) were used to capture the range of $O_3$ conditions in the data used to derive the observed $O_3$ dose-response relationships used here for calibration, often data were used from experiments using artificially manipulated conditions of ambient + 40 ppb $O_3$ for example. For each JULES $O_3$ simulation, the value of $F_{O3crit}$ was taken from the vegetation specific $O_3$ dose-response relationship as the threshold $O_3$ concentration above which damage to vegetation occurs. An initial estimate of the parameter '*a*' was used, then for each PFT and each simulation, hourly estimates of NPP (our proxy for biomass – although not identical they are related) and $O_3$ uptake in excess of $F_{O3crit}$ were accumulated over a PFT dependent accumulation period. The accumulation periods were ~6 months for broadleaf trees and shrubs, all year for needle leaf trees, and ~3 months for herbaceous species, through the growing season, following guidelines in CLRTAP (2017). Additionally, in accordance with the methods used in the CLRTAP (2017) that describe how the $O_3$ dose-response relationships are derived from observations, we use the stomatal $O_3$ flux per projected leaf area to top canopy sunlit leaves. The percentage change in total NPP was calculated for each $O_3$ simulation and plotted against the cumulative uptake of $O_3$ over the PFT-specific accumulation period. The linear regression of this relationship was calculated, and slope and intercept compared against the slope and intercept of the observed dose-response relationships. Values of the parameter 'a' were adjusted, and the procedure repeated until the linear regression through the simulation points matched that of the observations (Fig. S2, Table S1)."

Line 317-324: This discussion seems out-of-place here. It might be more appropriate in the conclusions w.r.t. the "next steps", please revise.

This has been moved, and is a new paragraph in section 4.3 (Line 879 - 887):

"In this work we implement the stomatal closure proposed in Medlyn et al., (2011), this uses the parameter $g_1$. Hoshika et al. (2013) show a significant difference in the $g_1$ parameter (higher in elevated $O_3$ compared to ambient) in Siebold's beech in June of their experiment. However, this is only at the start of the growing season, further measurements show no difference in this parameter between $O_3$ treatments. Quantifying an $O_3$ effect directly on $g_1$ would require a detailed meta-analysis of empirical data on photosynthesis and $g_s$ for different PFTs, which is currently lacking in the literature. With such information, here we take an empirical approach to modelling plant $O_3$ damage, essentially by applying a reduction factor to the simulated plant photosynthesis based on observations of whole plant losses of biomass with $O_3$ exposure, for which there is a lot more available data (e.g. CLRTAP, 2017)."

Line 324-325. Please include more details in the main text as to why the author would go to the supplemental for this analysis.

We have amended the main text as follows, and moved information from the SI into the main text (Line 382 - 401):

"The impact of gs model formulation (JAC versus MED) on simulated water, O3, carbon and energy fluxes is compared for two contrasting grid points - wet (low soil moisture stress) and dry (high soil moisture stress) in the European domain. JULES was spun-up for 20 years (1979-1999) at two grid points in central Europe representing a wet (low soil moisture stress, lat: 48.25; lon:, 5.25) and a dry site (high soil moisture stress, lat:

38.25; lon:, -7.75). The modelled soil moisture stress factor (fsmc) at the wet site ranged from 0.8 to 1.0 over the year 2000 (1.0 indicates no soil moisture stress), and at the dry site fsmc steadily declined from 0.8 at the start of the year to 0.25 by the end of the summer. The WFDEI meteorological forcing dataset was used (Weedon, 2013), along with atmospheric CO2 concentration for the year 1999 (368.33 ppm), and either no O3 (i.e. the O3 damage model was switched off) for the control simulations, or spatially explicit fields of present day O3 concentration produced using the UK Chemistry and Aerosol (UKCA) model from the run evaluated by O'Connor et al. (2014) for the simulations with O3. Following the spin-up period, JULES was run for one year (2000) with corresponding atmospheric CO2 concentration, and tropospheric O3 concentrations as described above. The control and ozone simulations were performed for both JAC and MED model formulations. Land cover for the spin-up and main run was fixed at 20% for each PFT. For the simulations including O3 damage, the high plant O3 sensitivity parameterisation was used. The difference between these simulations was used to assess the impact of gs model formulation on the leaf level fluxes of carbon and water. We calculate and report in the main manuscript (results section 3.1), the difference in mean annual water-use that results from the above simulations using the different gs models. For each day of the simulation we calculate the percentage difference in water-use between the two simulations, we then calculate the mean and standard deviation over the year to give the annual mean leaf-level water-use."

Line 326: Please briefly state the results of the FLUXNET model evaluation in the main text. Also, the text in the supplemental says that there are large improvements in the seasonal cycle. Large seems like a stretch - instead I would quantify the changes in the RMSE.

We have moved the results of the Fluxnet model evaluation from the SI to the main text (results section 3.1). We quantify the changes in RMSE (Line 554 - 572).

"Site level evaluation of the seasonal cycles of latent and sensible heat with both JAC and MED models compared to FLUXNET observations showed in general, the MED model improved the seasonal cycle of both fluxes (lower RMSE), but the magnitude of this varied from site to site (Fig. S12). At the deciduous broadleaf site, US-UMB, MED resulted in improvements of the simulated seasonal cycle particularly in the summer months for both fluxes (RMSE decreased from 42.7/31.5 to 38.5/28.0 $W/m^2$ for latent/sensible heat respectively). At the second deciduous broadleaf site IT-CA1 however, there was almost no difference between the two $g_s$ models. Both evergreen needle leaf forest sites (FI-Hyy and DE-Tha) saw improvements in the simulated seasonal cycles of latent and sensible heat with the MED model, primarily as a result of lower latent heat flux in the spring and summer months, and higher sensible heat flux over the same period. At FI-Hyy, RMSE decreased from 10.1/7.4 to 6.7/6.7 $W/m^2$ for latent/sensible heat respectively, and at DE-Tha, RMSE decreased from 16.0/11.9 to 10.5/10.6 $W/m^2$ for latent/sensible heat respectively. With the MED model the monthly mean latent heat flux was improved at the $C_3$ grass site (CH-Cha) as a result of increased flux in the summer months (RMSE decreased from 15.7 to 13.8 $W/m^2$), however there was no improvement in the sensible heat flux and RMSE with MED was increased (from 3.9 to 4.9 $W/m^2$). At the $C_4$ grass site (US-SRG), small improvements were made in the seasonal cycle of both latent and sensible heat with the MED model. At the deciduous savannah site (CG-Tch) which included a high proportion of shrub PFT in the land cover type used in the site simulation, large improvements in the seasonal cycle of both fluxes were simulated with the MED model, as a result of a decrease in the latent heat flux and an increase in the sensible heat flux (RMSE decreased from 39.5/31.6 to 30.4/24.4 $W/m^2$ for latent/sensible heat respectively)."

Line 335: I would urge the authors to stay away from suggesting that their analysis will allow a "full understanding"

This has been changed to "focus on" (Line 417).

Line 345: By "no form of land management", do the authors mean that there is no harvesting of crops or grazing of grasses? If so, how "big" do crops and grasses get, and what does this mean for their results?

We have modified this paragraph to clarify it (Line 424 - 433):

"JULES was run including dynamic vegetation with a land cover mask giving the fraction of agriculture in each 0.5º x 0.5º grid cell based on the Hurtt et al. (2011) land cover database for the year 2000. This means that whilst the model is allowed to evolve its own vegetation cover, within the agricultural mask only $C_3/C_4$ herbaceous PFTs are allowed to grow, with no competition from other PFTs. Therefore, through the simulation period, regions of agriculture are maintained as such and not out-competed by forests for example, allowing for a more accurate representation of the land cover of Europe in the model. No form of land management is simulated (i.e. no crop harvesting, ploughing, rotation or grazing), growth and leaf area index (LAI) are determined by resource availability and phenology. Outside of the agricultural mask, dynamic vegetation means that grid cell PFT coverage and LAI are the result of resource availability, phenology and simulated competition. Across the model domain, simulated mean annual LAI was dominantly within the range of 2 to 5 $m^2/m^2$ (Fig. S3 and S4)."

We mention the implications of no land management in the model in the discussion section 4.3 (Line 848 - 850):

"Additionally, this version of JULES does not have a crop module; it has no land management practices such as harvesting, ploughing or crop rotation – processes which may have counteracting effects on the land carbon sink."

Line 347: Does the change follow Hurtt et al. 2011? I would refrain from using the term "little" as this gives the reader little understanding of what is going on. Showing only 2050 is not too helpful.

We have modified this sentence to give more detail on the change in fractional land cover over the simulation period. This change does not follow Hurrtt et al., 2011 as we clearly state in the preceding sentences that the model is run with dynamic vegetation so is allowed to evolve its own vegetation cover, but that we apply an agricultural mask to maintain the extent of agricultural regions and these are based on the Hurrtt et al., 2011 data for the year 2000 (Line 433 - 437).

"Following a full spin-up period (to ensure equilibrium vegetation, carbon and water states), there was no significant change in the fractional cover of each PFT over the simulation period (1901 - 2050). By 2050, increases in boreal forest cover occurred, but this was less than 2% and limited to very small areas, given this small change we show just the land cover for 2050 in Fig. S1."

Lines 356-8: Whether emissions or meteorology matters more is going to depend on the emissions and climate variability. Langner et al. (2012b) only examine 1990 onwards and so I don't think the authors can use this work to comment on emissions vs. meteorology in 1900-1959.

We have amended this sentence as follows (Line 446 - 447):

"This procedure introduces some uncertainty of course, although Langner et al. (2012b) show that for the period 1990 to 2100 it is emissions change, rather than meteorological change, that drives modelled ozone concentrations."

The period 1990-2100 covers a period of 111 years, and a period in which climate-change is likely to show most effects, it therefore seems unlikely that things would be very different over the 60 year period of 1900-1959.

Lines 358-361: I don't really know what this means. The authors should show the trend in emissions of NOx, methane, and isoprene from 1900-2050 over Europe, which is standard practice in atmospheric chemistry papers, so that readers can fully understand the emission scenarios used, as this is central to the findings.

We didn't show such trends since there are many papers dealing with such emissions, and the focus of this paper is on the carbon sink and impacts at the land surface rather than on atmospheric chemistry. However, we have now added a new Figure, and text (Line 450 - 455):

"The trend in emissions of the major $O_3$ precursors $NO_x$, NMVOC and Isoprene are shown from 1900 to 2050 over Europe in Fig. S5. Isoprene emissions are not inputs to the EMEP model, but rather calculated at each time-step using temperature, radiation, and land-cover specific emission factors (Simpson et al., 2012). Changes in the assumed background concentration of $CH_4$ (from RCP6.0) (van Vuuren et al., 2011) are also shown in Fig. S5. Engardt et al. (2017) show the trend in emissions of $SO_2$ and $NH_3$ from 1900 to 2050 over Europe."

Line 362-3: This is confusing. I would cut this everything after "however"

This has been removed.

Line 367-369: Please clarify how the authors map the ozone concentrations from the land cover categories to the model. What do the differences in ozone concentrations over the different land cover types represent? Differences in dry deposition, BVOC emissions, or just turbulent mixing? Instead of saying "more accurate", the authors should just say something like ozone concentrations peak during the day so it's important to take the diurnal cycle into account.

This paragraph has been clarified (Line 458 - 469):

"$O_3$ concentrations from EMEP MSC-W were calculated at canopy height for two land-cover categories: forest and grassland (Fig. S6 and Fig. S7), which are taken as surrogates for high and low vegetation, respectively. These canopy-height specific concentrations allow for the large gradients in $O_3$ concentration that can occur in the lowest 10s of metres, giving lower $O_3$ for grasslands than seen at e.g. 20 m in a forest canopy (Gerosa et al., 2017;Simpson et al., 2012;Tuovinen et al., 2009). These canopy level $O_3$ concentrations are used as input to JULES, using the EMEP $O_3$ concentrations for forest for the forest JULES PFTs (broadleaf/needle leaf tree and shrub), and the EMEP $O_3$ concentrations for grassland for the grass/herbaceous JULES PFTs ($C_3$ and $C_4$). This study used daily mean values of tropospheric $O_3$ concentration from EMEP disaggregated down to the hourly JULES model time-step. The daily mean $O_3$ forcing was disaggregated to follow a mean diurnal profile of $O_3$, this was generated from hourly $O_3$ output from EMEP MSC-W for the two land cover categories (forest and grassland as described above) across the same model domain. $O_3$ concentrations follow a diurnal cycle and peak during the day, therefore accounting for the diurnal variation in $O_3$ concentrations allows for a more realistic estimation of $O_3$ uptake."

Line 377: Typo

This has been amended.

Lines 381-394: Some of this is incorrect and the discussion is lengthy. I simply wanted the authors to note changes in the seasonal cycle of ozone depend strongly on anthropogenic Nox (not because the timing of emissions during the year, rather nonlinear ozone chemistry), the emissions scenario matters for the results regarding uptake of ozone to vegetation, which it seems like they are getting at eventually. I would cut most of this.

This paragraph has been modified as follows (Line 471 - 483):

"Figure 1 shows large increases in tropospheric $O_3$ from pre-industrial to present day (2001), this is in line with modelling studies (Young et al., 2013) and site observations (Derwent et al., 2008;Logan et al., 2012;Parrish et al., 2012), and is predominantly a result of increasing anthropogenic emissions (Young et al., 2013). Figures S6 and S7 show this large increase in ground-level $O_3$ concentrations from 1901 to 2001 occurs in all seasons. Present day $O_3$ concentration show a strong seasonal cycle, with a spring/summer peak in concentrations in the mid-latitudes of the Northern Hemisphere (Derwent et al., 2008;Parrish et al., 2012;Vingarzan, 2004). Seasonal cycles have been changing over the past decades however, attributed to changes in $NO_x$ and other emissions, as well as changes in transport patterns (Parrish et al., 2013). These changes will likely continue in future as emissions and meteorological factors impact photo-chemical ozone production and transport patterns. Indeed, the $O_3$ concentrations used in the simulations in this study show increased $O_3$ levels in winter and in some regions in autumn and spring in 2050 compared to present day, this may be due to reduced titration of $O_3$ by NO as a result of reduced $NO_X$ emissions in the future (Royal Society, 2008). Summer $O_3$ concentrations are lower in 2050 however, compared to 2001. "

Lines 394-397: Jumping from surface ozone seasonality to plant phenology seems erratic. I would suggest moving this discussion elsewhere.

This has been deleted and moved into the discussion section 4.3 (Line 855 - 858).

Line 420: I see that the authors examine ozone impacts on stomatal conductance, which could be referred to as "plant physiology", but it doesn't seem to me like GPP and C sink are "plant physiology" entities.

We have changed this to (Line 505 - 507):

"We use these simulations to investigate the direct effects of changing atmospheric $CO_2$ and $O_3$ concentrations, individually and combined, on plant water-use, GPP and the land C sink through the twentieth century and into the future, specifically over three time periods:…"

Line 421-423: The authors should tell the reader why we should go to the supplemental. "for calculation of the effects due to" is vague.

This has been changed to move information from the SI into the main text (Line 508 – 525):

"For each time period we calculate the difference between the decadal means calculated at the start and end of the analysis period for each variable of interest. Therefore our results report the change in GPP, for example, over the analysis period. For each variable analysed (GPP, NPP, vegetation carbon, soil carbon, total land carbon and $gs$), we use the mean over 10 years to represent each time period, e.g. the mean over 2040 to 2050 is what we call 2050, 1901 to 1910 is what we refer to as 1901. The difference between the simulations gives the effect of $O_3$ and $CO_2$ either separately or in combination over the different time periods. We look at the percentage change due to either $O_3$ at pre-industrial $CO_2$ concentration (i.e. without the additional effect of atmospheric $CO_2$ on stomatal behaviour - O3 simulation), $CO_2$ (at fixed pre-industrial $O_3$ concentration, CO2 simulation) or the combined effect of both gases (CO2+O3 simulation), which is calculated as:

$$100 * (var[y_1] – var[y_2]) / var[y_2] \qquad (8)$$

Where $var[y_x]$ represents the variable in time period y, e.g. $100 * (varO_3[2050] – varO_3[1901]) / varO_3[1901]$ gives the $O_3$ effect (at fixed $CO_2$) over the full experimental period. The meteorological forcing is prescribed in these simulations and is therefore the same between the model runs. Other climate factors, such as VPD, temperature and soil moisture availability are accounted for in our simulations, but our analysis isolates the effects of $O_3$, $CO_2$ and $O_3 + CO_2$. We also use paired t-test to determine statistically significant differences between the different (high and low) plant $O_3$ sensitivities."

Line 435: What is a wet site? Specify in the main text.

This is now specified in the main text (Line 537):

"The impact of $g_s$ model on simulated $g_s$ is shown for the site with low soil moisture stress (wet site, Fig. 2)."

Line 441: Same for dry site.

This has been changed (Line 544):

"This comparison was also done for a dry site (high soil moisture stress)…"

Line 442-5: Why should one wet and one dry site represent the entire domain?

This has been removed.

Line 445-447: It's not clear what the authors' point here is. Since the authors' simulations are uncoupled, it's an added uncertainty that changing stomatal conductance is going to impact energy partitioning and thus meteorology. Is that all they are trying to get at here?

We simply show here that changes in the stomatal conductance of the model alter the partitioning between the energy fluxes in these uncoupled simulations. We discuss later that potentially this could have impacts on meteorology, but that fully coupled simulations would be necessary to detect these effects (see discussion section 4.1). This is an interesting discussion point and area of future work worth noting.

Line 455-457: Why do the authors show the bottom row? Is it giving more information then the top row? I would understand if the stomatal uptake and ozone damage fed back onto the ozone concentrations the authors would need the bottom row. As this is not the case, this bottom row should be cut; but I agree that the authors should make this point in the text, which they do. Please clarify in the text what further details are in the supplemental.

The bottom row was simply to show that the different gs models simulate different rates of gs for each of the PFTs, and that consequently this affects the flux of $O_3$ into stomata. We have moved the bottom row of this figure in the SI (Fig. S11).

Line 473: What do the authors conclude about the comparison between the simulations with high and low ozone sensitivity vs the MTE-GPP product?

We have added this paragraph to the discussion section 4.1 (Line 780 - 794):

"We evaluated the JULES $O_3$ model by comparing modelled GPP against the Jung et al (2011) MTE product. Similar spatial patterns of GPP were simulated by JULES compared to MTE. Zonal means also showed similar patterns of GPP, although JULES under predicted GPP compared to MTE at latitudes >45°N (temperate and boreal regions; all simulations) and over predicted GPP at latitudes <45°N (Mediterranean region; all simulations). The simulations with transient $O_3$ (i.e. $O_3$ only and $CO_2$ + $O_3$) showed large differences in GPP between the high and low plant $O_3$ sensitivity simulations, this is to be expected given that the high plant $O_3$ sensitivity simulations were parameterised to be 'damaged' more by $O_3$, i.e. greater reduction of photosynthesis/$g_s$ with $O_3$ exposure compared to the low plant $O_3$ sensitivity simulations. This difference was largest in the temperate zone, largely because of $C_3$ grass cover being the dominant land cover here and the difference in the sensitivity to $O_3$ between the high and low calibrations is significantly larger for $C_3$ grasses compared to the needle leaf trees that dominate in the boreal region. Additionally, a longer growing season in the temperate region may allow for greater uptake of $O_3$ into vegetation. $C_3$ grass is also the dominant land cover in the Mediterranean region with a different calibration used for Mediterranean grasses for the low plant $O_3$ sensitivity which is less sensitive to $O_3$ than the temperate $C_3$ grasses, but high soil moisture stress is common throughout the growing season in the Mediterranean limiting the uptake of $O_3$ through stomata, which likely diminishes the difference between the high and low calibrations."

Line 484-5: Please re-phrase so that it is clear that the GPP simulated by the low vs. high ozone sensitivity is significantly different

This has been changed to (Line 601 - 603):

"Over the historical period (1901-2001), $O_3$ reduced GPP under both the low and high plant $O_3$ sensitivity parameterizations by -3% to -9% respectively (Table 1), and this difference in simulated GPP was significant ($t$=102.2, $d.f.$=6270, $p$<2.2e$^{-16}$)."

Lines 483-503: It's confusing in the text whether the authors are discussing changes in the trend from 1901-2001, or changes in the average, due to ozone. Please revise the text accordingly.

In response to a comment above, in section 2.4.2 we have clarified this, and we state what we report in the results, we move details from the SI to the main text of how we calculate this (Line 505 - 525):

"We use these simulations to investigate the direct effects of changing atmospheric $CO_2$ and $O_3$ concentrations, individually and combined, on plant water-use, GPP and the land C sink through the twentieth century and into the future, specifically over three time periods: historical (1901-2001), future (2001-2050) and over the full time series (1901-2050). For each time period we calculate the difference between the decadal means calculated at the start and end of the analysis period for each variable of interest. Therefore our results report the change in GPP, for example, over the analysis period. For each variable analysed (GPP, NPP, vegetation carbon, soil carbon, total land carbon and $gs$), we use the mean over 10 years to represent each time period, e.g. the mean over 2040 to 2050 is what we call 2050, 1901 to 1910 is what we refer to as 1901. The difference between the simulations gives the effect of $O_3$ and $CO_2$ either separately or in combination over the different time periods. We look at the percentage change due to either $O_3$ at pre-industrial $CO_2$ concentration (i.e. without the additional effect of atmospheric $CO_2$ on stomatal behaviour – O3 simulation), $CO_2$ (at fixed pre-industrial $O_3$ concentration, CO2 simulation) or the combined effect of both gases (CO2+O3 simulation), which is calculated as:

$$100 * (var[y_1] - var[y_2]) / var[y_2] \qquad\qquad (8)$$

Where $var[y_x]$ represents the variable in time period y, e.g. $100 * (varO_3[2050] - varO_3[1901]) / varO_3[1901]$ gives the $O_3$ effect (at fixed $CO_2$) over the full experimental period. The meteorological forcing is prescribed in these simulations and is therefore the same between the model runs. Other climate factors, such as VPD, temperature and soil moisture availability are accounted for in our simulations, but our analysis isolates the effects of $O_3$, $CO_2$ and $O_3 + CO_2$. We also use paired t-test to determine statistically significant differences between the different (high and low) plant $O_3$ sensitivities."

Line 516-7: Again, suggesting that the O3 impact on the land carbon sink is a source of carbon is not really appropriate; re-phrasing would allow for the same take-away

This has been changed to (Line 636 – 638):

"By comparison with one of the largest anthropogenic emissions of carbon for Europe, we show here the effect of $O_3$ on reducing the size of the European land carbon sink is notable."

Line 523: Please quantify the "large" spatial variability
Line 527-529: With "therefore", are the authors suggesting that the decreases in GPP are from springtime increases in temperate/Mediterranean regions are because springtime ozone is increases? Please clarify in the text. What is going on in the boreal region?
Line 529: Ok, so the previous sentences are 13nalysing the simulations without CO2 fertilization? It would be best to make this clear before this point.

The changes is this paragraph address all three points above (Line 523 to 529).

We have added "as discussed below" in the first sentence because we go on to describe the large spatial variability in the next few sentences. We clarify that the results are referring to the simulations with $O_3$ only, and that the variability is due to the variability in the $O_3$ concentration:

"Over the 2001 to 2050 period, region-wide GPP with O3 only changing (O3 simulation) increased marginally (+0.1% to +0.2%, high and low plant O3 sensitivity, Table 1, with a significant difference between the two plant O3 sensitivities (t=57, d.f.=6270 p<2.2e-16)), although with large spatial variability as discussed below (Fig. 4g & h). Figures S6 and S7 show that despite decreased tropospheric O3 concentrations by 2050 in summer compared to 2001 levels, all regions are exposed to an increase in O3 over the wintertime, and some regions of Europe, particularly temperate/Mediterranean experience increases in O3 concentration in spring and autumn. Therefore, although in the O3 simulation, overall simulated GPP for Europe shows a small increase, large spatial variability is shown in Fig's 4g &h because of the variability in O3 concentration with region and season. Increased GPP (dominantly 10%, but up to 20% in some areas) on 2001 levels is simulated across areas of Europe, however, decreases of up to 21% are simulated in some areas of the Mediterranean, up to 15% in some areas of the boreal region and up to 27% in the temperate zone (Fig. 4g & h). "

Line 533-534: What are the implications of this?

We have changed this sentence as below (Line 658 - 660):

"Nevertheless, although the percentage gain is larger, the absolute value of GPP by 2050 remains lower compared to GPP with $CO_2$ only changing, highlighting the negative impact of $O_3$ at the land surface (Table S4)."

Line 567: "Over the Anthropocene" is ambiguous

We have removed the use of the term Anthropocene and refer to it as the full experimental period or give the years 1901 to 2050.

Line 634: The authors' use of "leaf-level" stomatal conductance in this paragraph is confusing; earlier they define leaf-level stomatal conductance as non-canopy integrated stomatal conductance; is this what they are examining here?

Apologies, we have removed use of the term leaf-level to stop confusion.

Lines 633-648: I would like to see some discussion of the model evaluation of the stomatal conductance models (e.g., FLUXNET). Regarding the last sentence of this paragraph, I would make this statement specific to the uncoupled approach. Higher deposition would reduce ozone concentrations in a coupled chemistry-land study.

We add discussion of the sites-level evaluation of the gs models here (Line 766 - 769):

"Site-level evaluation of the models against Fluxnet observations showed that in general the MED model improved simulated seasonal cycles of latent and sensible heat. The magnitude of the improvement varied with site, large improvements were seen at the deciduous savanna site, and at the NT sites and BT site (US_UMB) in the spring and summer. However, much smaller improvements were seen at the grass sites."

We changes the last sentence accordingly (Line 775 - 778):

"Therefore, given that $C_3$ herbaceous vegetation is the dominant land cover class across the European domain used in this study, this suggests a greater $O_3$ impact for Europe would be simulated with MED model compared to JAC in our simulations where chemistry is uncoupled from the land surface."

Lines 649-661: Do the authors have any hypotheses for why their study shows lower impact on GPP, or do the authors think their results are reasonable in comparison to the other work? On that note, I do not see any support for the last sentence of the paragraph. I would encourage the authors to change the phrasing to be more speculative (instead of saying that this is "likely" the result of).

We have amended this paragraph. It is difficult to hypothesis as to why estimates differ between the models and as such we have removed the last sentence (Line 806 - 812):

"Our estimates of changes in current day GPP and NPP are at the lower end of previously modelled estimates. Simulated $O_3$ impacts will depend in a large part on the scenario of $O_3$ concentrations used as forcing, meteorological forcing and how sensitive vegetation is parameterised to be to $O_3$ damage, in addition to the different process representation of $O_3$ damage in each model. It is therefore difficult to hypothesise as to exactly why modelled estimates differ, but suggests that an ensemble approach to modelling $O_3$ impacts on the terrestrial biosphere would be beneficial to understand some of these differences and provide estimates of $O_3$ damage with uncertainties."

Line 687-691: Using a stomatal conductance parameterization that simulates higher gs will certainly lead to higher uptake. The higher uptake may decrease ozone concentrations, but the stronger ozone damage may increase ozone concentrations. It's hard to say which will dominate in the authors' uncoupled simulations, especially because ozone is fairly well-buffered in models (one sink reduces, another sink kicks in), how the high vs. low ozone sensitivity simulations will be different, and if this high sensitivity study is indeed an "upper bound".

We have modified this paragraph accordingly and remove the sentences referring to this study as an upper bound (Line 836 - 848):

"We include a representation of agricultural regions through the model calibration against the wheat $O_3$ sensitivity function (CLRTAP, 2017), and in our simulations the high plant $O_3$ sensitivity scenario uses this calibration against wheat for all $C_3$/$C_4$ land cover which dominates our model domain. Wheat is known to be one of the most $O_3$ sensitive crop species however, so it is possible that our simulations over-estimate the $O_3$ impact at the land surface. However, the low plant $O_3$ sensitivity calibration against natural grasslands provides a counter estimate of the impact of $O_3$ at the land surface, therefore it is important to consider the range our results provide (i.e. both the high and low plant $O_3$ sensitivity) as an indicator of the impact of $O_3$ on the land surface. As with all uncoupled modelling studies, a change in $g_s$ and flux will impact the $O_3$ concentration itself. Therefore adopting the Medlyn formulation with a higher $g_s$ and subsequently higher $O_3$ flux for broadleaf and $C_3$ PFTs (Fig 2) would lead to reduced $O_3$ concentration, which in turn would act to dampen the effect of higher $g_s$ on $O_3$ flux, although the higher uptake of $O_3$ by vegetation may lead to more damage and increase $O_3$ concentrations, in an uncoupled chemistry-land modelling system such as this it is not possible to predict which process would dominate."

This has been changed.

Line 718: Here it is relevant to discuss the findings of Lombardozzi that there are separate impacts of ozone on photosynthesis and stomatal conductance

We discuss the modelling approach of Lombardozzi and the results at other points in the manuscript, so we prefer not to discuss again here. This paragraph is also discussing $O_3$ induced sluggish stomatal behaviour observed in plants, whilst Lombardozzi et al separate the impacts of $O_3$ on photosynthesis and stomatal conductance, it is not a representation of sluggish stomatal control.

Line 724: By gmax are the authors referring to input, or output of the model? If input, I don't think the authors' reasoning makes sense. The parameterizations are different and act to scale stomatal conductance by very different entities.

We have clarified this point below (Line 892 - 896):

"We acknowledge this inconsistency as a caveat of our study, however comparison of *gmax* (maximum $g_s$) values from both models (EMEP (*gmax* is an input parameter determining the maximum $g_s$) and JULES (*gmax* is not used as an input parameter in JULES, instead we calculated *gmax* for each PFT taking the mean across the model domain for the year 2001) suggests the differences are small for deciduous forest……".

Line 728-730: Ozone deposition can have a substantial impact on surface ozone concentrations (Val Martin et al., 2014). I would not argue this.

Line 730-732: Where is there evidence that stomatal conductance does not influence ozone concentrations above the canopy? I would not argue this. Further, strong vertical mixing above trees means that this is not the limiting factor for deposition – rather stomatal deposition and nonstomatal deposition are.

Line 732-733: Again, where in the literature is there evidence of this?

Line 733-736: I'm not sure what the authors are getting at here.

The points raised concerning lines 728-736 are related, and so we address them as one here. We have re-phrased the text of lines 728-736 to make a clearer distinction between the role of deposition in regional and above-canopy $O_3$ (Line 898 - 909):

[revised manuscript text omitted]

---

## Author Response (AR3)

[revised manuscript text omitted]

 Response to Reviewers comments

We would like to thank the reviewer again for their time to read the manuscript and comment on it. This has
improved the manuscript and we hope the reviewer finds our changes satisfactory. Please find our comments
below.

Thanks to the authors for moving the discussion of the calibration to the main text. I think this
strengthens the paper, and clarifies the novelty of the work. I think the work should be
published after the authors address minor revisions.

General comments.
I don't find the naming of the different simulations to O3, CO2, and CO2+O3 very helpful
especially when the authors are referring to the gases as O3 and CO2 at the same time.
Something like run_O3, or run_CO2, run_both_O3_CO2?

We have renamed the simulations as advised above.

I'm a bit confused with respect to the title of the paper, especially because the authors have
a
section dedicated to their findings that the impact isn't as large as expected.

The title reflects that our simulations show a large impact of $O_3$ from 1901 to present day, but this decreases
significantly into the future.

Line-by-line comments.
Lines 39: "its interaction with CO2" does not sit well with me because CO2 and O3 are not
directly interacting — rather there are interactive effects of CO2 and O3. Please revise.

This has been amended (Line 39):

"We conduct our analysis specifically for the European region to quantify the impact of the interactive effects of
tropospheric $O_3$ and $CO_2$ on gross primary productivity (GPP) and land carbon storage across Europe."

Lines 44-46: At this point in the abstract the readers don't know what high and low sensitivity
are

This has been amended (Line 45-46):

"This alleviation of $O_3$ damage by $CO_2$ induced stomatal closure was around 1 to 2% for both land carbon and
GPP, depending on plant sensitivity to $O_3$."

Line 62: citation should be parenthetical

This has been changed.

Line 86: authors should define what they mean by background ozone

We define background ozone (Line88-89):

"Background $O_3$ is generally defined as the $O_3$ pollution present in a region that is not attributed to local
anthropogenic sources (Vingarzan, 2004)."

Line 94: "long-range transport of ozone"

This has been changed (Line 96).

Line 100: why are there two "e.g."s? What is Sicard et al. (2013) a reference here for?

This has been changed (Line 102).

Line 103: I would suggest cutting "currently have poor emission controls"

This has been removed (Line 105).

Line 108-109: Is there a lot of transport of ozone into Europe? I think references are needed here, or this line should be cut.

Two references were added here (Line 111):

(Auvray and Bey, 2005;Derwent et al., 2015)

Line 112-113: Having this statement here could be misleading — for example, if a reader thought the impact of ozone on vegetation was through it's ability to trap heat. It would be more clear if after "direct radiative forcing of ozone", the authors added ", a potent greenhouse gas, "

This sentence has been removed, and suggested changes made (Line 113 to 120):

"Rising background $O_3$ concentrations impact agricultural yields and nutritional quality of major crops (Ainsworth et al., 2012;Avnery et al., 2011), with consequences for global food security (Tai et al., 2014).  Increasing background levels of $O_3$ are damaging to ecosystem health and reduce the global land carbon sink (Arneth et al.,

2010;Sitch et al., 2007). Reduced uptake of carbon by plant photosynthesis due to $O_3$ damage allows more $CO_2$

to remain in the atmosphere. This effect of $O_3$ on plant physiology represents an additional climate warming to the direct radiative forcing of $O_3$, a potent greenhouse gas (Collins et al., 2010;Sitch et al., 2007), the magnitude of which, however, remains highly uncertain (IPCC, 2013)."

Lines 112-114: I find this a bit confusing - the authors discuss "high levels" and "elevated"

ozone, but in the paragraph prior discuss mostly background concentrations. Can the authors make the transition a little smoother?

We have amended this, please see the paragraph above (Lines 113-120).

Lines 113: Please cut "future concentrations of ozone predicted for 2050"

This has been removed and replaced with (Line 135):

"The Wittig et al. (2007) meta-analysis of temperate and boreal tree species showed raised $O_3$ concentrations significantly….."

Line 139: Here the authors refer to 46 ppb as "elevated" - in the previous sentences much higher concentrations are used. I would recommend just giving the concentrations, not qualifying them here and elsewhere

As recommended by the reviewer, we just give the concentrations of $O_3$ and do not qualify them (Lines 134 -

148).

Line 153: What's the time frame for Lombardozzi?

This has been added (Line 156):

"A second study by Lombardozzi et al. (2015) predicted a 10.8% decrease of present-day (2002-2009) GPP
globally."
Line 194: please define ozone dose-response relationship
We have modified this sentence as follows (Line 197-199):
"We make a separate distinction for the Mediterranean region where possible because the work of Büker et al.
(2015) showed that the sensitivity of dominant Mediterranean trees to $O_3$ is different to temperate species."
We remove mention of dose-response relationships here as it is not necessary at this point, and these are
introduced and defined later in section 2.2 lines 279 – 284.
Line 220: I thought the authors re-arranged the supplementary figures to reflect the order
they
are mentioned in the text? This is the first occurrence of a supplementary figure and it's
figure
S5.
Apologies, this was an oversight and has now been changed.
Line 227: of "stomatal ozone deposition"
This has been modified (Line 231).
Line 231-233: "because the impact of cumulative ozone exposure on plant productivity has
already been calibrated with observations (described below)"
This sentence has been changed (Line 234-236).
Line 240: Can this be changed to interactive effects of CO2 and O3?
This has been changed (Line 245).
Line 225: Reference is missing the year - here and in line 332 - also I'm not sure why there
is a
comma before the parenthetical citation
The citation references data downloaded from a web page. We have included the year the data was downloaded
and removed the comma's (Lines 332 - 339).
Line 288: But how is the actually PODy/ FO3crit determined? Please specify here
The values for PODy/ FO3crit are taken from the observation-based dose-response relationships. As to how
these values are determined is a literature in itself and beyond the scope of this paper to describe here, so we
refer the readers to the relevant papers (Line 294):
"For JULES, $Fo_{3crit}$ is the threshold for $O_3$ damage, and values for this parameter are taken from the $O_3$ dose-
response relationships as the $POD_y$ value (see CLRTAP, 2017 and Buker et al. 2015 for derivation of $POD_y$
values)."
Line 298: Is "a" the ozone plant sensitivity? Please specify here
This isn't referring to the parameter '$a$' here, so has been removed to stop confusion (Line 305). The parameter
is always referred to as '$a$'.
Line 302: It's unclear to me how the authors incorporate the work of Büker et al. from this paragraph.

The work of Büker et al. (2015) shows that different $O_3$ dose-response relationships are needed to describe the
$O_3$ sensitivity of dominant Mediterranean trees, so we use the different functions provided for Mediterranean
trees instead of applying the function that has been derived dominantly from temperate/boreal tree species.

Line 380: The g0 term has not previously been defined. I would just say a version of the
Medlyn
(2011) model that does not have an intercept

We amend the sentence as follows, as when there is no g0 term the intercept is forced through zero (Line 387):

"The $g_1$ parameter represents the sensitivity of $g_s$ to the assimilation rate, i.e. plant water use efficiency, and was
derived as in Lin et al. (2015) by fitting the Medlyn *et al*., (2011) model to observations of $g_s$, photosynthesis,
and VPD, assuming an intercept of zero."

Line 416: "of" is missing. Can interaction be changed to interactive effects?

This has been changed (Line 423):

".......to allow us to focus on the impact of $O_3$, $CO_2$ and their interactive effects."

Line 431: phenology is misspelled

This has been changed (Line 439).

Lines 460-474: The point I wanted clarified here is that the agricultural mask does not
change
from 1900 to 2100

We have added the following sentence to clarify this point (Line 431 - 432):

"The agricultural mask is fixed and does not change over the simulation period."

Lines 517 & 519: I don't think an equation for the percentage change is necessary.

This has been removed (Line 524 - 528).

Figure S5 - Tg of N in b)? Tg Carbon in c)?, Gg of carbon in d)? Please specify. Are the NOx
numbers for anthropogenic sources or anthropogenic and natural? What about NMVOCs?

Units have been added to Figure S6.

Table 1 - One of the "O3"s actually reads "O2"

This has been changed.

Line 829: I would not say there are large improvements

'Large' has been removed (Line 777).

Lines 841-855: A discussion of how incorporating ozone damage into JULES leads to a
worse
agreement with the MTE product & relevance for the authors' work is needed here

We have added the following to the discussion (Lines 803-810):

"In general, incorporating plant $O_3$ damage into JULES leads to worse agreement with the MTE GPP product, however, this is expected to some degree as we are adding an explicit representation of $O_3$ damage to a model calibrated to reproduce current day GPP and draw down of atmospheric $CO_2$. Inevitably this implicitly includes $O_3$ damage to vegetation. Explicit representation of plant $O_3$ damage is important to investigate how $O_3$ damage changes through time, under different emissions scenario's, and the interactive effects with other gases (such as $CO_2$) and with climate change. The percentage changes we simulate are therefore important to demonstrate the sensitivity of modelled GPP and land Carbon to this process."

Line 860-1: Can the authors give the time frame here? Does the range represent high vs. low
sensitivity? Is this for O3 +CO2?

This sentence has been clarified (Line 814-815):

"Our estimates suggest $O_3$ (simulation O3) reduced GPP by 2001 by 3% to 9% on average across Europe and NPP by 5% to 11% for the low and high plant $O_3$ sensitivities respectively (Table S3)."

Line 870: "Simulated ozone impacts will dependent on model ozone concentrations, meteorology, plant sensitivity to ozone, and process representation of ozone damage"

Thank you for the simplification! This has been changed (Line 823-825).

Lines 857-876: This section title is a bit of an oversell if the authors can't explain the differences. How can the authors "expect" results if they end of concluding the studies are so different anyway?

We have changed the title of this section to (Line 812):

**"4.2 Comparison of modelled estimates of $O_3$ damage"**

Line 907: It seems like here the authors need to discuss the caveat that both high and low sensitivity simulations underpredict GPP

This has already been addressed in the discussion relating to the comparison of the JULES simulations to the MTE product (Section 4.1 Lines 787 to 808).

Lines 909-910: "may dampen"

This has been changed (Line 863).

Line 941: Can this be changed to interactive effects of O3 and CO2?

This has been changed (Line 970).

Line 884-886: please cut this discussion - the authors' point is made - the information is lacking.

This has been removed (Line 902-905).

Line 888-908: The authors need to spell out the transition at the beginning of this paragraph —
i.e., that another caveat of their study is that ozone is offline and the depositional sink is different here and in the model that was used to create forcing dataset. The comparison of the
two gmax values is an apples-to-oranges comparison (one is model input, one is model output!) and I think it should be cut. I find the rest of the discussion not appropriate here - it's is already discussed in the methods. Please clearly state the caveat and cut most of this discussion (i.e., after Lines 891-908).

This paragraph has been changed as follows (Lines 908 - 915):

"A further caveat of this study is that the $O_3$ concentrations used to force the model are offline, in this case
generated by the EMEP MSC-W model. This means the depositional sink is different in JULES (Medlyn
formulation), compared to the EMEP model which uses the $g_s$ formulation presented in Emberson et al. (2000)
and Emberson et al. (2001). Because we link two different model systems, the $g_s$ values in the EMEP model
differ from those obtained using the Medlyn formulation, which would ultimately lead to different $O_3$
concentrations. The role of EMEP in this study is to provide $O_3$ concentrations at the top of the vegetation
canopy to force JULES and not $g_s$, how the different depositional sinks would affect simulated $O_3$
concentrations at canopy height has not been investigated."

Line 933-934: Why are the future tropospheric ozone concentrations highly uncertain?

Future tropospheric $O_3$ concentrations will depend in a large part on how emissions of $O_3$ precursors change
locally, regionally and globally. This is uncertain.